# A shared speed encoding model for running and backing away behaviours in segregated neural circuits

Jiajia Chen [1,2,3,4,7], He Li [1,2,3,4,7], Na Lian[1,2,3,4], Linhui Yao[1,2,3,4], Yumei Huang[1,2,3,4], Peiran Yin[5], Ziyi Xu[1,2,3,4], Yingying Zeng[5], Jinhua Liu[6], Mingyu Fu[1,2,3,4], Xiaoxia Qin[6], Xuying Ji[1,2,3,4], Jie Tan[5], Zhongju Xiao [1,2,3,4,6,8] ✉ & Wen Zhong [5,8] ✉

How neuronal firing within a circuit encodes behavioural intensity, like running speed, is largely unknown. Projections from temporal association cortex (TeA) and superior colliculus (SC) to dorsal periaqueductal grey (dPAG) circuit can both trigger running behaviour. Using in vivo loose-patch recordings with circuit manipulations in mice, we quantified a firing - speed relationship and established its encoding model. Here, we report two behavioural patterns induced by circuit activation: backing away and rebound running. Mechanistically, dPAG CaMKIIα neurons receiving inputs from either TeA or SC, function as distinct "behavioural units", controlling two "unit behaviours" of running and backing away, respectively. The unidirectional inhibition from the backing away unit to the running unit is mediated by somatostatin (SOM) neurons in dPAG, enabling transitions among four behavioural states: running, backing away, stopping, and rebound running. Both running and backing away behaviours follow a unified motor encoding model, quantified by a single-phase association equation.

Motor control in mammals adheres to a classical hierarchical principle[1,2]. The periaqueductal gray (PAG) serves as a critical hub that transforms high-level behavioural intentions into specific motor outputs[3,4], yet the mechanisms by which it processes and integrates upstream instructions remain unclear.

Our previous work[5] has provided an ideal model system for this investigation: the temporal association cortex (TeA) is a critical node that drives running in sensory-evoked escape. Within this framework, we identified a direct TeA to dorsal periaqueductal grey (dPAG) circuit sufficient to drive running behaviour and, importantly, a class of "running-related neurons" (R-neurons) in TeA whose firing is insensitive to sensory stimuli but correlated with running speed. Within this hierarchical framework of an upstream driver and a downstream actuator, we can investigate how information is quantitatively transformed into motor output at the cellular level.

Based on this, the present study aims to elucidate the quantitative control logic and functional integration architecture within the motor control hierarchy. We begin by investigate: in the TeA-dPAG pathway, is there a common quantitative encoding rule shared between the neural activities of the upstream (TeA) and downstream (dPAG) components?

[1]Department of Physiology, School of Basic Medical Sciences, Southern Medical University, Guangzhou, Guangdong, China. [2]Key Laboratory of Psychiatric Disorders of Guangdong Province, Southern Medical University, Guangzhou, Guangdong, China. [3]Guangdong-Hong Kong-Macao Greater Bay Area Center for Brain Science and Brain-Inspired Intelligence, Southern Medical University, Guangzhou, Guangdong, China. [4]Key Laboratory of Mental Health of the Ministry of Education, Southern Medical University, Guangzhou, Guangdong, China. [5]School of Traditional Chinese Medicine, Guangdong Basic Research Center of Excellence for Integrated Traditional and Western Medicine for Qingzhi Diseases, Southern Medical University, Guangzhou, Guangdong, China. [6]The Seventh Affiliated Hospital, Southern Medical University, Foshan, Guangdong, China. [7]These authors contributed equally: Jiajia Chen, He Li. [8]These authors jointly supervised this work: Zhongju Xiao, Wen Zhong. ✉e-mail: xiaozj@smu.edu.cn; zhong1981@smu.edu.cn

Does this rule constitute a causal mechanism for driving behaviour? Furthermore, as a key hub for integration and execution[6–10], dPAG also receives various upstream inputs[3,4], such as from the superior colliculus (SC)[11,12], which likewise mediates escape running behaviour. This raises a deeper architectural question: in the SC-dPAG pathway, how is the upstream signal from SC transmitted to the dPAG? Does it follow the same quantitative encoding rule? Moreover, do distinct signals from TeA and SC converge onto the same neuronal population to execute the same behaviour, or are they processed via functionally segregated microcircuits to achieve coordinated or distinct behavioural outputs?

By combining in vivo loose-patch recordings, optogenetics, and chemogenetic manipulations targeting the TeA, dPAG, and SC. We have revealed a quantitative encoding of speed in the firing of both upstream and downstream neurons within the TeA-dPAG pathway. More importantly, we have discovered a functionally segregated microcircuit architecture within the dPAG, determined by input origin. This architecture, through segregated excitatory neurons and specific inhibitory connections, achieves a decoupling of the quantitative magnitude (speed) and qualitative behaviour (action selection) of motor signals, providing a principled framework for understanding how brain motor hubs organize multiple behavioural outputs.

## Results

### Quantification of TeA R-neuronal firing–running speed relationship by a single-phase association equation model

We performed in vivo loose-patch recordings of TeA neurons in awake running mice[13] exposed to random sound, light, and air puff stimuli to explore whether the firing of R-neurons quantified running speed (Fig. 1a, left panel). The recorded neurons were labelled with biocytin for localization (Fig. 1a, right panel). Sensory stimuli (Fig. 1b, middle panel) were applied to identify running-related neurons, which were defined as neurons whose firing rates were correlated with running speed (Fig. 1b, bottom panel) (Supplementary Movie 1), rather than with the parameters of the sensory stimuli[5]. Neuronal firing curve (dashed line) preceded running curve (solid line) (Fig. 1b, bottom panel). Based on the sensory stimulus–running time window established during sensory stimulation-induced running events on the turntable[5], we selected events where sensory stimulation did not successfully elicit running events. The time points of each sensory stimulation were aligned, and the example neuron did not respond to any sensory stimuli (Fig. 1c). After confirming that the neuron did not respond to the sensory stimuli, we further investigated the relationship between the firing and running. We selected events where the animal ran spontaneously without sensory stimulation, aligned the time points of running onset, and examined neuronal activity prior to the initiation of running. As shown in Fig. 1d, a consistent increase in the firing rate was observed before the onset of spontaneous running. The firing–running time window (−1.805–0 s) was defined by an analysis of the peri-event time histogram (PETH) (see the "Methods"). Based on this criterion, we performed the same analysis on all the neurons recorded in the TeA, identified R-neurons (Fig. 1e, f), and determined the firing–running time window (−3.01–0 s) for these R-neurons (Fig. 1f). We analysed a total of 36 R-neurons, which were obtained from a total of 132 neurons recorded in the TeA of 20 mice. Based on the PETHs for each neuron, the time difference between firing and spontaneous running onset was calculated. The results show that the average time difference is $2.225 \pm 0.582$ s (Fig. 1g). We recorded primarily from the rostral region of the TeA (Fig. 1h, left panel), in which R-neurons were predominantly located in the layer V of the cortex (Fig. 1h, right panel). These neurons exhibited a characteristic morphology, with long dendrites extending vertically into the superficial layers (Fig. 1i), consistent with the morphology of TeA neurons that project to dPAG (TeA$_{dPAG}$) neurons reported in our other study[5].

We conducted a cross-correlation function (CCF) analysis between the two curves to explore the potential correlation between

neuronal firing rates and running speed (Fig. 1j). The CCF analysis of the example neuron revealed that the firing rate and running speed curve achieved the highest alignment when the speed curve was shifted forward in time. The maximum correlation coefficient ($r_{max} = 0.898$) plateaued as a stable peak across a time lag range from $\tau = 1.6$ s to $\tau = 1.8$ s, indicating a positive correlation between the two variables. The temporal range covered by the peak distribution was relatively broad (full width at half maximum, FWHM = 10.570 s). Together with the optimal correlation plateau spanning 0.2 s, these findings indicate that the relationship between neural activity and running speed parameters is sustained rather than locked to a specific moment in time. This pattern supports the hypothesis of running speed encoding, wherein the firing rate of neurons is continuously modulated throughout both the preparation and execution phases of movement to reflect speed information[5]. We plotted the relationship between the firing rate and running speed at the zero-shift point (i.e., without shifting) to observe the correlation between firing rate and running speed (Fig. 1k). The plot shows that as neuronal activity increased, the animal's running speed also gradually increased, and a significant correlation ($r = 0.844$) was identified. To investigate the quantitative relationship between neuronal firing and speed, we time-shifted the running speed curve using the optimal lag value corresponding to the $r_{max}$ determined by the CCF (see Fig. 1j). This aligned the neural firing rate with the speed value that it was most strongly correlated with. We then re-plotted the firing rate against the time-shifted speed and explored the quantitative functional model between them (Fig. 1l). Although previous studies using calcium imaging and multiunit recordings reported a linear relationship between running speed and the activity of dPAG neurons[12,14,15], this relationship is somewhat counterintuitive. While the linear function also demonstrates a high goodness of fit (gray dashed, $R^2$ for linear function = 0.814, $P < 10^{-4}$), from a biological perspective, an animal's running speed cannot increase indefinitely. Furthermore, the observed trend in Fig. 1l suggests that each movement reaches a maximum speed. Therefore, a saturating nonlinear equation better captures the underlying neurobehavioral relationship in biological contexts. We employed a single-phase association equation (Fig. 1l, red line: fitted curve) that captured the observed trend for fitting well (red line, $v = v_m \times \left(1 - e^{-\frac{(f - f0)}{\tau}}\right)$, where $f_O$, $v_m$ and $\tau$ represent the initial firing rate, maximum running speed, and neuronal firing effect constant, respectively. The parameter $\tau$ specifically indicates the extent of the increase in firing ($f - f_O$) when $v = v_m \times 63.2\%$ (Fig. 1l).

The CCF analyses of all TeA recorded R-neurons (Fig. 1m) further demonstrated a strong and relatively stable positive correlation ($r_{max} = 0.872 \pm 0.054$) between firing and running speed (Fig. 1n). The optimal time lag ($2.064 \pm 0.273$ s) was identified as the lag at which the CCF reached its maximum value (Fig. 1o), which remained within the time window (−3.01–0 s) defined by the PETH analysis (Fig. 1f). This finding indicates that changes in discharge activity still precede behavioural changes, supporting the hypothesis that discharge activity encodes behaviour with a relatively consistent temporal relationship.

All recorded TeA R-neurons were fitted after being shifted according to the optimal time lag derived from the corresponding CCF analyses (Fig. 1o). It can be observed that all TeA R-neurons could be well fitted using a single-phase association equation (Fig. 1p). To examined whether the identified quantitative relationship remained valid under extreme conditions, we performed a peak-to-peak analysis to test the quantitative functional model in its most stringent and simplest form. This examined whether a neuron's maximum firing rate consistently corresponded to the highest running speed it encountered, thereby validating the relationship at the extremes of the physiological range. We obtained the peak firing rate and running speed for each neuron, respectively. The resulting peak firing rate-peak running speed plot could still be robustly fitted by the single-phase association equation, and this fit was superior

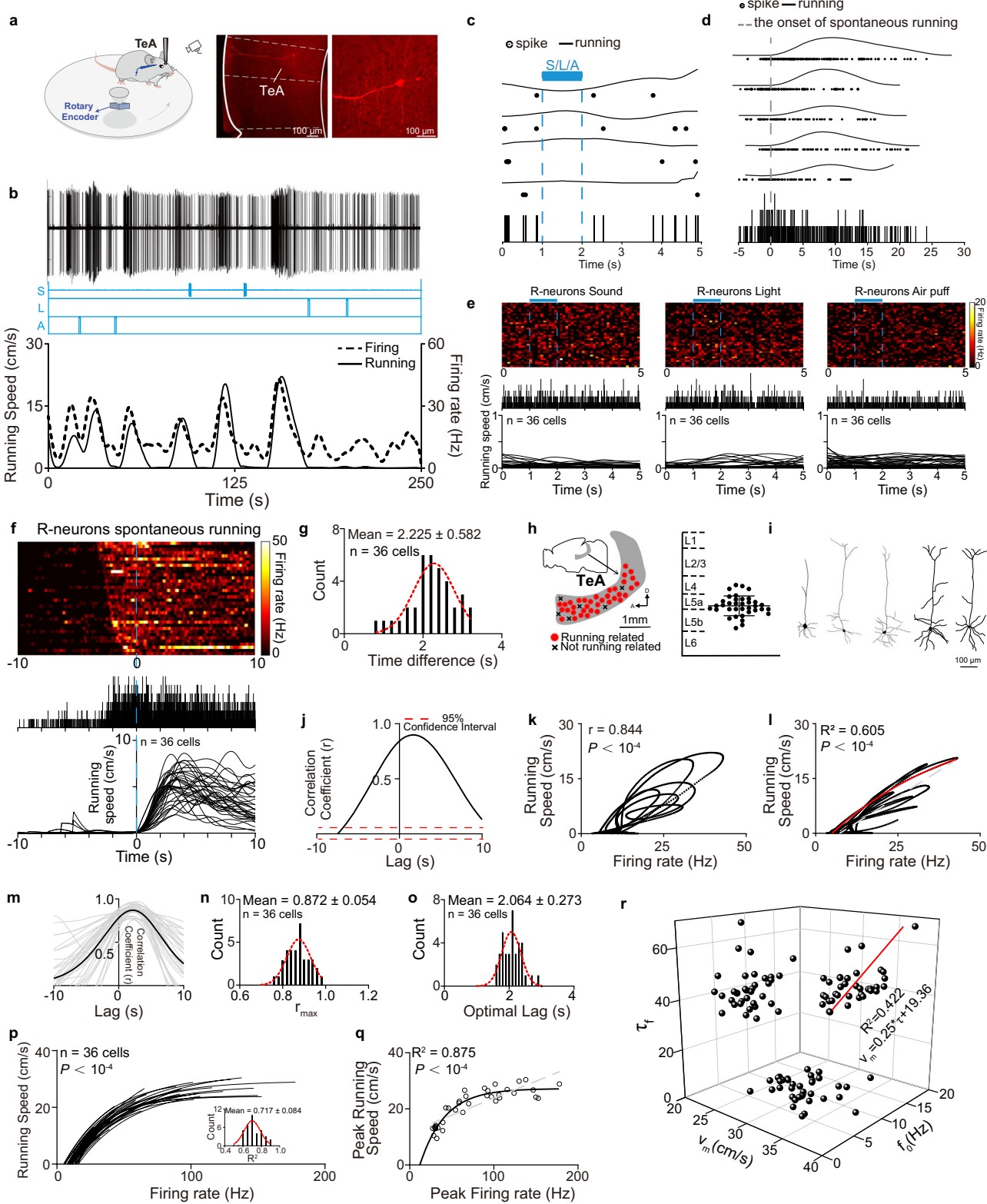

to that of a linear equation (gray dashed, $R^2 = 0.694$, $P = 2.92 \times 10^{-10}$), further confirming the reliability of the speed-quantitative model (Fig. 1q). In the fitted curves of all TeA R-neurons (Fig. 1p), the parameter $\tau$ for the running–firing function ($\tau_f$) showed a significant positive correlation with $v_m$, whereas no significant correlations were observed between $f_O$ and either $\tau_f$ or $v_m$ (Fig. 1r).

These results indicate a sustained and quantitatively covarying relationship between the firing rate of TeA-R neurons and the animal's

running speed. We quantified this relationship using a single-phase association equation, and this model robustly describes the dataset.

## dPAG R-neurons show a more sensitive firing rate-speed relationship than TeA R-neurons

TeA$_{dPAG}$ neurons drive dPAG neurons that receive projections from the TeA ($_{TeA}$dPAG)[5]. We subsequently identified dPAG R-neurons (Fig. 2a) and analysed the relationship between their firing rates and running

**Fig. 1 | Characteristics of TeA running-related neurons. a** Left, recording setup in awake, running mice. Right, a biocytin-labelled running-related neuron (R-neuron). **b** A representative R-neuron. S sound, L light, A air puff. **c** Representative neuron responses to three sensory stimuli. Top: raster plots of individual trials and running speeds. Bottom: peristimulus time histogram (PSTH) of firing ($n = 4$ trials). Blue bar: stimulus duration. **d** Same as (**c**) but for spontaneous running. Bottom: peri-event time histogram (PETH) of firing aligned to running onset ($n = 5$ trials). **e** Sensory responses in R-neurons. Top: firing heatmaps, no-running trials. Middle: population PSTHs. Bottom: per-neuron average running speed. Blue bars: 1 s stimulus. **f** Spontaneous running activity. Top: firing heatmaps aligned to running onset (blue dashed). Middle: population PETHs. Bottom: per-neuron average running speed. **g** Distribution of firing-spontaneous running time differences (from PETHs) (red dashed, Gaussian fit, $R^2 = 0.770$, $P = 4.95 \times 10^{-6}$, $F$-test). **h** Recording sites (left) and depth distributions (right) of recorded R-neurons. D dorsal, A anterior.

**i** Morphological reconstructions. **j** Representative neuron cross-correlation function (CCF). Peak correlation ($r_{max}$) was significant ($P < 10^{-4}$, two-sided permutation test; $n = 1000$ shuffles). **k** Original running speed vs. firing rate (Pearson's correlation, two-sided $t$-test). **l** Running speed vs. firing rate. Nonlinear fit (red) and linear fit (gray dashed), $F$-test; aligned at optimal lag. **m** CCF analyses for all R-neurons. Population mean (black) and individual CCFs (gray). **n** Population distribution of $r_{max}$ (red dashed, Gaussian fit, $R^2 = 0.791$, $P = 2.68 \times 10^{-6}$, $F$-test). **o** Population distribution of optimal time lag (red dashed, Gaussian fit, $R^2 = 0.757$, $P = 1.32 \times 10^{-7}$, $F$-test). **p** Population fits of firing rate to running speed (all fits significant). Bottom right: distribution of individual fit $R^2$ values (red dashed, Gaussian fit, $R^2 = 0.759$, $P = 0.008$, $F$-test). **q** Nonlinear (solid) and linear (dashed) regression fit, $F$-test. **r** Three-dimensional scatter plot. Linear fit (red line), $P = 1.79 \times 10^{-5}$, $F$-test. Data are presented as mean ± SD. Supplementary Data 1 for detailed statistics. Source data are provided as a Source Data file.

speed, as shown in Fig. 1. Similar to the observations for TeA R-neurons, a relationship was observed between running speed and the firing rate of a dPAG neuron (Fig. 2b). The example R-neuron (Fig. 2b) recorded in the dPAG region did not respond to any sensory stimuli (Fig. 2c) but consistently exhibited increased neuronal activity prior to the initiation of spontaneous running (Fig. 2d), with a firing–running time window of 0.960–0 s defined by the PETH analysis. We also performed the same analysis on all the neurons recorded in the dPAG (Fig. 2e, f) and determined the firing–running time window (−1.164–0 s). The time difference between dPAG firing and running onset (1.069 ± 0.277 s), calculated from PETHs, was significantly shorter than that of TeA neurons (2.225 ± 0.582 s) (Fig. 2g).

The relationship between the firing rate of the example neuron (Fig. 2b) and the running speed was analysed using CCF analysis (Fig. 2h). The results showed a high degree of alignment between neuronal firing rate and running speed, with the maximum correlation coefficient ($r_{max} = 0.734$) plateauing as a stable peak across a time lag range from $\tau = 0.6$ s to $\tau = 0.8$ s, indicating a positive correlation between the two variables. Firing exhibited sustained encoding of running speed (FWHM = 6.810 s). We plotted the relationship between the neuronal firing rate and animal running speed using the zero-shift point from the CCF analysis (Fig. 2i). Similarly, as neuronal activity increased, the animal's running speed gradually increased, and a significant correlation ($r = 0.717$) was identified. Consistent with the approach used in the TeA, we optimized the data by applying time shifts based on optimal time lag (Fig. 2h). The resulting depicting the relationship between the firing rate and running speed also conformed to the single-phase association equation (Fig. 2j).

We analysed a total of 30 R-neurons obtained from 77 recorded neurons in the dPAG region across 20 mice. The CCF analyses performed on all recorded dPAG R-neurons (Fig. 2k) revealed that, similar to TeA neurons, there existed a strong and relatively stable positive correlation ($r_{max} = 0.816 ± 0.042$) between firing and running speed (Fig. 2l). The optimal time lag of the dPAG ($0.903 ± 0.228$ s) (Fig. 2m) was largely contained within the time window (−1.164–0 s) defined by the PETH analysis (Fig. 2f). The firing–running time lag in the dPAG was shorter than that in the TeA (Fig. 2m).

All running–firing curves for the dPAG neurons also fit well when they were modelled using the single-phase association equation (Fig. 2n). Similarly, a peak-to-peak analysis was conducted on dPAG neurons to test whether the quantitative functional model remained applicable in its most stringent and simplest form. The results showed that the resulting peak firing rate–peak running speed plot could still be robustly fitted by the single-correlation equation, and better than that of the linear equation (gray dashed, $R^2 = 0.776$, $P = 1.35 \times 10^{-10}$) (Fig. 2o). The parameter $\tau$ for the running–firing function ($\tau_f$) differed considerably from neuron to neuron, with the mean $\tau_f$ for dPAG neurons being approximately half that for TeA neurons (Fig. 2p). The relationships among $f_O$, $v_m$ and $\tau_f$ for dPAG neurons were similar to those observed for TeA neurons (Fig. 2q).

For both TeA and dPAG R-neurons, the values of $f_O$, $\tau_f$, and $v_m$ calculated through curve fitting differed among different neurons (Fig. 2q). These findings suggest that the quantitative relationship between neuronal activity and running speed is defined by a set of parameters ($f_O$, $\tau_f$, and $v_m$) that are specific to each neuron. This analysis indicates that even when described by the same three-parameter model, individual neurons exhibit significant specificity in their parameter values. This individual variability suggests that the robust representation of running speed likely depends not on a single neuron but on the collective activity of a neuronal population. Based on this, we hypothesize that neurons belonging to the same functional nucleus (such as TeA or dPAG) may achieve coordinated computation by sharing a set of statistically similar model parameters.

We fixed the values of $f_O$ and $v_m$ for all neurons and recalculated the neuronal firing effect constant $\tau$ to compare within a nucleus. Specifically, we reduced the fit parameters and standardized the running–firing curves by setting the $v_m$ values to their mean and $f_O$ to zero (see "Methods"). Notably, all the running–firing functions for TeA and dPAG neurons could be subsequently classified into two distinct groups, each of which displayed good fits between the running speed and the firing rate. The $\tau_f$ in each group converged to a constant value, with the $\tau_f$ for TeA neurons tending to be twice that of dPAG neurons (Fig. 2r). However, the standardization coefficients along the $X$- and $Y$-axes for TeA neurons were not significantly different from those for dPAG neurons (Fig. 2s), indicating that the quantitative principles for TeA and dPAG neurons are fundamentally similar.

These results demonstrate that the quantitative relationships between neuronal activity and running speed in both TeA and dPAG R-neurons conform to a same functional form, yet exhibit systematic differences in key model parameters. Compared with dPAG neurons, TeA neurons require a higher firing effect constant $\tau_f$, suggesting that compared with dPAG neurons, TeA neurons must fire faster to achieve the same running speed (Fig. 2r, dashed horizontal line). Alternatively, for the same firing rate, $v$ is higher for dPAG neurons than for TeA neurons (Fig. 2r, dashed vertical line). Therefore, the quantitative model suggests that the dPAG R-neurons demonstrates greater sensitivity or gain in the relationship between their firing rate changes and changes in running speed compared to the TeA R-neurons.

## Optogenetic activation of the TeA–dPAG pathway drives running speed in a frequency-dependent and quantitative manner

The preceding analyses established a quantitative neuronal activity–speed relationship model in TeA and dPAG R-neurons and revealed parametric differences between them. However, this correlational approach cannot distinguish whether the relationship constitutes a causal driving mechanism or is merely an epiphenomenon. To directly test whether this quantitative model serves as a sufficient causal mechanism for R-neurons to drive behaviour, we conducted optogenetic manipulation experiments. Optogenetic activation was used in subsequent experiments to control these variables and better

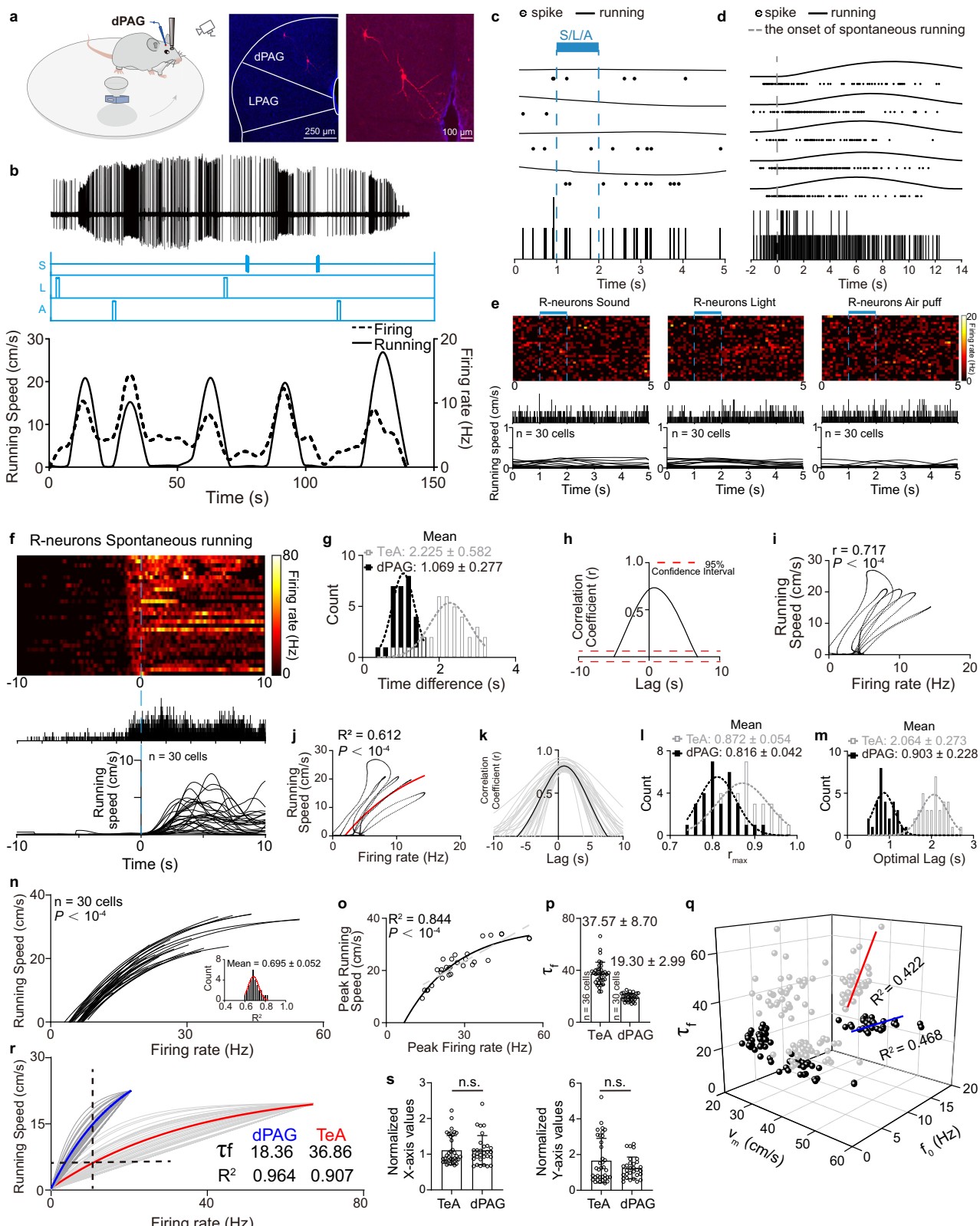

standardize the running speed and the neuronal firing rate (Fig. 2r). Typically, a greater stimulus intensity tends to activate a greater number of cells and produce greater neuronal activity (e.g., higher firing rates). However, altering the stimulation frequency while keeping the intensity constant has been reported to affect neuronal activity, such as firing, without significantly affecting the number of activated cells[16]. We aimed to manipulate these cells optogenetically to explore

whether running speed is encoded by the activity of a specific neuronal population (R-neurons). However, technical limitations prevented the direct labelling and manipulation of R-neurons. Therefore, we turned to a functionally relevant surrogate population.

Based on our previous findings[5] that the inhibition of TeA neurons, TeA$_{dPAG}$ CaMKIIα neurons, TeA-dPAG projections, and $_{TeA}$dPAG neurons effectively halts running behaviours. We selectively

**Fig. 2 | Firing–running functions of dPAG R-neurons. a** Recording setup and a biocytin-labelled R-neuron. **b** A representative dPAG R-neuron, similar to (**1b**). **c** Same as (**1c**), but for representative R-neuron ($n = 4$ trials). **d** Same as (**c**), but for spontaneous running. ($n = 5$ trials). **e, f** Same as (**1e**) and (**1f**), for the dPAG neuronal population (blue bars: 1 s stimulus; blue dashed: running onset). **g** Time differences distributions: dPAG neurons (black) and TeA neurons (gray, from (**1g**); black dashed, Gaussian fit, $R^2 = 0.893$, $P = 0.015$, $F$-test). **h** Same as (**1j**), representative neuron CCF. The $r_{max}$ was significant ($P < 10^{-4}$, two-sided permutation test; $n = 1{,}000$ shuffles). **i** Original running speed vs. firing rate (Pearson's correlation, two-sided $t$-test). **j** Running speed vs. firing rate. Nonlinear fit (red), $F$-test; aligned at optimal lag. **k** CCF analyses for all R-neurons. Population mean (black) and individual CCFs (gray). **l** Population distribution of $r_{max}$. Black dashed, Gaussian fit, dPAG:

$R^2 = 0.829$, $P = 2.37 \times 10^{-5}$, $F$-test. **m** Population distribution of optimal time lag. Black dashed, Gaussian fit, dPAG: $R^2 = 0.767$, $P = 3.8^6 \times 10^{-6}$, $F$-test. **n** Population fits of firing rate to running speed (all fits significant). Bottom right: distribution of individual fit $R^2$ values. Red dashed, Gaussian fit, $R^2 = 0.842$; $P = 6.87 \times 10^{-7}$, $F$-test. **o** Nonlinear (solid) and linear (dashed) fit, $F$-test. **p** Firing–running function $\tau$ values ($\tau_f$). **q** Three-dimensional scatter: dPAG (black, linear fit in blue: $v_m = 1.43 \times \tau + 7.37$, $P = 3.07 \times 10^{-5}$, $F$-test) and TeA (gray, from (**1r**)) neurons. **r** Normalized data for recorded R-neurons with fitted curves (TeA: $P < 10^{-4}$; dPAG: $P < 10^{-4}$, nonlinear fit, $F$-test). **s** Distribution of normalized values. Left, $X$-axis normalized to $f_o =$ zero, $P = {}_o.991$. Right, $Y$-axis scaling by (avg. measured $v_m$/measured $v_m$), $P = 0.105$. Two-sided, two-sample t test. Data are presented as mean ± SD. Supplementary Data 1 for detailed statistics. Source data are provided as a Source Data file.

manipulated both the upstream node (TeA$_{dPAG}$ neurons) and the downstream node ($_{TeA}$dPAG neurons) of TeA-dPAG pathway. This strategy is grounded in our prior findings: TeA$_{dPAG}$ neurons share striking similarities with the TeA R-neurons we recorded, in terms of laminar distribution within TeA, anatomical morphology, ability to directly drive running upon optogenetic stimulation, and tight temporal coupling between their firing and running onset[5]. These similarities indicate that TeA$_{dPAG}$ neurons constitute a subclass of TeA R-neurons. Therefore, manipulating TeA$_{dPAG}$ neurons is functionally equivalent to targeting the population of TeA R-neurons. Manipulation of $_{TeA}$dPAG neurons, in contrast, targets the downstream node of this pathway. These cells directly receive input from TeA$_{dPAG}$ neurons and constitute the downstream actuator neurons responsible for transforming TeA signals into behavior. The dPAG R-neurons we recorded under natural conditions exhibit a quantitative neuronal activity-speed relationship highly consistent with that of TeA R-neurons (Fig. 2), strongly suggesting that they likely belong to the same, or a functionally overlapping, population targeted by TeA R-neurons.

Using viral injection strategies, we achieved specific labelling and manipulation of both TeA$_{dPAG}$ and $_{TeA}$dPAG neurons and optogenetically activated them at varying frequencies to probe their encoding rules for running speed. We begin by expressed channelrhodopsin-2 (ChR2) in TeA CaMKIIα neurons by injecting adeno-associated virus (AAV)-CaMKIIα-hChR2 and activated these neurons with blue light at frequencies of 1, 5, 10, 15, 20, 25, 30, 35, and 40 Hz, which were presented pseudorandomly across the TeA (Fig. 3a). This broadly confirms, at the population level, that the TeA region is capable of issuing the control signals, thereby reaffirming that TeA is both necessary and sufficient for generating the running behaviour. However, this approach would excite all ChR2-expressing TeA CaMKIIα neurons, including those that did not project to the dPAG. Blue light was delivered locally to the dPAG to selectively activate only fibres projecting from the TeA to the dPAG (TeA$_{dPAG}$ fibres) (Fig. 3b). Subsequently, TeA CaMKIIα neurons projecting to dPAG neurons (TeA$_{dPAG}$ CaMKIIα neurons) were specifically targeted by injecting AAVretro-CaMKIIα-hChR2 into the dPAG (Fig. 3c). However, as the method might also affect neurons projecting to regions other than the dPAG, AAVretro-Cre was injected into the dPAG and AAV-DIO-hChR2 was injected into the TeA to restrict ChR2 expression exclusively to TeA$_{dPAG}$ neurons (Fig. 3d). Finally, AAV-Cre and AAV-DIO-hChR2 were injected separately into the TeA and dPAG to selectively activate $_{TeA}$dPAG neurons (Fig. 3e). We used anterograde transsynaptic AAV-Cre viruses designed to drive the expression of Cre in postsynaptic neurons that receive input from the TeA.

The animals responded to blue light stimulation by running, the maximum speed of which increased as the stimulation frequency increased (Fig. 3f–j); however, none of these behaviours were observed in the control group (Fig. 3f–j). Curves of the average maximum running speed versus the stimulation frequency (running–stimulation function, 10 trials) were well fitted with a single-phase association equation (Fig. 3k–o). This indicates that the specific pattern of neuronal activity described by our quantitative model is sufficient to cause

running at predictable speeds. Therefore, these results provide direct causal evidence supporting the hypothesis that running speed is encoded in the firing rate of these neurons within the TeA to dPAG pathway.

A comparison of the maximum speeds revealed differences in the driving effects of TeA and dPAG neurons. The $v_m$ for $_{TeA}$dPAG neurons ($35.10 \pm 3.92$ cm/s) was the highest, followed by that for TeA CaMKIIα neurons ($22.81 \pm 3.07$ cm/s). However, the $v_m$ values for TeA$_{dPAG}$ fibres ($9.44 \pm 0.77$ cm/s), TeA$_{dPAG}$ CaMKIIα neurons ($8.6 \pm 0.61$ cm/s), and TeA$_{dPAG}$ neurons ($8.68 \pm 0.53$ cm/s) did not differ significantly (Fig. 3p). Notably, the activation of TeA CaMKIIα neurons differed from that of TeA$_{dPAG}$ fibres, TeA$_{dPAG}$ CaMKIIα neurons, and TeA$_{dPAG}$ neurons, as the latter three involved projections from the TeA to the dPAG, whereas TeA CaMKIIα neurons included TeA$_{dPAG}$ neurons and neurons projecting to TeA$_{dPAG}$ neurons[5]. The neurons projecting to TeA$_{dPAG}$ neurons increased the activation of the latter, explaining why the activation of TeA CaMKIIα neurons evoked faster running speeds than did the activation of TeA$_{dPAG}$ fibres, TeA$_{dPAG}$ CaMKIIα neurons, or TeA$_{dPAG}$ neurons.

The $\tau$ value of the running–stimulation function ($\tau_s$) for $_{TeA}$dPAG neurons ($10.08 \pm 2.664$) was significantly lower than those of the other groups (Fig. 3q), whereas no significant differences were observed among the $\tau_s$ values of the other groups (TeA CaMKIIα neurons: $14.32 \pm 2.314$, TeA$_{dPAG}$ fibres: $16.05 \pm 1.582$, TeA$_{dPAG}$ CaMKIIα neurons: $15.04 \pm 1.714$, TeA$_{dPAG}$ neurons: $14.98 \pm 2.106$). A direct comparison between TeA$_{dPAG}$ neurons and $_{TeA}$dPAG neurons revealed differences in their ability to drive running speed in the TeA–dPAG neural circuit, indicating intrinsic differences between TeA and dPAG neurons.

We converted the stimulation frequency to the firing rate to enable a direct comparison between the dynamics obtained via optogenetic activation ($\tau_s$, Fig. 3q) and those from the running-firing function ($\tau_f$, Fig. 2r), which were measured on different abscissae (stimulation frequency vs. firing rate). Therefore, we selectively recorded the firing rates of TeA CaMKIIα ChR2$^+$ (ChR2-expressing) neurons (Fig. 3a), dPAG ChR2$^-$ (non-ChR2-expressing) neurons activated by TeA CaMKIIα fibres ($_{TeA\ fibre}$dPAG) across synapses (Fig. 3b), and $_{TeA}$dPAG ChR2$^+$ neurons (Fig. 3e) during optical stimulation. Meanwhile, by stimulating TeA CaMKIIα ChR2$^+$ neurons and recording the evoked firing rates in their downstream dPAG ChR2$^-$ neurons, we can elucidate how motor signals are synaptically transmitted downstream within this circuit. All the firing rates increased linearly with increasing stimulation frequency; TeA CaMKIIα ChR2$^+$ and $_{TeA}$dPAG ChR2$^+$ neurons responded directly to optical stimulation (Fig. 3r, pink and blue), whereas TeA CaMKIIα ChR2$^+$ neuronal activation drove dPAG ChR2$^-$ neurons to fire across synapses ($_{TeA\ fibre}$dPAG) at a ratio of nearly 6:1 (Fig. 3r, yellow).

TeA CaMKIIα ChR2$^+$ and $_{TeA}$dPAG ChR2$^+$ neurons responded to optical stimulation with a short latency of 1–5 ms (average $2.2 \pm 0.272$ ms) (Fig. 3s, top and bottom panels; Supplementary Fig. 1a). In contrast, dPAG ChR2$^-$ neurons, which were activated across one synapse by the optogenetic stimulation of TeA$_{dPAG}$ ChR2$^+$ fibres, exhibited a longer latency of approximately 1 s ($0.956 \pm 0.137$ s) (Fig. 3s, middle and bottom panels; Supplementary Fig. 1b). This one-synapse latency is similar to the delay

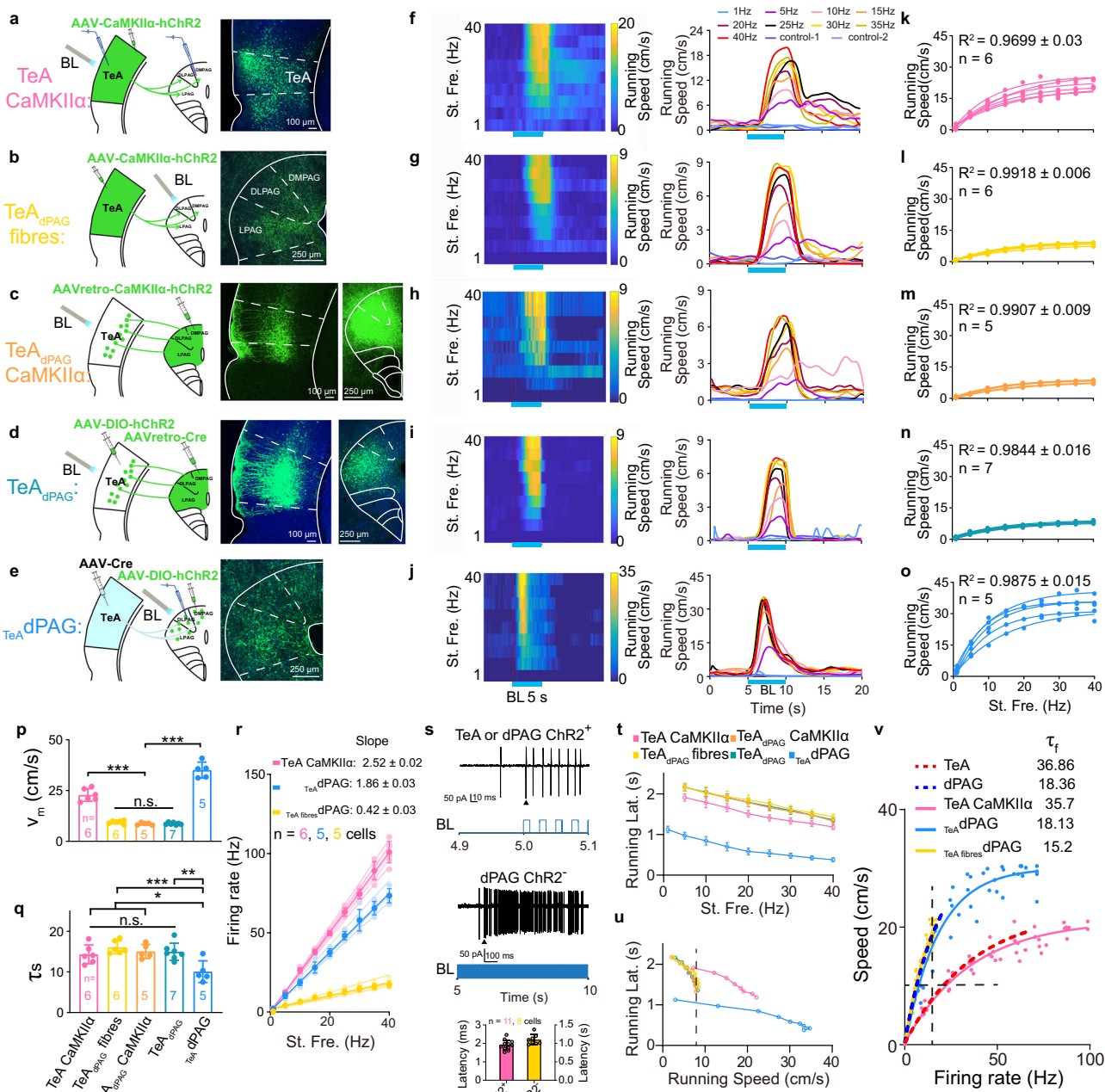

**Fig. 3 | Optogenetic stimulation of TeA–dPAG circuit triggers firing and running. a–e** Experimental schematic: virus injection, opto-activation, recording sites and channelrhodopsin-2 (ChR2) expression in TeA–dPAG circuit. BL blue light. **f–j** Representative example showing behavioural responses vary with stimulation frequency (St. Fre.). Two control groups were included: control-1(injected with EYFP-only virus variants (**a**, **c–e**) to exclude the effects of viral diffusion; control-2 (no virus, blue light only) to control for visual stimulus effects. **k–o** Running speed plotted against St. Fre. (every 10 trials) with fits. *n*: number of experimental animals. **p**, **q** Values of $v_m$ (**p**) and $\tau$ of running stimulation ($\tau_s$) (**q**) obtained from (**k–o**). * $P < 0.05$, ** $P < 0.01$, *** $P < 0.001$, one-way ANOVA with the Bonferroni post hoc correction. **r** Firing rates following stimulation at different frequencies for TeA

CaMKIIα neurons (**a**), TeA$_{dPAG}$ fibres (**b**), and $_{TeA}$dPAG neurons (**e**). Thick lines: average; values: fit slope. **s** In vivo loose-patch recordings from a TeA CaMKIIα neuron expressing ChR2 (ChR2$^+$, top panel) (**a**) and a dPAG neuron not expressing ChR2 (ChR2$^-$) ($_{TeA\ fibre}$dPAG) (middle panel) (**b**). Light pulse duration: 10 ms. Bottom: latency from photoactivation onset to the generation of neuronal action potentials. **t** Running latency vs. St. Fre. (every 10 trials). **u** Running latency vs. running speed. **v** Normalized running speed–firing rate curves for TeA CaMKIIα neurons (**a**), TeA$_{dPAG}$ fibres (**b**), and $_{TeA}$dPAG neurons (**e**) with fitted curves (*n* = 6, 5, 5 cells from 6, 5, 5 mice); dashed lines were obtained from (**2r**). Data are presented as mean ± SD. Supplementary Data 1 for detailed statistics. Source data are provided as a Source Data file.

observed between the TeA and dPAG (1.156 s) shown in Fig. 2g. We confirmed these findings by plotting the latency between the onset of running behaviour and the frequency of optogenetic stimulation (Fig. 3t) and showed that the running latency progressively decreased with increasing stimulation frequency. The differences in timing between $_{TeA}$dPAG neurons and TeA$_{dPAG}$ fibres, TeA$_{dPAG}$ CaMKIIα neurons, and TeA$_{dPAG}$ neurons were equivalent to the latency across one synapse from

TeA to dPAG neurons (approximately 1.147 s) (Fig. 3t), which was close to the delay (1.156 s) observed between the TeA and dPAG shown in Fig. 2g. This long latency across a single synapse from TeA to dPAG neurons suggests slow synaptic effects on the connection between the TeA and dPAG, similar to the findings of other reports[5,12]. TeA CaMKIIα neurons include TeA$_{dPAG}$ neurons and other TeA neurons that innervate the dPAG across multiple synapses and thus need time to activate dPAG neurons;

however, the running latencies evoked by the activation of TeA CaMKIIα neurons were significantly shorter than those evoked by the activation of $TeA_{dPAG}$ fibres, $TeA_{dPAG}$ CaMKIIα neurons, and $TeA_{dPAG}$ neurons. The latency of running behaviour depends on the neurons affected by the location of the injection of the virus; the type, titre and expression of the virus; and the source, intensity and frequency of optogenetic stimulation. Therefore, behavioural data should be normalized to account for differences in viral expression and other factors, as these factors also affect running speed. We converted the optogenetic stimulation frequencies into corresponding running speeds to normalize the behavioural data, as shown in Fig. 3k–o, and plotted the running latency against running speed to eliminate the influence of the latter (Fig. 3u). This plot shows that the running latency gradually decreased as the running speed increased and that the running latencies evoked by the activation of TeA CaMKIIα neurons were significantly longer than those evoked by the activation of $TeA_{dPAG}$ fibres, $TeA_{dPAG}$ CaMKIIα neurons, and $TeA_{dPAG}$ neurons for the same running speed (vertical dashed lines, Fig. 3u).

As shown in Fig. 2r, the firing rates of TeA CaMKIIα, $_{TeAfibre}dPAG$, and $_{TeA}dPAG$ neurons upon optogenetic stimulation were plotted against running speed (Fig. 3v). The running–firing function of $_{TeAfibre}dPAG$ neurons (yellow) nearly overlapped with that of dPAG neurons (dashed blue line, data from Fig. 2r) and the incremental portion of the curve for $_{TeA}dPAG$ neurons (blue). In contrast, the trend of TeA CaMKIIα neurons (pink) was consistent with that of TeA neurons (dashed red line, data from Fig. 2r). The input-output relationship obtained through optogenetic activation closely matched with the neural firing rate-speed relationship observed under natural conditions via loose-patch recordings. This congruence strongly validates the causal relevance of the quantitative model we established from the natural data. Parameter comparisons based on this validated model framework revealed that neurons with the same function have uniform $\tau$ and $v_m$ values; the behavioural encoding efficacy of the dPAG is approximately twice that of the TeA (Fig. 3v, dashed vertical line); and TeA neurons require approximately six times the firing rate to drive dPAG neurons and achieve the same running speed (Fig. 3v, dashed horizontal line).

**Neurons in SC to dPAG pathway follow the quantitative encode speed rule, but drive backing away behaviour**

Running behaviours can also be elicited by activating neurons in the SC that project to dPAG ($SC_{dPAG}$) neurons[11,12]. This raises a fundamental question about the organizational structure of the dPAG as an integrative hub: how does it process different inputs? Therefore, we characterized the SC-dPAG pathway to determine whether it employs the same encoding rule and, more importantly, whether inputs from TeA and SC converge onto the same or separate neurons within the dPAG. After both anterograde transsynaptic serotype AAV systems, Flp and Cre, were used (Fig. 4a), $_{SC}dPAG$ neurons were labelled with mCherry (shown in magenta for clarity), whereas $_{TeA}dPAG$ neurons were labelled in green, indicating that the two populations were distinct (Fig. 4b) with minimal overlap (Fig. 4c, white, 2.02 ± 0.99%). This result raises the question of whether two distinct mechanisms underlie running behaviour. Since most $TeA_{dPAG}$ neurons are CaMKIIα-positive[5], we expressed ChR2 in $SC_{dPAG}$ and $TeA_{dPAG}$ CaMKIIα neurons by injecting AAVretro-CaMKIIα-hChR2 into the dPAG (Fig. 4d). Upon the optogenetic activation of $SC_{dPAG}$ neurons at different stimulation frequencies, we observed frequency-dependent behavioural responses (Fig. 4e, f). Specifically, lower-frequency (< 20 Hz) stimulation elicited stopping (no movement) behaviour during the stimulation period, whereas high-frequency stimulation triggered backing away behaviours. Notably, rebound running was observed after stimulation in both frequency ranges ended. These behaviours were not observed in the control group injected with viral vectors lacking ChR2 (Fig. 4f). Moreover, the stopping and backing away behaviours observed on the turntable (see Supplementary Movie 2) were confirmed in an open field test (see Supplementary Movie 3). As shown in

Supplementary Movie 2, the mice maintained backing away behaviours during stimulation, although sustained backing away was not observed during high-frequency stimulation in either the open field test (see Supplementary Movie 3) or on a turntable (Fig. 4g–j). However, the mice maintained a backing stance, which differed from the stopping posture observed during low-frequency stimulation. Following the stopping behaviours, stimulation at lower frequencies (< 20 Hz) subsequently induced weak, short rebound running (Fig. 4e, f). These observations clearly indicated that stopping was the result of the activation of a specific type of neuron or neural circuit, leading to immobility by ending a motion, and that this behaviour was distinct from the resting behaviours observed after rebound running and normal running (see Supplementary Movies 2 and 4). As the stimulation frequency increased, the speed of both the backing away and the rebound running behaviours also increased.

Backing away had previously been observed accidentally as a form of escape when the deep layer SC (dlSC) to the PAG pathway was stimulated to evoke an escape response[17]. In another study, activation of projections from the deep layers of the medial SC (dmSC) to the PAG triggered running behaviours[12]. The observed differences may be due to differences in the sites of stimulation within the SC or the use of lower stimulation frequencies (e.g., 20 Hz) in these experiments, in which we observed stopping behaviours during stimulation and rebound running, as described above (Fig. 4e, f). When the stimulation duration was brief (e.g., 2 s), the delay in transitioning from stopping to rebound running might have been interpreted as the latency of the running response. We stimulated the SC for 5, 10, and 15 s at 40 Hz to explore this possibility further. The results revealed that rebound running behaviours occurred after the stimulation ended (Fig. 4g, h). Optogenetic activation of $SC_{dPAG}$ neurons typically evoked an initial backing away episode, which was often followed by a predominant period of movement cessation that lasted until the end of the light pulse. However, the detailed analysis revealed that sporadic backing away could occur intermittently during the entire stimulation period (Fig. 4g). Notably, the total duration of the evoked behavioural response (combining backing away and stopping) was positively correlated with the duration of the blue light pulse (Fig. 4i). Additionally, the peak backing away and rebound running speeds did not differ significantly across the three stimulation durations (Fig. 4j). In a comparative analysis, the same stimulation method used for the TeA exclusively elicited running behaviours (Fig. 4k–n) (see Supplementary Movie 4). These findings suggest that $_{TeA}dPAG$ and $_{SC}dPAG$ neurons mediate distinct behavioural responses: $_{TeA}dPAG$ neurons mediate running behaviours, whereas $_{SC}dPAG$ neurons mediate backing away, rebound running, and stopping behaviours.

The speed of rebound running or backing away was subsequently plotted against the stimulation frequency (Fig. 4o). The quantitative analysis revealed that the moving-stimulation functions were well fitted with a single-phase association equation. This indicates that neurons in the SC-dPAG pathway also adhere to the quantitative encode rule. Crucially, the motor output predicted by this rule differs in behaviour (backing away). Notably, the $\tau_s$ values of the $SC_{dPAG}$ CaMKIIα neurons in the induction of backing away ($SC_{dPAG}$ CaMKIIα B: 13.87 ± 1.122) and rebound running behaviours ($SC_{dPAG}$ CaMKIIα R: 8.166 ± 1.005) (Fig. 4p) did not differ significantly from those of the TeA CaMKIIα (14.32 ± 2.314) and $_{TeA}dPAG$ neurons (10.08 ± 2.664) in inducing running behaviours, as shown in Fig. 3q. These findings suggest that the strategies employed by SC and TeA neurons are consistent in terms of efficiency and that rebound running is likely mediated by $_{TeA}dPAG$ cells.

**Cells responsible for rebound running and stopping behaviours**

We expressed ChR2 in $SC_{dPAG}$ CaMKIIα neurons and iDREADD in $_{TeA}dPAG$ neurons to confirm that rebound running is associated with the activation of $_{TeA}dPAG$ cells (Fig. 5a). The inhibitory muscarinic designer receptor hM4Di (iDREADD), which is designed to be

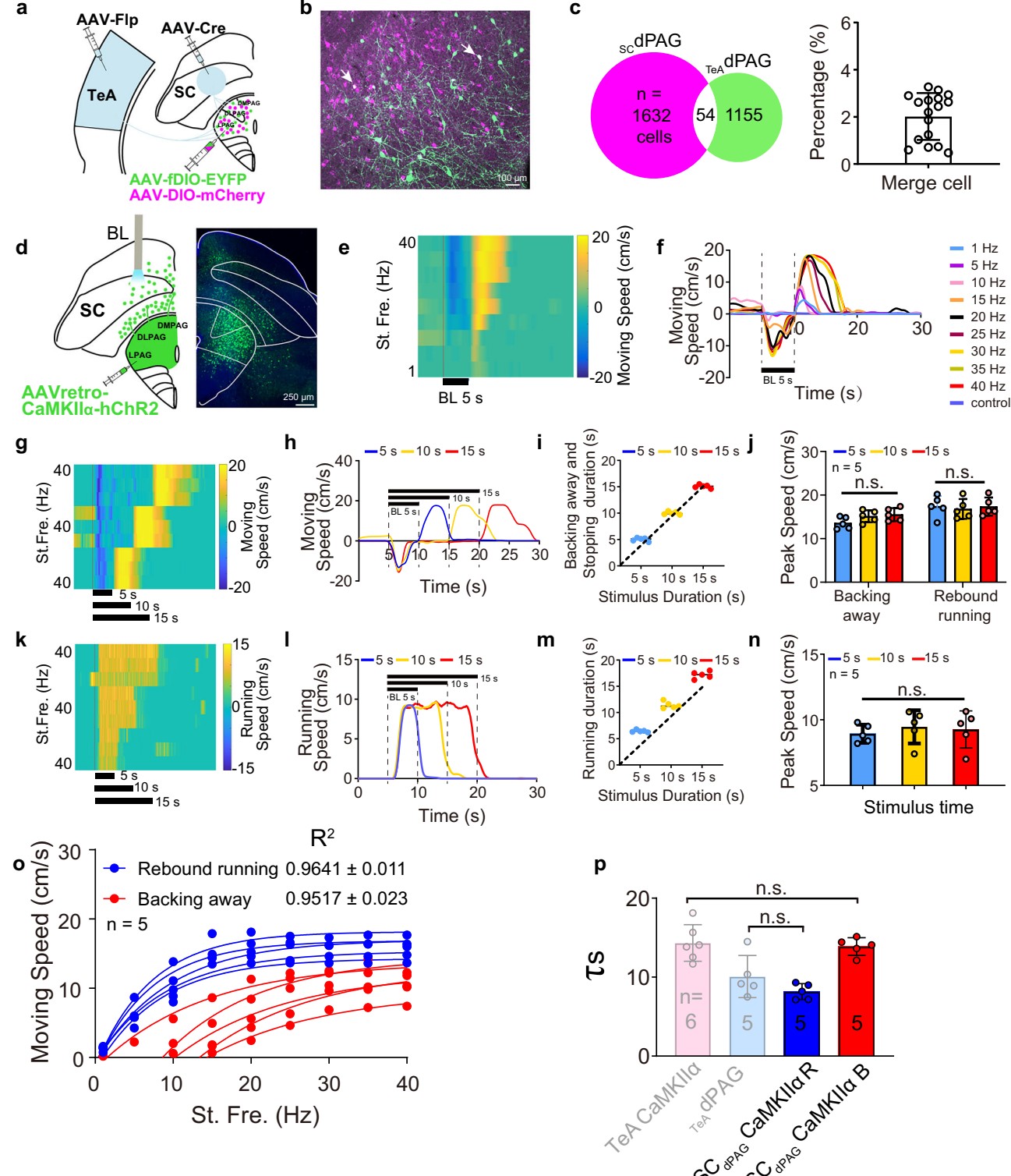

**Fig. 4 | SC_dPAG stimulation. a–c** Virus injection protocol (**a**), _TeA_dPAG (green) and _SC_dPAG neuronal labelling (magenta) (**b**), and statistics of merged cells (white) (**c**). $n = 1632$ and 1155 cells (17 slices, 3 mice). **d** Virus injection and ChR2 expression in the dPAG and SC. **e, f** Movement speed heatmap (**e**) and average speed over time (**f**) of mice stimulated at nine different optogenetic frequencies. **g–n** Behavioural responses to BL stimulation for durations of 5, 10, and 15 s and a frequency of 40 Hz in the SC (**g**) and TeA (**k**). **g, k** Movement speed heatmap, **h, l** average speed versus time, **i, m** response duration versus stimulation duration, **j, n** statistical comparison of the peak speed for backing away or rebound running behaviours. *n*: number of

experimental animals. n.s. non-significant. One-way repeated-measures ANOVA with the Bonferroni post hoc correction. **o** Movement speed–stimulation frequency curves for the backing away (red) and rebound running (blue) behaviours of all experimental mice corresponding to (**d**). **p** SC_dPAG CaMKIIα backing away $\tau_s$ obtained from (**o**) alongside TeA CaMKIIα $\tau_s$ (originally in Fig. 3q), and SC_dPAG CaMKIIα rebound running $\tau_s$ alongside _TeA_dPAG $\tau_s$ (originally in Fig. 3q). Two-sided, two-sample *t* test. Data are presented as mean ± SD. Supplementary Data 1 for detailed statistics. Source data are provided as a Source Data file.

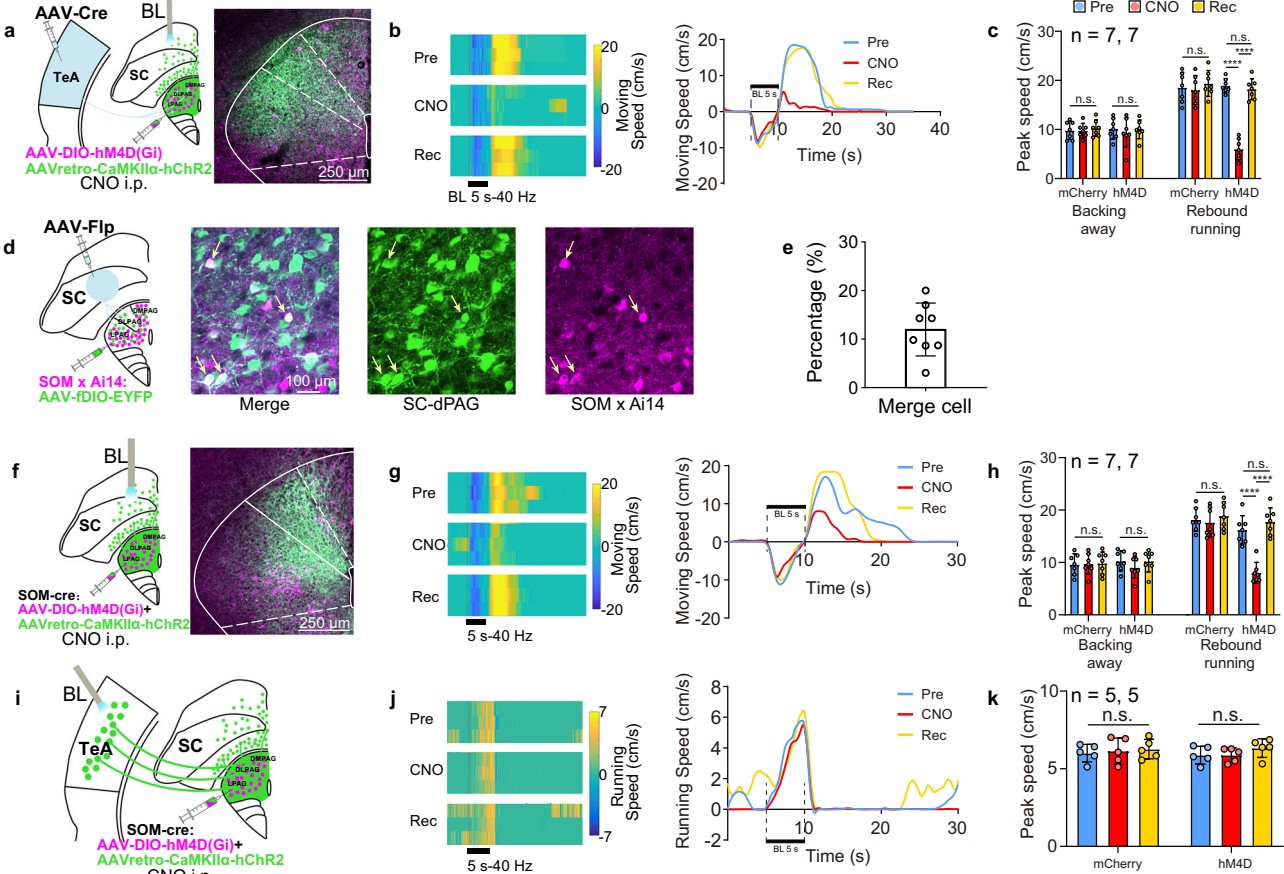

**Fig. 5 | Cellular mechanism underlying rebound running and stopping behaviours. a** Virus injection (left) and expression (right) of hM4D(Gi) (magenta) in $_{TeA}$dPAG cells and ChR2 (green) in SC$_{dPAG}$ neurons. **b** Behavioural responses shown in a heatmap (left) and average movement speed (versus time) (right) evoked by the photoactivation of SC$_{dPAG}$ neurons before CNO administration (Pre), during CNO administration (CNO), and after recovery (Rec) from the effect of CNO. **c** Comparison of the peak speeds of the hM4D(Gi) groups and the control group for rebound running and backing away during the Pre, CNO, and Rec periods. *n*: number of experimental animals. ****$P < 0.0001$, one-way repeated-measures ANOVA with the Bonferroni post hoc correction. **d, e** In SOM × Ai14 mice injected with Flp-fDIO virus (left), $_{SC}$dPAG neurons expressing EYFP (green) and SOM × Ai14

neurons (Ai14 reporter expression is shown in magenta) are shown (**d**). Merged neurons (white) are $_{SC}$dPAG SOM neurons and represent a percentage of SOM × Ai14 neurons among all $_{SC}$dPAG neurons (**e**) (*n* = 8 sections from 3 SOM × Ai14 mice). **f** and **i** Virus injection (left) and expression (right) of hM4D(Gi) (magenta) in dPAG SOM neurons and ChR2 (green) in SC$_{dPAG}$ or TeA$_{dPAG}$ neurons with optogenetic activation in the SC (**f**) and TeA (**i**), respectively. **g** and **j** Same as (**b**), behavioural responses for (**f** and **i**). **h** and **k** Same as (**c**), statistics for (**f** and **i**). ****$P < 0.0001$, one-way repeated-measures ANOVA with the Bonferroni post hoc correction. Data are presented as mean ± SD. Supplementary Data 1 for detailed statistics. Source data are provided as a Source Data file.

chemogenetically activated via clozapine-N-oxide (CNO), was administered intraperitoneally (i.p.) to inhibit neurons expressing this receptor[18]. SC$_{dPAG}$ CaMKIIα neurons were optogenetically stimulated to induce rebound running, while $_{TeA}$dPAG cells were silenced by an i.p. injection of CNO. The induced rebound running was significantly attenuated but not completely abolished, without affecting the backing away behaviours (Fig. 5b, c). The control group (ChR2 virus + hM4D-free virus) exhibited no CNO-induced specific blockade of their rebound running behaviours (Fig. 5c). These findings confirmed that rebound running results from the activation of $_{TeA}$dPAG cells.

Rebound running occurred when the light stimulation was withdrawn (Fig. 4e, g), suggesting that light stimulation might activate inhibitory neurons, which in turn inhibit $_{TeA}$dPAG neurons. Specifically, when light stimulation was halted and the inhibition was removed, $_{TeA}$dPAG neurons became excited through rebound, leading to the observed running behaviours.

Inhibitory neurons, including vasoactive intestinal polypeptide (VIP) and parvalbumin (PV) neurons, are sparsely distributed in the PAG. However, a small population of somatostatin (SOM) neurons has also been identified[5]. We injected AAV-Flp into the SC and AAV-fDIO-EYFP into the dPAG of *SOM-Cre* × Ai14 mice to identify the inhibitory

cell type associated with $_{SC}$dPAG neurons (Fig. 5d). Subsequently, the SOM × Ai14 neurons were visualized in magenta, the $_{SC}$dPAG neurons were labelled in green, and the SOM neurons receiving projections from the SC were visualized in white. Our data indicated that 12% of the $_{SC}$dPAG neurons were SOM cells (Fig. 5e).

We employed *SOM-Cre* transgenic mice expressing hM4Di in SOM neurons and ChR2 in SC$_{dPAG}$ neurons (Fig. 5f) and TeA$_{dPAG}$ CaMKIIα neurons (Fig. 5i) to test whether SOM neurons are activated during blue light stimulation. When SOM neurons were chemically inhibited by an i.p. injection of CNO, the amount of induced rebound running behaviours decreased, whereas the backing away behaviours were unaffected (Fig. 5g, h). The control group (ChR2 virus + hM4D-free virus) exhibited no CNO-induced specific blockade of rebound running behaviours. In contrast, the running behaviours induced by the activation of TeA$_{dPAG}$ CaMKIIα neurons remained unchanged, which was also consistent with the findings from the control group (Fig. 5i–k).

Previous studies have confirmed that $_{TeA}$dPAG CaMKIIα neurons are responsible for running behaviour. Although more than half of $_{TeA}$dPAG neurons are CaMKIIα-positive, a small subset consists of SOM neurons[5]. Therefore, other $_{TeA}$dPAG neuronal subtypes could

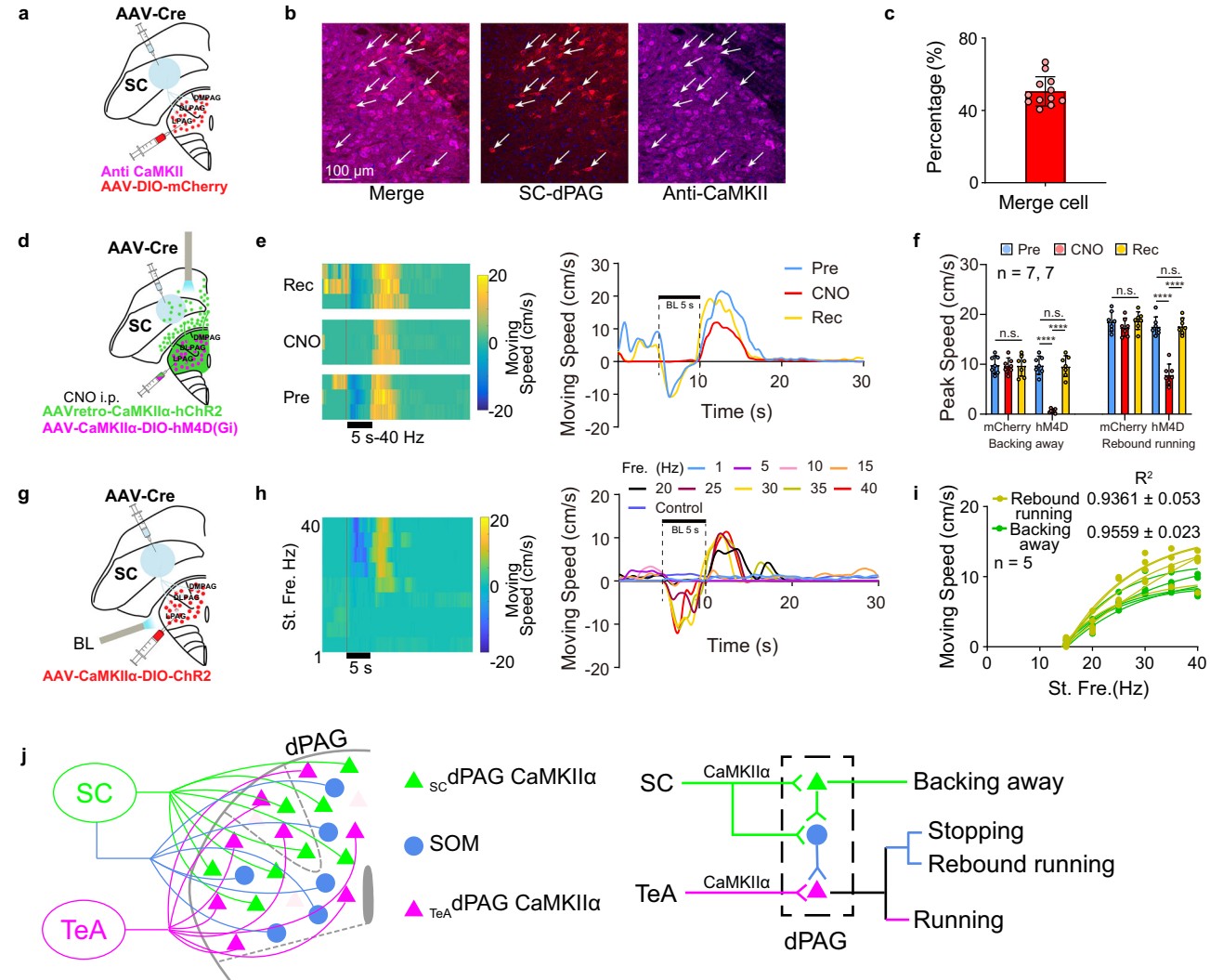

**Fig. 6 | $_{SC}$dPAG neurons responsible for backing away and rebound running behaviours.** **a–c** Experimental protocol (**a**) and expression (**b**) of $_{SC}$dPAG neurons colocalized with signals from the anti-CaMKII antibody in the dPAG. The corresponding percentage of $_{SC}$dPAG neurons is shown (**c**) (anti-CaMKII: $n = 12$ sections from 3 mice). **d** Schematic of the photoactivation of SC$_{dPAG}$ CaMKIIα neurons expressing ChR2 (green) and chemical inhibition of $_{SC}$dPAG CaMKIIα neurons expressing hM4d(Gi) (magenta). **e** Behavioural responses are shown in a heatmap (left), and the average movement speed (versus time) (right) evoked by photo-activation of SC$_{dPAG}$ neurons before CNO administration (Pre), during CNO administration (CNO), and after recovery (Rec) from the effect of CNO are depicted. **f** Comparison of the peak speeds of the hM4D group and the control group for rebound running and backing away during the Pre, CNO, and Rec periods. $n$: number of experimental animals. ****$P < 0.0001$, one-way repeated-measures ANOVA with the Bonferroni post hoc correction. **g** Virus injection to photoactivate $_{SC}$dPAG CaMKIIα neurons. **h** Behavioural responses are shown in a heatmap (left), and average speed (versus time) (right) of mice stimulated at nine photoactivation frequencies are depicted. **i** Movement speed–stimulation frequency function with their fitted curves. **j** Summary of the cellular mechanisms underlying running, backing away, stopping, and rebound running behaviours. Light pink represent merged $_{SC}$dPAG CaMKIIα and $_{TeA}$dPAG CaMKIIα neurons. Data are presented as mean ± SD. Supplementary Data 1 for detailed statistics. Source data are provided as a Source Data file.

participate in the regulation of running behaviour. The results shown in Fig. 5i indicate that $_{TeA}$dPAG SOM neurons are not involved in the circuit controlling running behaviour. Based on these findings, we conclude that rebound running behaviour is triggered by the discharge of $_{TeA}$dPAG CaMKIIα neurons following the cessation of blue light stimulation of SC$_{dPAG}$ CaMKIIα neurons. During blue light stimulation, $_{TeA}$dPAG CaMKIIα neurons were inhibited—an effect resulting from the activation of SC$_{dPAG}$ neurons, which further led to the activation of SOM neurons in the dPAG.

### Cells responsible for encoding backing away behaviours

The activation of SC$_{dPAG}$ CaMKIIα neurons resulted in stopping, backing away, and rebound running behaviours (Fig. 4e). Having clarified the cellular mechanisms underlying stopping and rebound running behaviours (Fig. 5), the following question arises: which

$_{SC}$dPAG neurons are responsible for encoding backing away behaviours?

$_{TeA}$dPAG neurons are mainly CaMKIIα-positive, and a small percentage are SOM cells[5]. Some $_{SC}$dPAG neurons are also SOM neurons, as shown in Fig. 5d. We injected AAV-Cre into the SC and AAV-DIO-mCherry into the dPAG of wild-type C57BL/6J mice to determine whether the remaining $_{SC}$dPAG neurons expressed CaMKIIα (Fig. 6a). A CaMKII antibody was then applied to brain slices from these mice (Fig. 6b), and the colocalization rates of $_{SC}$dPAG neurons labelled with the anti-CaMKII antibody were calculated. The results revealed that the majority of $_{SC}$dPAG neurons were CaMKII-positive, with a colocalization rate of approximately 50% (Fig. 6c).

Thus far, the evidence from the above experiments suggests that $_{SC}$dPAG CaMKIIα neurons may play a key role in inducing backing away behaviours. Next, we expressed ChR2 in SC$_{dPAG}$ CaMKIIα neurons and

hM4Di in $_{SC}$dPAG CaMKIIα neurons by injecting AAVretro-CaMKIIα-hChR2 along with AAV-CaMKIIα-DIO-hM4D(Gi) into the dPAG and AAV-Cre into the SC (Fig. 6d). When $_{SC}$dPAG CaMKIIα neurons were chemically blocked by an i.p. injection of CNO, the optogenetically induced backing away behaviours disappeared, and the extent of the rebound running behaviours was reduced. The control group, which received coinjections of ChR2-expressing virus and hM4D-free virus, showed no CNO-induced specific blockade of backing away or rebound running behaviours (Fig. 6e, f). These findings indicate that $_{SC}$dPAG CaMKIIα neurons predominantly control backing away behaviours; however, these neurons may also activate SOM neurons, which in turn inhibit rebound running behaviours. We investigated this possibility by specifically activating $_{SC}$dPAG CaMKIIα neurons with blue light after injecting AAV-Cre into the SC and AAV-CaMKIIα-DIO-ChR2 into the dPAG (Fig. 6g). Both backing away and rebound running behaviours were evoked (Fig. 6h, i), suggesting that the activation of $_{SC}$dPAG CaMKIIα neurons also activated SOM neurons in the dPAG. Similar to the findings above, these behaviours were not observed in the ChR2-negative control group (Fig. 6h).

## Discussion

### Two behavioural units and a modulatory circuit determine four behavioural states

In this study, we identified three types of neurons in the dPAG that are responsible for four innate behaviours.

The PAG is known to be a crucial centre responsible for certain behaviours, especially defensive behaviours[6,7]. Specifically, the dPAG is typically associated with innate defence behaviours such as running[8,14,15,19,20], whereas the ventral PAG (vPAG) is crucial for learned fear-related defensive behaviours such as freezing[10,21,22]. However, optogenetic and electrical stimulation of dPAG neurons elicits not only running but also freezing behaviours as well as other defensive responses, such as jumping and avoidance[14,23]. Additionally, optogenetic activation of the ventromedial hypothalamus to the dPAG pathway induces immobility without triggering running or avoidance behaviours[20]. These findings suggest that some animal behaviours may not be strictly determined by specific subregions within the PAG.

Although behaviours are generally governed by highly complex neuronal circuits, individual behaviours may be controlled by more specific groups of cells. However, these specific cells remain largely unidentified, leading to inaccuracies and confusion when they are used to identify behavioural indicators.

Similar to how olfactory sensory neurons that detect specific odours are mixed within the olfactory epithelium, the cells governing different behaviours may be intermixed within the PAG[14,23] (Fig. 6j, left panel). Nonetheless, clusters of cells that determine specific behaviours can be isolated according to their innervating nuclei, similar to how olfactory sensory neurons expressing specific odourant receptor genes are concentrated in the glomeruli as olfactory functional units[24–26] (Fig. 6j, left panel). A cell type that determines a particular behaviour can be defined as a "behavioural unit", and the corresponding behaviour can be termed a "unit behaviour".

In this study, $_{TeA}$dPAG CaMKIIα neurons (Figs. 3e, 4k, and 5j), which are identified by their innervation by TeA neurons (Fig. 6j, magenta), represent a behavioural unit whose unit behaviour is running. Similarly, $_{SC}$dPAG CaMKIIα neurons (Fig. 6j, green), which are driven by SC neurons, form another behavioural unit whose unit behaviour is backing away. Activation of either SC neurons or SC$_{dPAG}$ CaMKIIα neurons can excite $_{SC}$dPAG SOM neurons (Fig. 6j, blue). These $_{SC}$dPAG SOM neurons, in turn, inhibit the activity of $_{TeA}$dPAG CaMKIIα neurons, thereby generating two behavioural outcomes: stopping and rebound running (Figs. 4e, g, 5b, g, and 6e, h), depending on the dynamic balance between the two behavioural units. This classification method enables the separation of specific functional neurons (behavioural units) in a mixed population and the identification of different behaviours (unit behaviours) from previously ambiguous definitions.

Our data reveal an active motor arrest mechanism in the dPAG mediated by a specific neural circuit. This stopping state is not merely the absence of movement but results from the selective inhibition of the running behavioural unit ($_{TeA}$dPAG CaMKIIα neurons), driven by the activation of SOM interneurons receiving SC input (Fig. 5). Crucially, during this state, the activity of the backing away behavioural unit ($_{SC}$dPAG CaMKIIα neurons) is not equally suppressed. This indicates that stopping represents an equilibrium state under asymmetric competition, rather than a general quiescence of all neuronal activity.

This distinction raises a central theoretical question: should the SOM neurons executing this crucial inhibitory function be defined as an independent "behavioural unit"? Functionally, their activity is sufficient to trigger the stopping state, resembling a state switch. However, in terms of coding properties, they do not directly represent any movement parameters (e.g., speed, direction) but only encode the inhibitory strength applied to the running behavioural unit. In terms of dependency, their activity is primarily controlled by the backing away behavioural unit, making them more akin to a dedicated inhibitory effector or functional module of that unit. Therefore, we posit that classifying SOM neurons as an "embedded regulatory unit" is more precise than elevating them to a "behavioural unit" on par with the $_{TeA}$dPAG or $_{SC}$dPAG driver units. This classification highlights an efficient division of labor within the action selection circuit: a few core driver units can produce diverse discrete behavioral states through specific local regulatory modules.

In summary, we propose a parsimonious action selection model implemented by the dPAG: two core driver units, running and backing away, compete via an asymmetric inhibitory connection executed by SOM neurons. Modulation of the strength of this inhibitory pathway directly determines whether the network output is backing away, stopping, or running. Rebound running can be understood as a dynamic overshoot following the sudden release of strong inhibition. This model explains how flexible behavioural outputs can be achieved with limited hardware. Future studies recording the natural activity patterns of SOM neurons in freely behaving animals and manipulating them independently of their upstream drivers will ultimately clarify their computational role in the complete process from decision to action execution.

### Behavioural states defined in this study versus instinctive behaviours

The ultimate goal of modern neuroscience is to understand the neural circuits that underlie various brain functions. Instinctive behaviours in animals, such as escape (flight) and freezing, are frequently used as indicators to investigate the neural circuits underlying movement, emotion, instinct, learning, and memory[8,27–30,5], in which the corresponding neural pathway includes at least the perception of cues, processing these cues as sensory signals, making a decision, the transformation of decisions into motor commands, and the initiation of appropriate behaviours[31]. Here, by identifying behavioural units within a mixed neuronal population, we classified confusing behaviours as unit behaviours for dPAG behavioural units. These unit behaviours are the elements of the neural circuits involved in the execution of instincts rather than the instinctive behaviours themselves or other neural circuit elements.

The running behaviours encoded by the activity of $_{TeA}$dPAG neurons[5] (Fig. 3) obviously differ from the escape or flight behaviours evoked by sensory stimuli such as threats[8,14,15,19,20]. They also differ from escape-like gaits, ranging from slower, alternating locomotor patterns to high-speed synchronous locomotor behaviours such as walking, trotting, or galloping[32], as well as from running behaviours controlled by dPAG cells innervated by auditory cortical ($_{AC}$dPAG)[4] and superior colliculus ($_{SC}$dPAG) neurons[12].

The running behaviours (flight or escape) evoked by SC activation[11,12] are often used jointly as indicators for investigating neural circuits involved in movement, emotion, instinct, learning, and memory. However, these may not be typical running behaviours but rather rebound running behaviours. Rebound running differs from regular running in terms of the underlying neuronal basis, the behavioural components involved, and the timing of its occurrence. Regular running (Figs. 3j, 4k, and 5j) occurs during the stimulation of $_{TeA}$dPAG neurons, whereas rebound running occurs after the stimulation of $_{SC}$dPAG neurons following the implementation of stopping or backing away behaviours (Figs. 4e, g, 5b, g, and 6e, h).

Moreover, backing away behaviours, observed upon SC[17] or PAG activation[23], are considered defensive behaviours, given the association of the latter with PAG activation[33,34]. However, backing away behaviours constitute a unit behaviour under the control of the $_{SC}$dPAG CaMKIIα behavioural unit, which is involved in controlling multiple behaviours, including stopping, stopping and rebound running, backing away and rebound running, which depend on the frequency and intensity of the stimulus (Figs. 4e and 6h). Based on our results, we inferred that the activation of $_{SC}$dPAG CaMKIIα neurons excites $_{SC}$dPAG SOM neurons, which in turn inhibit $_{TeA}$dPAG neurons, resulting in the stopping of running behaviours. When SOM-mediated inhibition is withdrawn, $_{TeA}$dPAG neurons become excited through rebound, leading to the generation of running behaviours. Stopping is an active cessation of movement achieved through the activation of specific cells that manifests as an immobile state in mice during low-frequency SC stimulation. Even when high-frequency stimulation induces backing away behaviours, the inhibitory effect mediated by these specific neurons remains persistent and active. The classification method proposed in this study enables the separation of specific functional neurons (behavioural units) in a mixed population and the identification of different behaviours (unit behaviours) from previously ambiguous definitions.

The stopping induced by specific circuit manipulations in this study may behaviourally resemble learned, fear-related defensive behaviour—freezing. Classic freezing can manifest in a variety of ways, such as immobility[33,35], the cessation of locomotion (e.g., escaping, running, and walking)[36,37], immobility accompanied by shaking[38], or halting during an intermediate action state[34]. As shown in Figs. 4e and g, 5b and g, and 6e and h, the stopping or ending of motion that occurs during neural activation can be viewed as a form of active behavioural inhibition, falling within the broad category of freezing. However, it differs fundamentally in triggering context and neural substrates from the amygdala-[39] and ventral midbrain-[31] involving freezing typically induced by fear conditioning. The stopping defined here is more akin to a core, hardwired switch for motor competition.

Furthermore, it is essential to distinguish this active stopping/freezing from inaction due to the natural quiescence of motor units. When immobility occurs at the end of a running or rebound running behaviours, it represents a state of resting rather than a cessation of locomotion, as the firing rates of the corresponding running or rebound running units are insufficient to trigger the associated behaviours (Figs. 1b and 2b). This is mechanistically and conceptually distinct from the stopping caused by active inhibition from SOM neurons, which prevents movement even when the motor unit has the drive to act. The discovery that three types of cells drive four instinctual behaviours reveals a modular cellular mechanism for behavioural selection.

### The TeA to dPAG projection as a driver of running behaviour

In this study, optogenetic stimulation of TeA following injection of AAV-CaMKIIα-ChR2 successfully induced running behaviour (Fig. 3a), with a higher running speed compared to stimulation of TeA$_{dPAG}$ neurons (Fig. 3p). We tend to consider that one of the major pathways through which TeA CaMKIIα neurons drive running is their direct projection to dPAG ($_{TeA}$dPAG neurons). Of course, we cannot rule out the possibility that optogenetic stimulation simultaneously activates other neural circuits within TeA, such as neurons from different sensory cortices that project to TeA and, in turn, innervate TeA$_{dPAG}$ neurons[5], or TeA neurons projecting to other brain regions. For example, studies have shown that activation of SC can also drive running[12]. However, the focus of this study and the data directly support the sufficiency and necessity of the TeA to dPAG pathway. Whether TeA CaMKIIα neurons directly regulate SC or influence running through other indirect pathways requires further investigation using circuit-specific tracing and manipulation tools in future studies. More importantly, this study discovered and demonstrated that running driven by the SC to dPAG pathway differs significantly in behavioural pattern from that driven by the TeA to dPAG pathway (Fig. 6j). Inputs from upstream regions such as TeA or SC converge onto distinct microcircuits and cell types in dPAG, thereby generating difference behaviours. This does not, however, negate the role of SC in driving running, because as illustrated in Fig. 4, differences in the precise location of stimulation within SC, as well as in the frequency and duration of blue-light stimulation, could all contribute to the observed behavioural variations.

### A shared speed code for divergent actions

Our results demonstrate that the firing rate of R-neurons in both TeA and dPAG exhibits a sustained, quantitative covariation with running speed, and a quantitative model was established where this relationship is well fitted by a single-phase association equation. For TeA$_{dPAG}$ neurons and $_{TeA}$dPAG neurons, which likely constitute functional subsets of the TeA and dPAG R-neurons, optogenetic manipulation of these neuronal populations revealed that the relationship between optogenetic stimulation frequency and movement speed could still be accurately described by the same quantitative model. This model also held when optogenetically manipulating SC$_{dPAG}$ neurons. These findings go beyond mere correlation, demonstrating a causal relationship between firing rate and movement speed—that is, the firing rate encodes movement speed. This indicates that speed information is not transmitted vaguely within this pathway but is instead represented and transformed through a specific, high-fidelity input-output function.

The parameters in the function ($v_m$, $\tau$, and $f_0$) are determined by factors such as the strength and quantity of sensory inputs (Fig. 3a–p), synaptic efficacy, and cellular excitability, all of which vary from neuron to neuron. The encoding model (single-phase association equation) applies across different neurons, suggesting that it is not a phenomenon unique to individual neurons but rather a universal encoding principle. The functional form of this model can be normalized to eliminate individual differences among neurons in terms of response gain ($v_m$) and baseline activity level ($f_0$), thereby revealing a core encoding transformation relationship for running speed that is conserved across neuronal populations. This result implies that this neuronal population uses the same language or algorithm to encode movement parameters. The parameter ($\tau$) can be normalized within the same nucleus, whereas different neural nuclei exhibit distinct parameters ($\tau$), indicating that different neural nuclei encode distinct features. Our results revealed that the $\tau$ values differed between TeA and dPAG neurons but did not differ significantly among the neurons in the TeA group (Fig. 3q). The $\tau$ values associated with running speed did not differ significantly from those associated with rebound running, as both behaviours are mediated by $_{TeA}$dPAG neurons (Fig. 4p). Surprisingly, however, the $\tau$ values of TeA neurons were similar to those of SC neurons (Fig. 4p). This indicates that SC, functioning as another upstream source, exhibits the same functional characteristic of coding efficiency as TeA.

A linear correlation between running speed and stimulation frequency or intensity has been previously reported[12,14]. We also performed this analysis (Figs. 1l, q and 2o). After a thorough analysis of the

data trends and careful consideration of their physiological implications, we concluded that the single-phase association equation provided a more biologically plausible representation of the underlying neural–behavioural relationship. Two explanations support this choice. First, although both the linear equation and the single correlation equation capture the firing–running speed trend well, under extreme conditions, when we plot peak firing rate against peak running speed—the single correlation equation demonstrates robust fitting performance, yielding a better fit than the linear equation (Figs. 1q and 2o). Second, the linear fit function $y = k \times (x-x_0)$ yields a slope ($k$) with units of cm/(s·Hz)—representing distance (cm) rather than movement speed (cm/s)—which contradicts the physiological interpretation of R-neuron activity as a controller of speed. In contrast, the single-phase association equation yields a dimensionless time constant ($\tau$), enabling direct comparison with the data from dPAG neurons in Fig. 2.

Within the functionally segregated microcircuit architecture of the dPAG, inputs from TeA and SC activate two distinct neuronal populations, which respectively mediate different behaviours. These two populations are interconnected via unidirectional inhibition mediated by $_{SC}$dPAG SOM interneurons, forming a competitive selection network. Our results demonstrate how the quantitative magnitude (speed, determined by a shared encoding function) and the qualitative behaviour (action selection, i.e., forward running or backing away) of a motor signal are decoupled in the dPAG. The quantitative magnitude information is defined by the parameters of the encoding function, while the qualitative behaviour information is conferred by the input source–specific neurons and microcircuit pathways. Our study indicates that higher brain regions can achieve flexible control by transmitting parameterized signals to downstream hubs, which then parse these signals into specific action programs via their inherent anatomical and functional architecture. This work not only reveals the speed-encoding mechanism of a specific circuit but, more importantly, establishes a research paradigm for dissecting the internal functional architecture of complex brain regions through quantitative behavioural correlations.

## Methods

### Animals

Adult male and female C57BL/6 mice (4–12 weeks old) were purchased from the Laboratory Animal Center of Southern Medical University (Guangzhou, China). The mice were housed in a temperature-controlled vivarium (21–25 °C) with a 12-h light/dark cycle (lights on at 8 am). Food and water were provided ad libitum. Transgenic *SOM-Cre* mice (Jackson Laboratory, JAX Stock #013044) were used in this study. For the cell type labelling experiments, *SOM-Cre* mice were crossed with Ai14 (a Cre-dependent tdTomato reporter line, JAX Stock#007914) of both sexes (male and female). All experiments were conducted in accordance with the Regulations on the Management of Laboratory Animals (China) and were approved by the Animal Ethics Committee of Southern Medical University (L2017207).

### Viral vectors

For optogenetic activation, we used the following adeno-associated viral constructs: AAV2/9-CaMKIIα-hChR2(H134R)-EYFP ($1.99 \times 10^{13}$ particles/ml; OBiO, Shanghai, China), AAV2/retro-CaMKIIα-hChR2(H134R)-EYFP ($1.68 \times 10^{13}$ particles/ml; OBiO), AAV2/9-EFf1α-DIO-hChR2(H134R)-EYFP ($2.31 \times 10^{13}$ particles/ml; OBiO), and AAV2/9-CaMKIIα-DIO-ChR2-mCherry ($6 \times 10^{12}$ particles/ml; BrainVTA, Wuhan, China). For the empty vector control group, we used AAV2/9-CaMKIIα-EYFP ($5.12 \times 10^{12}$ particles/ml; OBiO), AAV2/retro-CaMKIIα-EYFP ($5.23 \times 10^{12}$ particles/ml; OBiO), AAV2/9-EFf1α-DIO-EYFP ($5.31 \times 10^{12}$ particles/ml; OBiO), and AAV2/9-CaMKIIα-DIO-mCherry ($5.72 \times 10^{12}$ particles/ml; BrainVTA).

For chemogenetic inhibition, we used AAV2/9-EFf1α-DIO-hM4D(Gi)-mCherry ($5.18 \times 10^{12}$ particles/ml, BrainVTA) and AAV2/9-CaMKIIα-DIO-hM4D(Gi)-mCherry ($5.5 \times 10^{12}$ particles/ml, BrainVTA). For the empty vector control group, we used AAV2/9-Ef1α-DIO-mCherry ($5.13 \times 10^{12}$ particles/ml, BrainVTA) and AAV2/9-CaMKIIα-DIO- mCherry ($5.72 \times 10^{12}$ particles/ml, BrainVTA).

For retrograde monosynaptic tracing, we used AAV2/retro-hSyn-Cre-EYFP ($5.18 \times 10^{12}$ particles/ml, BrainVTA).

For anterograde tracing, we used AAV2/1-hSyn-Cre-WPRE-hGH ($1.16 \times 10^{13}$ particles/ml, BrainVTA), AAV2/1-hSyn-Flpo-WPRE ($1.04 \times 10^{13}$ particles/ml, BrainVTA), AAV2/9-nEf1α-fDIO-EYFP ($5.27 \times 10^{12}$ particles/ml, BrainVTA), and AAV2/9-Ef1α-DIO-mCherry ($5.13 \times 10^{12}$ particles/ml, BrainVTA). Specifically, AAV2/1 was used for anterograde monosynaptic tracing[40–42].

### Animal preparation

All animal preparation steps were performed in a relatively sterile room maintained at 23–26 °C.

**Viral injection.** Four- to five-week-old mice were anaesthetized with sodium pentobarbital (60–70 mg/kg, i.p., Sigma, USA) and positioned in a stereotaxic frame (RWD, Shenzhen, China). Body temperature was maintained at 37 °C using a heating pad (RWD; Shenzhen, China). Viral preparations were injected through a small opening in the skull ($\leq 0.5 \, mm^2$) using a glass micropipette (Drummond Scientific, USA) with a tip diameter of 20–30 μm, driven by a microinjector (KD Scientific, USA) at a rate of 30 nl/min. The stereotaxic coordinates, as reported by the Allen Institute (2011), were as follows: TeA, −2.9 mm anteroposterior (AP), 4.7 mm mediolateral (ML), 0.5 mm dorsoventral (DV); dPAG, −4.0 mm AP, 0.2 mm ML, 2.2 mm DV; and SC, −4.0 mm AP, 0.6 mm ML, 1.5 mm DV. The head was rotated laterally at a 90° angle to administer the virus precisely to the TeA. The glass micropipette was advanced from the superficial to the deep layers of the TeA, reaching a depth of 500 μm. The virus was allowed to spread completely by elevating the injection site of the viral needle to a depth of approximately 100 μm and allowing it to remain in place for 5–10 min. The wound was closed with medical sutures, and erythromycin ointment was applied to prevent infection. After waking, the mice were placed back in the animal facility for recovery. Four weeks after the injection, the mice were used in the relevant experiments.

**Head-fixed preparation.** The mice were anaesthetized with sodium pentobarbital (60–70 mg/kg, i.p.) and fixed on a stereotaxic instrument (RWD Life Science, China) when the pedal withdrawal reflex disappeared. Lidocaine was injected locally under the scalp, and the skull was exposed and cleaned of exudate. The recording sites were exposed and labelled as follows: TeA (1.82–3.88 mm posterior to the bregma, 4.7 mm lateral to the midline) and dPAG (3.8–4.24 mm posterior to the bregma, 0.2 mm lateral to the midline) (The Allen Institute 2011). A custom-made tripod-type head nail was mounted on top of the skull with dental cement. The surgical procedures were based on established methods[43,44].

**Cannula Implantation.** During head fixation surgery, two fibre-optic cannulas (fibre core: 0.2 mm, 0.37 NA; Newdoon, China) were implanted bilaterally prior to performing optogenetic activation of the SC in open-field behavioural experiments. Cannula implantation was not performed for the flat turntable experiment; instead, optical fibre jumpers attached to a fibre core pin were connected directly to a manual micromanipulator. Dental cement was used to fix the cannulas vertically within the SC for the open-field tests. The stereotaxic coordinates for the SC were −4.0 mm posterior to the bregma, 0.6 mm lateral to the midline, and 1.3 mm below the brain surface.

After the operation, the mice were returned to their cages and treated with antibiotic ointment to prevent wound infection.

**Running training.** During the recovery period, the mice were secured by screwing the tripod pins into a metal pole and trained to run or rest on a flat, custom-made turntable (20 cm in diameter). Training sessions were repeated for 1–2 h per day over 3 days until the mice adapted to the experimental recording environment and could run freely. Mice exhibiting uncoordinated limb movements on the turntable, as well as those who ran continuously and vigorously, were excluded from the study (5%), as such behaviours would be detrimental to the quality of the electrophysiological recordings.

**Electrophysiological recording window.** On the final day of adaptive training, a skull drill (RWD Life Science, China) was used to expose the recording sites of the head-fixed mice while preserving the dura. The exposed area was covered with Vaseline.

## Sound, light, and air puff stimulation

The goal of this study was to analyse the relationship between the firing rate and running speed during autonomous running in mice. We incorporated various sensory stimuli during the recording sessions to identify running neurons—those that exclusively encode running speed without responding to sensory stimuli—among the recorded neurons. Specifically, we used sound[43], light[45] (white light flashes), and air puffs[46] to encourage running in the mice, depending on their state. A TDT3 system (Tucker-Davis Technologies, Alachua, FL, USA) was employed to produce the three types of stimuli. All the experimental control programs were developed using RPvdsEx software and subsequently loaded into the multifunction processor (RX6) of the TDT3 system.

**Acoustic stimuli.** Acoustic stimuli were generated using the TDT3 system. All the sound programs were compiled using RPvdsEx software and then loaded into RX6 to generate sound waveforms, with sine curves characterized by a zero phase and a 5 ms rise/fall time. The sound intensities were controlled by a programmable attenuator (PA5). The synthesized signals from RX6 were amplified using an electrostatic speaker driver (ED1) and delivered through a loudspeaker (ES1, frequency range 2–110 kHz) placed 10 cm in front of the animal. Prior to the experiments, the loudspeaker was calibrated using an amplifier (Brüel & Kjær 2610) connected to 1/4- and 1/8-inch microphones (Brüel & Kjær 4135). A pure tone of 80 dB SPL (sound pressure level; $0\,dB = 20\,\mu PA$) and a duration of 1 s were used for the experiments.

**Air puff stimulation.** Air puffs were delivered using a microvalve (Kamoer-KVP04, China), and the tube was positioned 10 cm from the animal. The air puff duration was 1 s, with a flow rate of 0.8 L/min (calibrated with Darhor-LZB-2, USA) to achieve the desired intensity.

**Light stimulation.** Light stimulation was implemented through LED lamp beads controlled by the RP2.1 real-time processor of the TDT3 system. The white light flash duration was 1 s, with an intensity of 2 lux (calibrated with KOMAX, Germany). The LED lamp was placed 10 cm in front of the animal.

The parameters for all three stimuli were controlled using BrainWare software.

## Optogenetic activation

For the optogenetic activation experiments performed in head-fixed mice on the flat turntable, the optical fibre (fibre core: 0.2 mm, 0.37 NA; Newdoon, China) was connected to a blue LED source (473 nm; THINKERTECH, China) controlled by a manual micromanipulator (Narishige, Japan). The optical fibre was embedded horizontally in the TeA (embedding depth: 500 μm), vertically in the dPAG (embedding depth: 2000 μm), and vertically in the SC (embedding depth: 1300 μm) for recording. For TeA stimulation, the same approach used for

electrophysiological recording was employed. The micromanipulator (Siskiyou, USA) was replaced with a manual three-axis micromanipulator (MM-3; Narishige, Japan), enabling the gripper (RWD; Shenzhen, China) held by the fibre holder to be advanced horizontally into the TeA. The location of optogenetic activation corresponded to the location of virus injection. The axis of the light pathway was aligned with the central axis of the optical fibre, and the illumination angle was determined by the NA value of the fibre[43].

For optogenetic experiments involving electrophysiological recording, the position of the optical fibre was adjusted to avoid interference with the patch pipette. The frequency of blue laser pulses (10 ms pulse duration, 5 s stimulation duration) was set to 1, 5, 10, 15, 20, 25, 30, 35, and 40 Hz, with each stimulus frequency presented in a pseudorandom manner and repeated 10 times with intervals ranging from 30 to 90 s. The fibre tip power, as measured with an optical power metre (Thorlabs, USA), was 10–15 mW. The recording duration was 15 min.

## Chemogenetic inhibition

For the chemogenetic inhibition experiment, mice expressing hM4D(Gi) were intraperitoneally injected with clozapine-N-oxide (CNO; 1 mg/kg) following optogenetic activation and behavioural testing. 30 min after drug administration, 15 min of mouse speed data were collected. Recovery tests were performed 24 h after drug administration.

A stock solution of CNO (BrainVTA, China) was prepared at a concentration of 10 mg/ml by adding sterile dimethyl sulfoxide (DMSO) (HY-Y0320; Med-Chem Express, USA) directly to a tube containing the powdered compound. The tube was securely capped and subjected to vigorous vortex mixing for 30–60 s to achieve a preliminary dispersion of the solid in the solvent. The mixture was subsequently sonicated for 5–10 min in an ultrasonic bath to ensure complete dissolution, as this process utilizes sound energy to markedly increase solute dissolution. Following sonication, the solution was visually inspected for clarity. The vortexing and sonication steps were repeated iteratively until no particulate matter was visible and the solution was entirely clear and transparent. Upon complete dissolution, the stock solution was promptly aliquoted into single-use vials to prevent degradation from multiple freeze–thaw cycles and stored at −20 °C until use. For intraperitoneal injection into the animals, the stock solution was diluted to the desired working concentration (0.33 mg/ml) using sterile physiological saline immediately prior to use.

## In vivo cell-attached recording in awake running mice

In vivo cell-attached recording was performed based on established methods[43,44]. Briefly, successfully trained mice were positioned on the flat turntable within a custom-made, soundproof room maintained at 23–26 °C. After the Vaseline and dura were removed, the pia mater was penetrated. A patch pipette (Sutter, USA) with a tip diameter of 1.5 μm and a resistance of 5–8 MΩ filled with artificial cerebrospinal fluid (ACSF; concentrations in mM: 126 NaCl, 2.5 KCl, 1.25 $NaH_2PO_4$, 26 $NaHCO_3$, 1 $MgCl_2 \cdot 6H_2O$, 2 $CaCl_2$, 2 sodium pyruvate, 10 glucose; pH 7.35–7.45; osmolarity 290–310 mOsm/kg) was advanced with a micromanipulator (Siskiyou, USA). The pipette was positioned horizontally for TeA recordings (depth: 100–900 μm) or vertically for dPAG recordings (depth: 2000–2500 μm). By adjusting the configuration of the three-axis series manipulator (MX1641 Series Manipulator, Siskiyou, USA), we aligned the direction of advancement to a horizontal orientation (parallel to the coronal axis of the brain). During the electrophysiological recordings, the pipette was advanced vertically from the superficial layers to the deep layers of the TeA with the micromanipulator. While a positive pressure of 0.5–1 psi was applied, the pipette was advanced in 1 μm increments until the impedance changed to 5–10 MΩ. At this point, the positive pressure was switched

to a negative pressure (−0.3 psi) to establish a loose seal (30–100 MΩ), indicating successful cell attachment. Following the establishment of a cell-attached seal, voltage-clamp mode (Vcmd = 0 mV) was employed to monitor and record both sensory-evoked and spontaneous action potentials in the neuron. For the visualization of the cell morphology and confirmation of the recording location, 1% biocytin (wt/vol) was added to the ACSF. The neuron was patched with a loose seal (0.2–1 GΩ) during cell-attached recording. The signals (spikes) were recorded using a MultiClamp 700B amplifier (Axon, USA) in current-clamp mode, and positive current pulses (3–10 nA, 1 Hz) were applied for 20–30 min to iontophoretically deliver biocytin into the neuron. The signals were filtered at 300–3000 Hz and sampled at 20 kHz.

Regarding the acquisition of neuronal coordinates, during the recordings, we primarily localized the positions of recorded R-neurons using biocytin labelling; however, the labelling efficiency was sub-optimal. Consequently, for a subset of neurons, we adopted an alternative approach: after recording from an R-neuron, we filled glass micropipettes with larger tip diameters than standard recording electrodes with an eosin dye solution. This dye was then pressure-ejected (-2 psi) at the recorded site. Subsequent cryosectioning and histological examination allowed us to verify whether the recording site was within the TeA nucleus and determine its rostrocaudal position along the TeA axis.

### Running recording

**Running on the flat turntable.** Each mouse was placed on a flat turntable connected to a rotary encoder (US Digital, USA) to record the running speed. The behaviour of the mice on the turntable was mon-itored and recorded using an external, high-definition infrared camera (LRCP10620, 20 FPS, China). The spike signals, speed signals, and video footage were recorded simultaneously through a customized program. Spike signals were captured using BrainWare 32 (Tucker-Davis Technologies, USA) and Clampex 10.2 software (Axon, USA). Speed signals and videos were recorded using USD Device Explorer software (US Digital, USA) and Windows Cafittinmera software (Microsoft, USA), respectively. All the data were digitized and stored on a computer for offline analysis.

**Open field running model.** The behavioural responses of freely moving mice were recorded in a specifically designed circular cage (30 cm in diameter and 40 cm in height) during the SC optogenetic activation experiments. Prior to the behavioural test, the animals were allowed to acclimate to the cage for 5 min. A video camera (LRCP10620, 20 FPS, China) was positioned above the cage to monitor the animal's activity. The speed of the motion induced by optogenetic stimuli was calculated from the video images.

### Immunohistochemistry

After each experiment, the animals were sacrificed, and the brains were removed, sectioned and imaged under a confocal microscope to confirm viral expression and confirm the specificity of the optogenetic stimulation. Specifically, the mice were deeply anaesthetized with pentobarbital sodium (60–70 mg/kg, i.p.) and then perfused with 50 ml of 4% paraformaldehyde (PFA) in 0.1 M phosphate buffer (PB, pH 7.4). The brains were then prepared for immunohistochemical staining by removing them and storing them in PFA fixative for 24 h at 4 °C. After fixation, the samples were washed with running water and sub-jected to gradient dehydration using 20% and 30% sucrose solutions. Coronal sections were cut using a cryostat (Leica CM1860, Germany) at a thickness of 100 μm for the detection of biocytin staining and 40 μm for immunofluorescence staining. The brain sections were washed with PBS (15 min, three times) and incubated with 0.3% (v/v) Triton X-100 for 1 h. After blocking with 5% normal goat serum (NGS, Boster) for 1 h at room temperature, primary antibodies, including an anti-CaMKII monoclonal rabbit antibody (1:200; ab52476, Abcam, USA) and

streptavidin-Cy3 (1:200; 438315; Thermo Fisher Scientific, USA), were applied in 5% NGS and incubated overnight at 4 °C. For anti-CaMKII immunofluorescence staining, the sections were incubated with Alexa Fluor 647-conjugated goat anti-rabbit IgG (1:500; A21244; Invitrogen™, USA) in PBS for 2 h at room temperature, followed by washes with PBS (10 min, three times). Finally, all the sections were mounted onto microscope slides, covered with coverslips, and treated with an anti-fade reagent containing DAPI (S2110; Solarbio, Beijing).

### Imaging and quantification

The fluorescence signals were visualized using a laser scanning con-focal microscope (A1R, Nikon) and analysed with ImageJ 1.4 (NIH) or NIS Elements software. The virus-infected or immunolabelled cells were quantified by manually counting the fluorescent cells using ImageJ (NIH Image 1.4). Neurons were counted from 3 to 4 slices per mouse, with the sections centred on the coordinates of the dPAG. The imaging settings were kept consistent across all groups of sections. The percentage of merged neurons was calculated as the number of merged neurons divided by the total number of virus-infected neurons or the sum of virus-infected and antibody-labelled neurons.

### Data processing and analysis

BrainWare software of the TDT3 system and Clampfit 10.2 software (Axon, USA) were used to export and analyse the recorded electro-physiological data. Video data were initially analysed and exported using custom software, Animal.exe (Shanghai Jiliang Company). All experimental data, including the number of action potentials, time, action potential waveforms, and mouse movement speed and time, were extracted and analysed offline using MATLAB 2016b (Math-Works). Data processing and further analysis were performed using Excel 2016, OriginPro 2017 (OriginLab Corporation), and MATLAB 2016b. Custom MATLAB scripts were employed for data analysis. The average time-dependent speed profile and neuronal firing rate curve were smoothed using the smooth function in MATLAB.

During the test, the rotation speed of the turntable was measured by a rotary encoder and recorded in real[13]. We defined the onset of running as the point at which the running speed of the mouse excee-ded 0.5 cm/s, indicating the beginning of apparent running. Neural data were analysed using custom MATLAB (MathWorks) scripts. A mean spike density function was constructed for each neuron by applying a Gaussian kernel (σ = 10 ms) to each spike.

**Identification of R-neurons.** The identification of R-neurons first required confirming their lack of responsiveness to sensory stimuli. To isolate purely sensory response trials, we analyzed only those trials in which sensory stimulation did not successfully elicit a running response. A trial was defined as a non-running trial if the running speed remained consistently below 0.5 cm/s within the post-stimulation sensory–running time window[5]. For each neuron and each sensory stimulation (sound, light, air puff), a post-stimulus time histogram (PSTH; bin size = 10 ms) aligned to stimulus onset was constructed. A neuron was considered responsive to a given stimulus if its mean firing rate during the post-stimulus response period ([0, 1] s) showed a sig-nificant increase compared to that during a 1-s pre-stimulus baseline period (Wilcoxon signed-rank test, $P < 0.05$). Neurons exhibiting a significant response to at least one sensory stimulus were classified as sensory-related neurons. Neurons that did not meet this criterion proceeded to evaluation for running-related activity.

Running-related activity was assessed by evaluating all sponta-neous running events, defined as running that began in the absence of any sensory stimulation. A trial was classified as a spontaneous run if no sensory stimulation was delivered within the pre-run sensory stimulation-running time window[5]. Spontaneous running onsets were aligned to time zero. Peri-event time histograms (PETHs) were con-structed for each neuron using a 10-ms bin width centred on the

running onset. Neurons were defined as exhibiting significant running-related if their mean firing rate during the designated pre-running period was significantly higher than during a stationary baseline period, as assessed by the Wilcoxon signed-rank test ($P < 0.05$). The baseline firing rate was defined as the average firing rate when the mouse was not running.

**The definition of the firing-running time window.** The start of the running-related activity window was identified as the first-time bin where the firing rate in the PETH exceeded the threshold (Baseline + $2 \times$ SD). The temporal window of running-related activity was then defined as the period from this identified start time until running onset (time zero). This window represents the period of a significant increase in the firing rate preceding observable running.

**Heatmap analysis of R-neurons.** To investigate the relationship between spontaneous running behaviour and the firing rate of R-neurons, we generated heatmaps of PETHs aligned to the onset of spontaneous running events. Each row represents the average activity of a single neuron across all trials. For each R-neuron, we detected the onset of spontaneous running events using a speed-threshold method over the entire recording period. The running-related neural activity was analyzed as follows: For each spontaneous running event, we extracted spike data from 10 s before to 10 s after the running onset (time zero). Only running events that met the following criteria were included: no other running events occurred within the 10 s preceding the onset, and the running speed reached at least 0.5 cm/s during the event.

For each neuron $i$, the mean firing rate within time bin $j$ was calculated as:

$$Frequency_{ij} = SpikeCount_{ij} \div N_i \times \Delta t$$

where $SpikeCount_{ij}$ denotes the total spike count of neuron $i$ in time bin $j$ across all spontaneous running trials, $N_i$ is the number of spontaneous running trials for neuron $i$, and $\Delta t$ is the bin width (0.1 s).

Heatmaps were generated using a hot colormap, with color gradients ranging from black (no firing) to white (highest firing rate). Each row represents the average, trial-averaged firing pattern of an individual R-neuron along the time axis. Each heatmap was internally normalized using min-max scaling to preserve relative firing patterns and facilitate visual comparison across different neurons.

The response patterns of R-neurons to three types of sensory stimuli were visualized in the same manner. Only trials without running were included in this analysis. Non-running trials were defined based on the stimulus-running window: trials in which no running event occurred within the post-stimulus running window were considered stimulus-only trials. The time bin was set to 0.1 s to balance temporal resolution and statistical stability. The stimulus parameters were as follows: stimulus onset at 1.0 s, stimulus offset at 2.0 s, stimulus duration of 1.0 s, and an analysis time window from 0 to 5 s (aligned to stimulus onset at 1.0 s).

**Cross-correlation function (CCF) analysis.** For each neuron, the CCF between its firing rate and running speed was computed over a time lag range of [−10, 10] s with 100 ms bins. The maximum correlation coefficient ($r_{max}$) and its corresponding time lag ($\tau$) were recorded.

Significance Testing: The statistical significance of the observed correlation was assessed using a permutation test. A null distribution was generated by performing 1000 random circular shifts of the running speed trajectory relative to the firing rate. The observed $|r_{max}|$ was considered statistically significant if it exceeded the 95th percentile of this null distribution ($P < 0.05$). R-neurons (running-related neurons) exhibited a significant positive correlation between their firing rate and

movement speed ($P < 0.05$). The maximum correlation coefficient ($r_{max}$) and its optimal lag were extracted for each neuron.

**Functional fitting.** To characterize the relationship between neuronal firing rates in mice and running speed, we begin by aligned the speed curve to the firing rate curve by shifting it forward according to the neuron-specific optimal time lag, with the firing curve held fixed. A firing rate versus speed plot was then generated and fitted using an exponential function:

$$V = V_m \times \left(1 - e^{-\frac{(f - f0)}{\tau}}\right)$$

Normalization: All the fitted firing–running speed curves of the TeA and dPAG neurons were classified into two groups for standardization. First, the minimum firing rate and minimum running speed were subtracted from each neuron's respective values. Next, the average of the maximum firing rates and the average of the maximum running speeds were calculated across neurons for each group. Finally, each neuron's firing rate and running speed were standardized using the average maximum firing rate and average maximum movement speed as benchmarks. The data for TeA R-neurons presented in this study differ from those reported in a previous paper by H.L. and J.C.[5]

**Statistical analysis and reproducibility.** For all representative micrographs shown in the figures, each image is representative of at least three independent experiments with similar results. The sample size ($n$) for all quantitative data is defined in the figure legends and represents biological replicates. This refers either to the number of individual animals or to the number of cells sampled from distinct animals, as explicitly stated in each legend. Statistical details for each comparison, including the specific tests used and exact $p$-values, are provided in the figure legends or Supplementary Data 1 files. Statistical analysis was performed using SPSS (SPSS 21, IBM). The dataset was begun by tested for normality using the Shapiro–Wilk test and Levene test. OriginPro 2017 (OriginLab Corporation) or GraphPad Prism 7 (GraphPad Software) was used for statistical analysis and graphing. Statistical significance was evaluated using Student's $t$-test and one-way repeated-measures ANOVA, unless stated otherwise. The Mann–Whitney $U$ test was applied if the data were not normally distributed. Significance levels are indicated as follows: *$P < 0.05$, **$P < 0.01$, ***$P < 0.001$, and ****$P < 0.0001$. The results are presented as the means $\pm$ SDs, unless specified otherwise.

### Reporting summary
Further information on research design is available in the Nature Portfolio Reporting Summary linked to this article.

## Data availability
All relevant data are including within the paper and its Supplementary Information files. Source data are provided with this paper.

## Code availability
We used MATLAB 2016b to write custom code for subsequent data processing. The MATLAB code for processing the neural, behavioural, and electrophysiology data is available at https://github.com/LiHe0606/Animal-speed-and-Neuron-firing-2-XiaoLab.

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

## Acknowledgements

This work was supported by grants from the National Natural Science Foundation of China (Grant Nos. 32371044 and 32070994), the Regional Joint Fund Key Projects of Guangdong Province (Grant No. 2022B1515120088) and the Key Research and Development Plan of Guangzhou Science and Technology Plan Project (Grant No. 2023B03J1337). We acknowledge American Journal Experts LLC for their professional language editing and proofreading services.

## Author contributions

Z.X. and W.Z. conceived and supervised the study. J.C., H.L., W.Z., N.L., L.Y., Y.H., P.Y. and M.F. performed all of the experiments. J.C., H.L., W.Z., P.Y., L.Y., Y.Z. and J.T. contributed to the data collection. J.C., H.L., W.Z., Z.X., P.Y., X.Q., J.T., Y.Z., X.J. and J.L. analysed the data. Z.X., W.Z., H.L. and J.C. wrote the manuscript. J.C. and H.L. contributed equally to this work.

## Competing interests

The authors declare no competing interests.
