## [Transparent Peer Review file · Nature Communications]

A shared speed encoding model for running and backing away behaviours in segregated neural circuits

Corresponding Author: Dr Wen Zhong

Version 0:

Reviewer comments:

Reviewer #1

(Remarks to the Author)

Review on "Cells and their coding for running, backing away, rebound 1 running, stopping" by Chen et al. (NCOMMS-24-78436-T).

General comments on the two companion papers:

This paper has been submitted as a companion paper (NCOMMS-24-78434-T) by Zhongju Xiao and Wen Zhong. The two papers investigate neuronal circuits driving different defensive behaviors (running away, stopping, backing away and rebound running) in the dorsal periaqueductal grey (dPAG), temporal association areas (TeA) and superior colliculus (SC). The authors use a combination of viral tracing, projection selective neuronal manipulation and single-unit extracellular recordings in head-fixed behaving mice to define the neuronal circuits driving defensive behaviors.

The two papers cover two distinct aspects of the circuitry with only little overlap. The first paper (Chen et al., NCOMMS-24-78436-T) describes the role of two pathways, TeA-to-dPAG and SC-to-dPACG in driving running and backing away, involving two distinct cell populations in the dPAG. The second paper (Li et al., NCOMMS-24-78434-T) investigates in more details the TeA neuronal circuits involved in integrating sensory cues from different modality into motor action (running) showing that L5 IT cells in TeA integrate sensory information from sensory cortical areas and thalamic nuclei and drive L5PT cells projecting to dPAG to initiate running.

Both papers present exciting and new findings that are largely supported by an impressive set of experimental data using state of the art methodologies. I find particularly attractive the combination of cell-type- and projection- specific manipulations combined with single-cell electrophysiology that provides a straightforward functional characterization of the neuronal circuits involved in simple innate behaviors.

On a less positive note, I found both papers not easy to fully grasp due to lack of clarity. This comes both from a purely formal aspect with the writing of the paper – some sentences are just difficult to understand (starting with the title!) - and from some lack of explanation about the experimental procedures and data analysis.

Both papers would also require a better and more throughout analysis of the neuronal spiking data using more standard approaches to disentangle what part of the activity is related to sensory cues and movements (not only running speed). It would be good to homogenize as much as possible the analyses across the two papers, in particular for the neuronal activity (see my comments below).

In addition, I have a series of more specific concerns and questions for each paper that I detail below for this paper (Chen et al. NCOMMS-24-78436-T):

Major comments:

- It is really not clear in this paper whether we are dealing with spontaneous or sensory-evoked running behavior. Line 98

seems to indicate that sensory stimuli are presented but there is no indication of sensory stimuli in the figures (1a-b and 2a-b) and no further mention of sensory stimuli in the rest of the paper. If running is largely evoked by sensory stimuli, that should be taken into account in the analyses: the sensory-evoked responses might become a confounding factor for the correlation between neuronal activity and running, and a more careful analysis of sensory-evoked and running related activity need to be carried out – in line with what is done in the second paper.

- Overall, I am not convinced by the analysis of the neuronal activity and the correlation of firing rate with running speed. My understanding is that the correlation is computed after realigning the neuronal activity and the running speed curves for each event (running bout), which will likely artificially maximize the correlation and does not seem right to me. If the increase in neuronal activity does not reliably correlate with running at short and fixed latencies across trials, one cannot conclude that this neuron encodes running speed. The time-lag between the neuronal activity and the running speed could be measured in a more unbiased and simpler way by computing the cross-correlogram and defining the time-lag at the peak of the cross-correlogram. Then this specific time-lag should be used to fit the one-phase function only for the neurons, if any, with strong significant correlation. My impression when looking at the data presented is that the activity of TeA neurons at single cell level might be a good predictor of running onset time (initiation) but is a poor predictor running speed (due to important trial-to-trial variability). A better approach would be to use a more standard Generalized Linear Model (GLM) approach (fitting neuronal activity using running speed as regressor) or decoding approach (predict running speed from neuronal activity). Altogether I believe that a simpler and more standard analysis using event triggered PSTHs aligned to different event times (running onset, stimulus onset) and ROC analysis would be just as good. It would also help to clarify how running-responsive cells are defined. In any case, the authors should show single-neuron examples and grand-average PSTHs to be show overall time course of the neuronal activity relative to different events (onset of running, sensory cues...).

- I have also a general conceptual problem with the time delays between neuronal activity and movement onset time reported in this (and the companion) paper. In a similar study, Evans et al. (Nature 2018 <https://doi.org/10.1038/s41586-018-0244-6> see figure 3) report running latencies in the range of 0.2 s when stimulating dPAG neurons and 0.5 s when stimulating SC. In the current paper, stimulation of TeA or dPAG neurons also seems to evoke running with short latencies (based on the example sessions displayed in the figures). However, the time differences between running and firing rate reported (figure 2c) are really long (about 0.8 s for dPAG neurons and 2 s for TeA neurons on average) which seems incompatible with the claim of a direct monosynaptic connection from TeA to dPAG driving running.

- Anatomy of the recorded neurons (Figure 1f-i): this part would also need clarification. There is no explanation about how the coordinates of each recorded neuron are determined to obtain the distribution map in panel 1g. Figure 1g should indicate orientation of the map (DV/AP) and a scale bar. I find really odd that R-Neurons are so well confined within TeA area considering that movements affect neuronal activity everywhere in the brain (see Steinmetz et al, Nature 2019 and Stringer et al., Science 2019 just as examples). Looking at the image in panel 1f, it seems clear that the anatomy of the recorded neurons could only partially be recovered. That is certainly enough to identify the neurons as pyramidal cells and their respective location, but I do not think the reconstructions in panel 1i are meaningful.

- Provide more details about the extracellular recording procedure. It seems that the recordings in TeA are performed by inserting the pipette orthogonal relative to the cortex surface, which suppose a close to horizontal angle. How is it achieved practically speaking (size of the craniotomy, recording chamber, reference electrode, how is the brain protected during the recording)? Also describe the protocol for juxtacellular labelling.

- line 158-163: this sentence does not make sense to me.

- line 302-303, figure 4g: Backing and stopping should be dissociated. Interestingly, the duration of 'backing' appears to be constant, regardless of the duration of the opto-stimulation, but the duration of 'stopping' lasts as long as SC is stimulated!

- line 360-363: This is a reasonable assumption but it is not shown in the current study. That could go to the Discussion.

- Provide more details about the extracellular recording procedure. It seems that the recordings in TeA are performed by inserting the pipette orthogonal relative to the cortex surface, which suppose a close to horizontal angle. How is it achieved practically speaking?

Minor comments:

- First paragraph of Results (lines 89-94) should move to the Introduction.

- Line 210: it should be specified that the AAV-Cre virus used here is an anterograde transsynaptic serotype that will drive expression of Cre in the post-synaptic neurons receiving inputs from TeA.

- Line 281, figure 4e, f: I do not see any sign of 'backing away' when the stimulus is withdrawn?!

- Clearly indicate on each figure panel when is the stimulus presented (ex: figure 3 a-e2 right panel, 4f, 4g2 and h2 ...)

Sylvain Crochet

(Please do leave my name apparent to the authors)

Reviewer #2

(Remarks to the Author)

The manuscript by Chen et al. evaluates the role of neuronal projections from the temporal association cortex (TeA) and superior colliculus (SC) to the dorsal periaqueductal gray (dPAG) in regulating running behavior. Using in vivo loose-patch recordings, optogenetic, and chemogenetic approaches, the authors find that the neuronal firing rate of the TeA-dPAG pathway correlates with running speed, while the SC-dPAG pathway regulates backward movements in mice. They further suggest that rebound running and stopping behaviors on the rotating disk involve a complex circuit motif between these regions. While the manuscript presents intriguing data, several critical issues limit its impact, and it is not suitable for publication in Nature Communications in its current form.

Major Concerns:

- 1. Lack of Ethological Relevance:** While the study elegantly demonstrates the involvement of the TeA-dPAG and SC-dPAG pathways in modulating running behavior, the connection between the observed behaviors and their ethological relevance is unclear. The authors frequently describe rebound running and backward movement as "defensive responses." However, these behaviors are not typically observed in naturalistic studies of rodent defense. It is possible that these behaviors result from the head-fixed setup on the rotating disk, rather than representing natural defensive responses.
- 2. Misinterpretation of Stopping Behavior:** The authors have misinterpreted stopping as freezing, a well-characterized defensive state that differs fundamentally from the absence of locomotion. Freezing is associated with distinct postural adjustments, reduced respiratory rate, and increased muscle tension (Roelofs, 2017). Clarifying this distinction is essential for accurately interpreting the data.
- 3. Lack of Proper Control Groups:** The manuscript does not include optogenetic or chemogenetic control data, making it difficult to determine whether the observed effects result specifically from neuronal modulation. Including appropriate control groups would substantially strengthen the conclusions by ruling out potential confounds and providing stronger evidence for the causal role of these pathways in the observed behaviors.
- 4. Normalization of Viral Expression Data:** Direct comparisons of the effects of different viral preparations (e.g., in Figure 3) may be confounded by differences in expression efficiency and tropism. To address this, data on the number of neurons with viral expression should be provided. Ideally, behavioral data should be normalized to account for variations in viral expression between different stimulation approaches.
- 5. Clarity of Statistical Analysis:** The manuscript lacks clarity on the statistical analyses used, and some data comparisons, such as those in Figure 3f, are confusing. Detailed explanations of statistical methods and justification for the comparisons made are necessary to ensure transparency and reproducibility.
- 6. Reciprocal Connectivity of Pathways:** Since AAV-Cre transsynaptic viruses can express retrogradely, the authors should investigate whether the TeA-dPAG and SC-dPAG pathways have reciprocal connectivity. This information would provide critical insights into the circuit dynamics underlying the observed behaviors.

Reviewer #3

(Remarks to the Author)

Version 1:

Reviewer comments:

Reviewer #1

(Remarks to the Author)

This revised version of the manuscript present convincing evidence that SC neurons induce backing away movement via the monosynaptic activation of dPAG CaMKII positive neurons followed by rebound running via the activation of SOM neurons in the dPAG likely producing rebound excitation of another, non-overlapping population of dPAG CaMKII positive neurons receiving inputs from TeA. Despite an evident effort of clarification, this revised version of the manuscript still suffers from important flaws in the way the data are presented and analyzed in my opinion. In particular, I am still not convinced by the way the correlation between the firing-rate and the running speed is analyzed. I still strongly believe that aligning each running event to the previous peak in firing rate present major risk of false positive. In fact, figure 1 shows quite a few neurons for which the Vm value fitted to the data is very low, close to 0, indicating almost no correlation between the firing rate and the running speed. In addition, the overall logic throughout the result section is often difficult to follow with some experiment lacking clear rational and some results that are misinterpreted.

Major concerns:

- One of the major issues is the way the correlation between the neuronal activity and the running speed is analyzed. First, it

is still unclear how exactly R-neurons are defined: which statistical analysis is used to determine whether the neurons respond to the sensory stimuli or not? is the correlation between firing rate and running speed computed throughout the entire recording or just for running events? I would find surprising to see any significant correlation between the firing rate and the running speed with a ~2 s delay between the two. Again, it would seem fairer to me to compute the cross-correlogram between the firing rate and running speed to obtain the peak of the correlation and time lag between the two.

- Also, in the companion paper the authors clearly state that the activity of some neurons is not correlated with running. I suppose that the authors only report running related neurons but no indication is provided as to how many neurons were recorded in total, in how many mice, and how many of them where running-correlated. It is also not clear whether the neurons recorded in TeA are newly recorded neurons for this study or simply from the companion paper study. This should be explicitly mentioned.

- All along the paper, many experiments seem redundant or only bring minor additional information. A better selection of what is shown in the main figures could be done to make the flow of the paper easier and clearer.

- Some statements are inaccurate or lack logical link or state the obvious (see below)

Other less critical issues:

- Introduction, line 73-74: in fact, there is no direct evidence that TEAdPAG neurons do not respond to sensory stimuli in the companion paper.

- Introduction, line 78: this paper does not “investigate the mechanism by which [TEAdPAG neurons] encode running speed”.

- Results, lines 150-1452: “To initiate a movement, the animal appears to calculate its maximum running speed (v_m) on the basis of τ , which corresponds to synaptic efficiency.” The link between the τ of the fitted function that links firing rate and running speed, and efficiency of a synapse (which one?) is totally speculative.

- Results, line 160: “this relationship is somewhat counterintuitive” Why ?!

- Results, line 179-180: Statistics missing

- Results, line 184-189: trial-by-trial variability in neuronal response is highly expected in the most neuronal circuits. I suppose the authors mean to suggest that the encoding of the running speed most likely occurs at the level of a population of neurons. Which something well described and largely admitted in both sensory and motor systems.

- Results, line 225-227: I do not understand this sentence. Also, the fact that TeAdPAG neurons are CaMKII positive is largely expected since most cortical projection neurons are excitatory (glutamatergic) and CaMKII is a widely used marker for cortical excitatory neurons.

- Results, line 274-277: isn't it obvious?

- Results, line 342-343: On the picture in Figure 4b there seems to be much more double positive (yellow) cells than reported ?

- Results, line 380-381: I stand on my first impression that activation of SC neurons evoke backing-away with fixe duration followed by an absence of movements that last until the end of the blue-light pulse.

- Results, line 442-444: ???

- Results, line 443-444: This is a valid assumption but there is no direct evidence showing that SOM neurons innervate TeAdPAG CaMKII+ neurons.

- Results, paragraph 453-470: lack of logical link and rational for the experiments. The only difference between the experimental protocols of Fig 6b1-2 and Fig 4d-f is that in the former, only SC CaMKII + neurons projecting to dPAG express ChR2 whereas in the latter all projecting neurons are labelled. First, it is not immediately clear why the authors want to do this experiment to explain the mechanism of backing away evoked by the stimulation of SCdPAG CaMKII+ neurons. Second, the interpretation does not seem entirely correct to me. It cannot be concluded from the latter experiment that SC neurons make inhibitory connection to SOM SC neurons. The only conclusion that can be made is that stimulating all SC neurons projecting to dPAG results in less excitation of dPAG neurons responsible for backing away. That could as well result from local mechanisms in SC. In my opinion all this part is unnecessary to the story and just bring confusion. The key experiment is the one presented in Fig 6f-g.

Minor:

- Figure 6b3: impossible to see the difference between Backing away and Rebound running.

- Results, line 492-493: “the above experiments suggests that CaMKII α in the SCdPAG may play a key role in inducing

backing away behaviours.”. No! that ‘SCdPAG CaMKII positive neurons’ may play a key role. CaMKII is just a marker. There is no indication that it may play a role in this study.

Sylvain Crochet

Reviewer #2

(Remarks to the Author)

The authors put forth a great effort to address previous concerns and comments. They have significantly improved the figures and statistics, and the text of the manuscript is also greatly improved. It is appreciated that they removed reference to defensive behavior. While the relevance of some of the behavior to more naturalistic settings is a minor concern, the experiments are elegant and the data will be of interest to the neuroscience community.

Reviewer #3

(Remarks to the Author)

Version 2:

Reviewer comments:

Reviewer #1

(Remarks to the Author)

In this revised version of the manuscript, the authors have taken some of my comments into account. The second part of the study, which breaks down the neural circuits involved in running and backing away (Figures 3 to 6), is now clear and convincing. However, I still have doubts about the first part (juxtacellular recordings in the TEA and dPAG). I maintain that the method for measuring the correlation between neural activity and running speed—by realigning each running event with the preceding neural event (increase in the firing rate) with varying delays—is flawed and carries a high risk of spurious correlations that are not properly accounted for (e.g., by using random permutations between neural activity and running speed) (see Harris 2020: <https://doi.org/10.1101/2020.11.29.402719>). This is particularly problematic given the timescale: lines 196-198 state, "The time difference between a firing event and its corresponding running event, calculated from their onset, ranged from 0.3 to 5.1 s." So much can happen in the brain during such long periods of time. This should also be considered in light of the escape latency of less than 2 s observed in the related study. I expect that the correlation analysis as conducted in this study would yield similar results regardless of where neuronal activity was recorded from. I remain convinced that the most appropriate method for assessing correlation is to define the time lag from the peak of the cross-correlogram. This defined and fixed time lag for each neuron should then be used to realign neuronal activity and running velocity.

Furthermore, the way the results are reported still contains some inaccuracies and imprecisions:

- The objective of the study remains unclear: the title and introduction are centered on "running speed coding" - which, in my opinion, is the weak point of the article - while two-thirds of the article actually deals with the TeA/SC/dPAG circuits that control the different behaviors observed.

- In several places, the reported mean values are incorrect. This appears to be due to the authors calculating the mean values by fitting the distributions to a normal distribution, even when the observed distribution is clearly not normal. This is particularly evident in Figure 1f, where the distribution is clearly bimodal and certainly cannot be accurately modeled by a single normal distribution. This results in an overestimation of the mean value of r_{max} , whereas a significant proportion of neurons exhibit a much weaker correlation. The authors should report the true mean or median values. It is also difficult to reconcile these high correlation values with the authors' emphasis on variability between trials (events) (see lines 168–171).

- Figure 1r and 2l plots the fittings of neuronal activity for 42 trials from 29 cells (1r) and 30 trials from 22 cells (2l), which suggest that the fitted parameters (τ , v_m and f_0) were estimated from only 1 trial (running event) for many cells! This further calls into question the reliability of these analyses.

- When looking at the PETHs presented in figure 1d2 and 2d2, it is not easy to be convinced that there is indeed a clear causal relationship between the neuronal activity and running onset.

Minors:

- Lines 352-356: this is again just a valid hypothesis but this is not supported by any data. There is no indication that TeA CaMKIIa neurons that do not project to dPAG excite TeAdPAG neurons. They may as well project to other structures, such as SC?

- Figure 3 i: we would like to see more than just one example trial for the characterization of dPAG ChR2+ and dPAG ChR2- neurons.

- Discussion. An alternative interpretation would be the existence of 2 only behavioural units in the dPAG ('running' and

'backing away') with inhibition of one onto the other through SOM interneurons resulting in 3 possible behaviors depending on the balance between the two: backing away, stopping and running.

- Rebuttal lines 285-290: these are just mean values not statistics supporting the comparisons.

- The legends of figure 1 and 2 describe panels c2 and d2 as 'population neuronal responses' but I believe that it is in fact several trials from 1 neuron (not a population of neurons).

Sylvain Crochet

Dear Reviewer,

The line numbers referenced in our point-by-point responses correspond to the
"Revised manuscript without track changes-chen.pdf" file.

**Reviewer #1 (Remarks to the Author):**

**Review on "Cells and their coding for running, backing away, rebound 1 running,
stopping" by Chen et al. (NCOMMS-24-78436-T).**

**General comments on the two companion papers:**

**This paper has been submitted as a companion paper (NCOMMS-24-78434-T) by
Zhongju Xiao and Wen Zhong. The two papers investigate neuronal circuits
driving different defensive behaviors (running away, stopping, backing away and
rebound running) in the dorsal periaqueductal grey (dPAG), temporal association
areas (TeA) and superior colliculus (SC). The authors use a combination of viral
tracing, projection selective neuronal manipulation and single-unit extracellular
recordings in head-fixed behaving mice to define the neuronal circuits driving
defensive behaviors.**

**The two papers cover two distinct aspects of the circuitry with only little overlap.
The first paper (Chen et al., NCOMMS-24-78436-T) describes the role of two
pathways, TeA-to-dPAG and SC-to-dPACG in driving running and backing away,
involving two distinct cell populations in the dPAG. The second paper (Li et al.,
NCOMMS-24-78434-T) investigates in more details the TeA neuronal circuits
involved in integrating sensory cues from different modality into motor action
(running) showing that L5 IT cells in TeA integrate sensory information from
sensory cortical areas and thalamic nuclei and drive L5PT cells projecting to
dPAG to initiate running.**

**Both papers present exciting and new findings that are largely supported by an**
**impressive set of experimental data using state of the art methodologies. I find**
**particularly attractive the combination of cell-type- and projection- specific**
**manipulations combined with single-cell electrophysiology that provides a**
**straightforward functional characterization of the neuronal circuits involved in**
**simple innate behaviors.**

**On a less positive note, I found both papers not easy to fully grasp due to lack of**
**clarity. This comes both from a purely formal aspect with the writing of the paper**
**– some sentences are just difficult to understand (starting with the title!) - and**
**from some lack of explanation about the experimental procedures and data**
**analysis.**

**Both papers would also require a better and more throughout analysis of the**
**neuronal spiking data using more standard approaches to disentangle what part**
**of the activity is related to sensory cues and movements (not only running speed).**
**It would be good to homogenize as much as possible the analyses across the two**
**papers, in particular for the neuronal activity (see my comments below).**

**In addition, I have a series of more specific concerns and questions for each paper**
**that I detail below for this paper (Chen et al. NCOMMS-24-78436-T):**

We sincerely appreciate the reviewers' time and insightful comments, which have
significantly improved our manuscript. In response to the key concerns raised:

Language clarity: The text has been professionally edited by American Journal Experts
LLC;

Analysis of neuronal spiking data: We have supplemented additional data analyses
based on your raised concerns and provide detailed point-by-point responses below

regarding the specific revisions;

Other suggestions: All points have been addressed in the revised version.

Point-by-point responses to the reviewers' comments are provided below:

**Major comments:**

- **It is really not clear in this paper whether we are dealing with spontaneous or**
**sensory-evoked running behavior. Line 98 seems to indicate that sensory stimuli**
**are presented but there is no indication of sensory stimuli in the figures (1a-b and**
**2a-b) and no further mention of sensory stimuli in the rest of the paper. If running**
**is largely evoked by sensory stimuli, that should be taken into account in the**
**analyses: the sensory-evoked responses might become a confounding factor for the**
**correlation between neuronal activity and running, and a more careful analysis of**
**sensory-evoked and running related activity need to be carried out – in line with**
**what is done in the second paper.**

We sincerely appreciate your constructive comments. As clarified in the revised
manuscript, this study extends the foundational work of Li et al. (H.L. J.C.) by further
investigating “running-related neurons” (R-neurons). We employed multiple sensory
stimuli to identify only those R-neurons that showed no sensory responses but exhibited
firing rates correlated with running speed (**Fig. 1d; 2d**).

Fig. 1d

Fig. 2d

**Figure legends**

**1d**, Peristimulus time histogram (PSTH) of the neuronal response to three sensory modalities:
auditory (sound), visual (light), and somatosensory (air puff). The blue line indicates the stimulation
period, with a duration of 1 s.

**2d**, PSTH of neuronal responses to three sensory modalities: auditory (sound), visual (light), and
somatosensory (air puff). The blue line indicates the stimulation period, with a duration of 1 s.

We now explicitly state this connection in **the Results section (Line 90-95)**: “*To*
*explore how the firing of R-neurons encodes running speed, we performed in vivo loose-*
*patch recordings of TeA neurons in awake running mice* *11 exposed to random sound,*
*light, and air puff stimuli (Fig. 1a). Sensory stimuli (Fig. 1b, middle) were applied to*
*identify running-related neurons, defined as those whose firing rates are correlated with*
*running speed (Fig. 1b, bottom, Fig. 1c) (Supplemental Video 1), rather than with the*
*parameters of the sensory stimuli (H.L., J.C.) (Fig. 1d).*” to avoid ambiguity in the text.

- **Overall, I am not convinced by the analysis of the neuronal activity and the**
**correlation of firing rate with running speed.**

**My understanding is that the correlation is computed after realigning the neuronal**
**activity and the running speed curves for each event (running bout), which will**
**likely artificially maximize the correlation and does not seem right to me. If the**
**increase in neuronal activity does not reliably correlate with running at short and**
**fixed latencies across trials, one cannot conclude that this neuron encodes running**
**speed.**

**The time-lag between the neuronal activity and the running speed could be**
**measured in a more unbiased and simpler way by computing the cross-**
**correlogram and defining the time-lag at the peak of the cross-correlogram.**

**Then this specific time-lag should be used to fit the one-phase function only for the**
**neurons, if any, with strong significant correlation.**

**My impression when looking at the data presented is that the activity of TeA**
**neurons at single cell level might be a good predictor of running onset time**
**(initiation) but is a poor predictor running speed (due to important trial-to-trial**

**variability).**

Thank you for your valuable comments, which have helped us recognize narrative
deficiencies in the submitted manuscript.

We have deleted the r value in the original Fig. 1b. Furthermore, this panel now shows
the firing rate of R-neurons alongside the animal's running speed, clearly illustrating a
covariance between the two traces. To assess the potential correlation between the R-
neuron firing rate and animal running speed, we show the firing rate versus running
speed curve (**Fig. 1c**) and performed Pearson correlation analysis, the results of which
revealed a strong positive correlation ($r = 0.844$, $P < 0.0001$), confirming a statistically
significant relationship between the two parameters. This correlation analysis was
performed by comparing the firing rate derived from all recorded firing events of an
individual neuron, and its results suggest that these neuronal firing rates predict animal
running speed rather than running onset (initiation). However, the relationship between
the firing rate and running speed differed across individual events (**Fig. 1c**).

To further characterize the relationship between the firing rate and running speed, we
performed quantitative fitting analysis. Neuronal firing events (dashed) preceded
running events (solid), but the time difference between the firing event and the
corresponding running event differed across trials (**Fig. 1b, values**). Therefore, we
hypothesized that the firing rate determines running speed and treated each running
bout as a discrete event where the firing rate and running speed represent two
interdependent parameters of the event. To clarify this point, we have added an insert
(**Fig. 1e, insert derived from 1b, the dashed red box**) as an example to show time
shifting, aligning the firing rate and running speed curves according to their peak (**Fig.**
**1e**) or onset (**Fig. 1f, the insert**) latency differences, and established their temporal
relationship across multiple running events. Subsequent plotting and fitting revealed
how the R-neuron firing patterns dictate running speed (**Fig. 1e, f**), which yielded
significantly improved fitting with respect to onset alignment (**Fig. 1g**).

Fig. 1

**1b**, An example of the firing rate (dashed line) of a TeA running-related neuron (R-neuron), which
 is correlated with running speed (solid line) but not with the timing of sensory stimulation (blue
 line). The values in the bottom panel represent the time difference between a firing event and the
 associated running event, measured at the peak point (triangle, grey values) or onset point (circle,
 black values), respectively. The red dashed box indicates the example corresponding to the fitted
 curve in **e and f**. **c**, Firing rate plotted against the original running speed. r : correlation coefficient
 ($n = 6$ running events, $P = 1 \times 10^{-10}$, Pearson's r). **d**, Peristimulus time histogram (PSTH) of the
 neuronal response to three sensory modalities: auditory (sound), visual (light), and somatosensory
 (air puff). The blue line indicates the stimulation period, with a duration of 1 s. **e and f**, Firing rates
 plotted against the optimized running speed. The inset on the bottom right of each graph shows the
 aligned plots of a representative firing rate curve (grey arrow, corresponding to the red dashed box
 in **b**) and running curve after time shifting on the basis of the difference between their peak or onset
 158 times. Red line: fit with a single-phase association equation (nonlinear regression, $P < 10^{-4}$ for both
 **e and f**). **g**, Statistical comparison chart of goodness-of-fit for the single-phase association equation
 modelled on the basis of peak and onset points. $n = 6$ running events.

**-A better approach would be to use a more standard Generalized Linear Model**
 **(GLM) approach (fitting neuronal activity using running speed as regressor) or**
 **decoding approach (predict running speed from neuronal activity).**

Thank you for your suggestion. Indeed, we plotted the relationships among the animal
 running speed, distance, acceleration, and neuronal firing rates and analysed them with
 generalized linear and nonlinear model methods (data not shown). However, the reason

we ultimately chose to use a nonlinear model to analyse the relationship between
neuronal firing rates and animal running speed was that we found the strongest
correlation between the animal's instantaneous running speed and neuronal firing rates,
and the nonlinear fitting model performed better than the linear fitting model did
(demonstrating a higher R^2). Owing to space constraints in the description of our initial
analysis, we did not include peak-to-peak plots of the overall data (neuronal firing rates
vs. animal running speed). Therefore, to further demonstrate the rationality of our
analytical approach as per your suggestion, we fitted the following data with both linear
and one-phase associated equations: (1) firing rates and running speed after aligning
their peaks (**Fig. 1m**), and (2) firing rate peaks against corresponding running speed
peak times after aligning their onset times (**Fig. 1n and 2g**). The results showed that
the single-phase associated equation provided a better fit than did the linear fit (higher
R^2), and onset alignment yielded a better fit than did peak alignment. **(The relevant**
**content is described in the Results section: line 131-141:** *“Aligning the data on the*
*basis of the peak firing rate and speed, as shown in Fig. 1e (the points corresponding*
*to the example grey line shown in the inset), we plotted the peak firing rate against the*
*peak running speed and found that these data fit to a single-phase association equation*
*(Fig. 1m) better than to a linear equation ($R^2 = 0.2318$, $p = 0.0012$; data not shown).*
*After alignment on the basis of the onset of the firing and movement events, as shown*
*in Fig. 1f (the points corresponding to the example grey line shown in the inset), we*
*also plotted the peak firing rates against the corresponding running speeds at the peak*
*firing times and fitted them (Fig. 1n). This fit was better than that in Fig. 1m and that*
*for the linear equation ($R^2 = 0.2503$, $p = 0.0007$; data not shown), suggesting that*
*firing and running events initiate from their onset.”; **line 168-171:** *“The corresponding*
*plots, similar to those in Fig. 1n, indicating good fitting of the data to the single-phase*
*association equation (Fig. 2g) that was better than that to the linear equation ($R^2 =$*
*0.4356, $p = 4.345 \times 10^{-5}$; data not shown).”).**

Fig. 1m and n

Fig. 2g

**1m**, Fitting curves for peak firing–peak running events. $R^2 = 0.3018$, $P = 4.27 \times 10^{-11}$ (nonlinear
 regression). **1n**, Fitting curves for peak firing and running speed corresponding to the peak firing
 rate obtained after onset-aligned shifting, $R^2 = 0.3401$, $P = 9.916 \times 10^{-12}$ (nonlinear regression).

**2g**, Fitting curves for the peak firing rate and running speed corresponding to peak firing obtained
 after onset-aligned shifting; the points correspond to the example grey line are shown in the inset of
 e. $R^2 = 0.455$, $P = 9.371 \times 10^{-13}$ (nonlinear regression).

**-Altogether I believe that a simpler and more standard analysis using event**
 **triggered PSTHs aligned to different event times (running onset, stimulus onset)**
 **and ROC analysis would be just as good. It would also help to clarify how running-**
 **responsive cells are defined.**

**In any case, the authors should show single-neuron examples and grand-average**
 **PSTHs to be show overall time course of the neuronal activity relative to different**
 **events (onset of running, sensory cues...).**

We fully agree with your valuable suggestion, which provides a more comprehensive
 perspective on the dataset. As outlined in our neuron selection criteria, we prioritized
 analysing neurons that exhibited speed-correlated firing during self-initiated running
 episodes. Secondary consideration was given to neurons that showed poststimulus
 speed modulation (after confirming that they exhibited no direct sensory responses). As
 suggested by the reviewers, we performed PSTH analysis (**Fig. 1d and 2d**) for the
 recorded R-neurons in response to sensory stimuli (sound, light, air puffs) to confirm
 their lack of sensory responsiveness.

However, we could not use sensory stimuli (e.g., auditory cues) to align neuronal firing
for PSTH construction. Instead, we analysed all recorded firing-movement events by
plotting the peak firing rate against the corresponding maximum movement speed for
each event, thereby assessing the overall correlation across the dataset. We plotted the
peak firing rate against the maximum running speed for each event to generate firing
rate-running speed profiles (**Fig. 1m**). Alternatively, after aligning on the basis of the
onset of the firing rate and speed, we plotted the peak firing rate-corresponding running
speed values on the peak firing rate times graph (**Figs. 1n and 2g**).

With respect to ROC curve analysis, we believe that this method can determine only
whether neuronal firing is running related. In contrast, our Pearson correlation analysis
(**Fig. 1c**) provides a more direct assessment of whether the firing rate is correlated with
running speed, revealing a positive correlation between the two parameters. This
approach allows more precise characterization of the functional coupling of firing rate
and running speed.

**- I have also a general conceptual problem with the time delays between neuronal**
**activity and movement onset time reported in this (and the companion) paper. In**
**a similar study, Evans et al. (Nature 2018 [https://doi.org/10.1038/s41586-018-0244-](https://doi.org/10.1038/s41586-018-0244-6)**
**[6](https://doi.org/10.1038/s41586-018-0244-6) see figure 3) report running latencies in the range of 0.2 s when stimulating**
**dPAG neurons and 0.5 s when stimulating SC.**

**In the current paper, stimulation of TeA or dPAG neurons also seems to evoke**
**running with short latencies (based on the example sessions displayed in the**
**figures). However, the time differences between running and firing rate reported**
**(figure 2c) are really long (about 0.8 s for dPAG neurons and 2 s for TeA neurons**
**on average) which seems incompatible with the claim of a direct monosynaptic**
**connection from TeA to dPAG driving running.**

We appreciate your insightful comment, which has helped us identify limitations in our

data processing methodology. We have added two subplots (**j** and **k**) to **Fig. 3** to address
this issue.

Fig. 3 j,k

**3j**, Running latency versus St. Fre. (every 10 trials, error bars: SDs). **k**, Running latency versus
running speed.

The time latency recorded in our experiments can be affected by a variety of factors,
such as the synaptic structure, the expression level of the optogenetic viral construct,
and the intensity and frequency of the light stimulus. As shown in the newly added **Fig.**
**3j**, under the same stimulus intensity, the stimulation-running latency progressively
decreases with increasing optogenetic stimulation frequency. Differences in these
factors ultimately manifest as differences in behavioural speed. We therefore correlated
stimulation frequency with behavioural speed and constructed a speed–latency plot (**Fig.**
**3k**). The results suggest that when the behavioural speed increases to approximately
100 cm/s (as demonstrated in **Evans DA et al¹. Fig. 3**), comparable response latencies
can be achieved.

Evans DA et al Fig. 3

**a**, Speed traces with increasing light intensity (10Hz pulse, black lines) from one mouse (mSC left,
 dPAG right). **b**, Psychometric curve (mSC: 278 trials, N=4, slope=4.0, 95% CI [2.75, 5.25]; dPAG:
 590 trials, N=7, slope =26.3, 95% CI [22.1, 30.4]). Lines are logistic fits (pooled across all animals
 and binned light intensities), inset shows fit slope (error bars are SD). **c**, Chronometric curve (mSC:
 149 trials, slope=-0.21, 95% CI [-0.27, -0.15]; dPAG: 328 trials, slope=-0.07, 95% CI [-0.11, -0.03]).
 Lines are linear fits, inset as (b). **d**, Correlation between light intensity and escape speed (mSC: 149
 trials, $P=0.04$; dPAG: 328 trials, $P=1.5 \times 10^{-5}$; Pearson's r). Error bars are SEM unless otherwise
 indicated, mSC data is shown in purple and dPAG in blue.

From the beginning, we noticed a long delay in information processing across a single
 synapse from TeA to dPAG neurons. Therefore, we first demonstrated a monosynaptic
 function is indeed present from TeA to dPAG neurons with in vivo recording (**Fig. 3i**)
 and that the running triggered by activation of TeA_{dPAG} neurons can be blocked by
 inhibiting TeA_{dPAG} cells (**Li et al, Fig. 7k-n**). Then, we showed that the delay we

observed was accurate from a number of perspectives: 1. The time difference between
 R-neuron discharge and the running of the animal recorded by the two researchers under
 different experimental contents was essentially the same (2.117 ± 0.242 s by **Li et al.**,
 **Fig. 4j** versus 2.031 ± 0.937 s by our group, **Fig. 1o and 2h**); 2. The time difference
 between R-neuron discharge and the running of the animal caused by the activation of
 ChR2^- -TeA_{dPAG} (2.062 ± 0.2133 s) and ChR2^+ -TeA_{dPAG} R-neurons (2.076 ± 0.1480 s;
 **Li et al.**, **Fig. 7j**) tended to be consistent; 3. ChR2^- dPAG neurons, which were activated
 across one synapse by optogenetic stimulation of ChR2^+ TeA_{dPAG} fibres, exhibited a
 longer latency of approximately 1 second (0.956 ± 0.137 s) (**Fig. 3i, middle, bottom**),
 corresponding to the latency across one synapse from TeA to dPAG neurons. This
 latency was also close to the delay observed between the TeA and dPAG (1.163 s)
 shown in Fig. 2h, which was similar to the timing differences between TeA_{dPAG} neurons
 and TeA_{dPAG} fibres, TeA_{dPAG} CaMKII α neurons, or TeA_{dPAG} neurons, reflecting the
 delay across one synapse from TeA to dPAG neurons (approximately 1.147 s) (**Fig. 3j**).

**Fig. 1o and 2h**

**Fig. 3i**

**1o**, Count distribution of time differences on the basis of onset time ($n = 42$ trials from 29 cells of
 15 animals). The red dotted line represents the Gaussian fit curve ($R^2 = 0.87$, $P = 2.3 \times 10^{-4}$).

**2h**, Distribution of time differences for dPAG neurons (black) and TeA neurons (gray), originally
 shown in **Fig. 1o**. Gaussian fitting curve for dPAG; $R^2 = 0.983$, $P = 1.5 \times 10^{-4}$.

**3i**, In vivo loose-patch recordings from a TeA CaMKII α neuron expressing ChR2 (ChR2^+ , top) (**a**)
 and a dPAG neuron not expressing ChR2 (ChR2^-) (_{TeA fibre}dPAG) (middle) (error bars: SDs) (**b**). The

light stimulation pulse duration was 10 ms (scale bar, 100 ms, 50 pA). Bottom, the latency between
 the onset of photoactivation and the generation of neuronal action potentials.

Li et al Fig. 4j

Li et al Fig. 7 j,k

Li et al Fig. 7 l,m,n

 **4j**, Summary of the firing–running time difference for each type of neuron. unSR-SS: 2.49 ± 0.11 s,
 unSR-LS: 2.45 ± 0.13 s, unSR-AP: 2.56 ± 0.21 s, SR-SS: 2.45 ± 0.18 s, SR-LS: 2.38 ± 0.15 s, SR-
 AP: 2.47 ± 0.19 s, R: 2.12 ± 0.24 s. $K-W = 47.16$, $P < 10^{-8}$, Kruskal–Wallis one–way ANOVA with
 Bonferroni post hoc correction.

**7j**, Time difference between firing and running events. $F_{2,72} = 45.16$, $P < 10^{-8}$, one-way ANOVA
 with Bonferroni post hoc correction. ChR2⁺-_{AC}TeA: 2.483 ± 0.1661 , ChR2⁻-TeA_{dPAG}: $2.075 \pm$
 0.2017 , ChR2⁺-TeA_{dPAG}: 2.059 ± 0.1474 , $n = 12, 7, 8$ cells. **k**, Virus injection and chemogenetic
 (CNO) and optogenetic (LED) interventions. _{AC}TeA and _{S1}TeA neurons expressing ChR2-EYFP;
 TeA_{dPAG} neurons expressing hM4D(Gi). **l, m**, Example LED light (20 Hz, 5 s, 9 trials)-induced
 speed raster (**l**) and running speed trace (**m**) (10 trials) before (Pre), during (CNO), and recovery
 (Rec) from the effect of CNO i.p. injections **n**, Induced peak speed for the Pre, CNO, and Rec
 periods. Pre: 6.149 ± 1.179 cm/s, CNO: 0.9806 ± 0.6357 cm/s, Rec: 6.27 ± 1.366 cm/s, $F_{2,12} =$
 142.444 , $P < 10^{-6}$, one-way repeated-measures ANOVA with Bonferroni post hoc correction. $n = 7$
 animals.

- **Anatomy of the recorded neurons (Figure 1f-i): this part would also need**
**clarification. There is no explanation about how the coordinates of each recorded**
**neuron are determined to obtain the distribution map in panel 1g. Figure 1g**
**should indicate orientation of the map (DV/AP) and a scale bar.**

We sincerely appreciate your suggestion. In response, we now incorporate both the map
orientation indicators (DV/AP) and scale bars in the revised version (**Fig. 1i**).

Fig. 1i

**1i**, Recording sites (**i**) of 29 recorded cells from 15 animals (error bars: SDs). D, dorsal; A, anterior.

Regarding the acquisition of neuronal coordinates, during the recordings, we primarily
localized the positions of recorded R-neurons using biocytin labelling; however, the
labelling efficiency was suboptimal. Consequently, for a subset of neurons, we adopted
an alternative approach: after recording from an R-neuron, we filled glass micropipettes
with larger tip diameters than standard recording electrodes with eosin dye solution.
This dye was then pressure-ejected (~2 psi) at the recorded site. Subsequent
cryosectioning and histological examination allowed us to verify whether the recording
site was within the TeA nucleus and determine its rostrocaudal position along the TeA
axis.

-**I find really odd that R-Neurons are so well confined within TeA area considering**
**that movements affect neuronal activity everywhere in the brain (see Steinmetz et**
**al, Nature 2019 and Stringer et al., Science 2019 just as examples).**

Indeed, not all recorded neurons were strictly confined to the TeA nucleus. While our
surgical approach unavoidably sampled adjacent areas (the auditory cortex, AC, and the
Ectorhinal cortex, Ect), the limited surgical window ensured that the majority of
recordings were from the TeA region, and thus we obtained relatively few running-
related neuronal recordings from non-TeA regions. As this study focused specifically
on TeA R-neurons, we excluded recordings from neurons from other regions from the
current analysis. The figure legend was intended to illustrate the distribution of R-
neurons recorded primarily within TeA and the systematic exclusion of non-R-neurons
(either inside or outside TeA). We acknowledge that the original presentation was
imprecise and have corrected it in the revised version (**the Results section: line 124-**
**128:** *“The recorded neurons were labelled with biocytin for localization (Fig. 1h). We*
*primarily recorded from the rostral region of the TeA (Fig. 1i), in which R-neurons*
*were predominantly located in the fifth layer of the cortex (Fig. 1j). These neurons*
*exhibited a characteristic morphology, with long dendrites extending vertically into the*
*superficial layers (Fig. 1k), consistent with the findings of our other reports (H.L., J.C.)”*).

**-Looking at the image in panel 1f, it seems clear that the anatomy of the recorded**
**neurons could only partially be recovered. That is certainly enough to identify the**
**neurons as pyramidal cells and their respective location, but I do not think the**
**reconstructions in panel 1i are meaningful.**

We acknowledge your valid critique regarding the reconstructions in panel 1i. While
these preliminary reconstructions may not yet fully meet the expected analytical
standards, we have included them to transparently present our complete methodological
workflow.

We included these reconstructions to enable direct comparison with the morphological
characteristics of L5 TeA_{dPAG} neurons reported in our previous study (**Li et al., Fig. 6c**).
Importantly, this serves as a methodological cross-validation of Li et al., in which
neuronal morphology was obtained via viral labelling followed by 3D reconstruction.

In the present study, neuronal morphology was determined by biocytin filling during
the recordings and thus represents actual recorded neurons rather than virus-labelled
populations. The side-by-side presentation demonstrates consistency between these
separate labelling approaches, improving the reliability of our morphological
observations across the studies. We have now clarified the purpose of this comparison
in the revised figure legend (**the Results section: line 126-128:** “*These neurons*
*exhibited a characteristic morphology, with long dendrites extending vertically into the*
*superficial layers (Fig. 1k), consistent with the findings of our other reports (H.L., J.C.)*”).

[FIGURE REDACTED]

**6c**, Example morphology of TeA_{dPAG} neurons in L5a (c).

- **Provide more details about the extracellular recording procedure. It seems that**
**the recordings in TeA are performed by inserting the pipette orthogonal relative**
**to the cortex surface, which suppose a close to horizontal angle. How is it achieved**
**practically speaking (size of the craniotomy, recording chamber, reference**
**electrode, how is the brain protected during the recording)? Also describe the**
**protocol for juxtacellular labelling.**

Thank you for your comment. We have revised and expanded the Methods section to
include a more detailed description (**line 843-847:** “*By adjusting the configuration of*
*the three-axis series manipulator (MX1641 Series Manipulator, Siskiyou, USA), we*
*aligned the direction of advancement to a horizontal orientation (parallel to the coronal*
*axis of the brain). Subsequently, during the electrophysiological recordings, the pipette*
*was advanced vertically from the superficial layers to the deep layers of the TeA with*
*the micromanipulator.*”). For the most direct demonstration, please refer to

**Supplementary Video 1** provided with our submission.

In our early attempts to record from the TeA region, we employed a vertical recording
approach involving a three-axis micromanipulator (MX1641 Series, Siskiyou, USA),
with the microelectrode drive (Siskiyou, USA) positioned above the mouse's head, and
advanced the electrode from dorsal to ventral. However, since the TeA is located
superficially near the ear, this traditional method makes it difficult to reliably target
neurons in the superficial layers of the TeA. Additionally, this approach limited our
ability to accurately determine the recording depth within the TeA.

We reconfigured the three-axis micromanipulator to achieve a horizontal insertion
trajectory (parallel to the brain's coronal plane). This new configuration allowed the
microelectrode drive (Siskiyou, USA) to advance the electrode perpendicularly through
the cortex of the TeA from the superficial to deep layers during the electrophysiological
recordings.

Regarding the surgical preparation phase, we localized the position within the TeA
along the sagittal plane using the same approach previously established for viral
injections into this region (**refer to Luo, B. et al²**). After determining the TeA
distribution range posterior to bregma, we combined this with brain atlas measurements,
which revealed that the TeA region forms an $\sim 85^\circ$ angle relative to the midsagittal plane.
This allowed us to rotate the mouse within the stereotaxic apparatus during preparation
to align the TeA with the vertical axis.

At this location, we thinned the skull using a miniature drill bit (diameter = 0.5 mm).
Postsurgery, erythromycin ointment was applied to prevent infection, and the exposed
skull area was protected with Vaseline. On the final training day under the head-fixed
condition, we completed the craniotomy and carefully opened the dura mater, which
was again covered with Vaseline. During the recording sessions the following day, the
Vaseline was removed to allow electrode insertion. This approach provides direct visual

access to the cortical surface, enabling precise determination of electrode depth.

**Supplementary video 1:** Firing of a single TeA neuron recorded in a mouse running on a
turntable.

- **line 158-163: this sentence does not make sense to me.**

Line 158-163: *“However, this concept challenges our general understanding that*
*bioinformation is typically encoded by populations of neurons with similar parameters.*
*It seems unlikely that a group of neurons would compute different v_m values to encode*
*the same running speed. Therefore, we hypothesized that the running signal is encoded*
*by the extent of the firing rate increment ($f - f_0$) within the R-neurons population, where*
*the v_m values are uniform to drive running events.”*

We appreciate your suggestion. Indeed, we also agree that the assumptions in this
section lack a sufficient basis and are not particularly meaningful. In fact, we simply
adopted an appropriate processing approach to reduce the variables, aiming to identify
measurement variables that can distinguish the different encoding capabilities between
TeA and dPAG neurons (**The relevant content is described in the Results section:line**
**183-193:** *“For both TeA and dPAG R-neurons, the values of f_0 , τ and v_m calculated*

*through curve fitting differed among events and different neurons (Fig. 2m). This*
*finding suggests that neuronal activity encodes running speed through the interaction*
*of different neurons, each with distinct coding parameters (f_0 , τ and v_m), to generate a*
*running event. From this perspective, each neuron, even when described in the context*
*of an event with three fit parameters (f_0 , τ and v_m), is special and cannot be compared*
*with others. To compare the encodings within a neuron or a nucleus, we fixed the values*
*of f_0 and v_m for all events or neurons and recalculated the neuronal firing effect constant*
*τ . Specifically, we reduced the fit parameters and standardized the running–firing*
*curves by setting the v_m values to their mean and f_0 to zero (see Methods).”).*

**- line 302-303, figure 4g: Backing and stopping should be dissociated. Interestingly,**
**the duration of ‘backing’ appears to be constant, regardless of the duration of the**
**opto-stimulation, but the duration of ‘stopping’ lasts as long as SC is stimulated!**

In this revision, we now distinguish the concepts of "backing" and "stopping." Strictly
speaking, as referenced in our **Supplementary Video 2**, during low-frequency
optogenetic stimulation (<20 Hz), stimulation of the SC activates SOM neurons in the
dPAG and inhibits T_e AdPAG neurons, causing the mouse to exhibit a stopping behaviour.
As the optogenetic stimulation frequency increases, the mouse displays a backing
behaviour, which is caused by activation of s_{cd} PAG neurons controlling their ability to
back away. At this point, the inhibition of T_e AdPAG neurons by SOM neurons in the
dPAG persists. However, as seen in the video, the posture of the mouse when not
backing away differs from the stopping posture observed during low-frequency
stimulation—the mouse still maintains a backing stance. However, some mice exhibited
a sustained backing away posture during SC stimulation (**Supplementary Video 2**).

Therefore, we believe that during high-frequency SC stimulation, the mouse exhibited
backing-away behaviours. However, the lack of sustained backing away behaviours
may be attributed to two reasons. First, as shown in **Fig. 4e and f**, the duration of the

backing away behaviour is affected by the stimulation frequency; indeed, the data
presented in **Fig. 4g** were obtained at the maximum stimulation frequency, at which
point neuronal stimulation may have reached saturation. Second, the mouse may
actively resist involuntary backing away behaviours by using its limbs to counteract the
corresponding movements.

We now clarify the distinction between stopping and backing away behaviours in the
main text (**line 346-365**: “*Upon optogenetic activation of SC_{capag} neurons at different*
*stimulation frequencies, we observed frequency-dependent behavioural responses (Fig.*
*4e and f). Specifically, lower-frequency (<20 Hz) stimulation elicited stopping (no*
*movement) during the stimulation period, whereas high-frequency stimulation*
*triggered backing away behaviours. Notably, rebound running was observed after*
*stimulation in both frequency ranges was ended. These behaviours were not observed*
*in the control group injected with viral vectors lacking ChR2 (Fig. 4f). Moreover, the*
*stopping and backing away behaviours observed on the turntable (see supplemental*
*video 2) were confirmed in an open field test (see supplemental video 3). As shown in*
*supplementary video 2, the mice maintained backing away behaviours during*
*stimulation, although sustained backing away was not observed during high-frequency*
*stimulation in either the open field test (see supplemental video 3) or on a turntable*
*(Figs. 4g1-4). However, the mice did maintain a backing stance, which differed from*
*the stopping posture observed during low-frequency stimulation. Following the*
*stopping behaviours, stimulation at lower frequencies (<20 Hz) subsequently induced*
*weak, short rebound running (Fig. 4e and f). These observations indicated that stopping*
*was clearly the result of the activation of a specific type of neuron or neural circuit,*
*leading to immobility by ending a motion and that this behaviour was distinct from the*
*resting behaviours observed after rebound running and normal running (see*
*supplemental videos 2 and 4).”, **line 378-383**: “To explore this possibility further, we*
*stimulated the SC for 5, 10, and 15 s at 40 Hz. The results revealed that rebound running*
*behaviours occurred after the stimulation ended (Fig. 4g1 and g2) and that the duration*
*of the backing response was correlated with the stimulation duration (Fig. 4g3).*

Additionally, the peak backing away and rebound running speeds did not differ
 significantly across the three stimulation durations (Fig. 4g4.)” accordingly.

Fig. 4 d-g

 **4d**, Virus injection and ChR2 expression in the dPAG and SC. **e and f**, Movement speed heatmap
 (**e**) and average speed over time (**f**) of mice stimulated at nine different optogenetic frequencies. **g**,
 Behavioural responses to BL stimulation for durations of 5, 10, and 15 s and a frequency of 40 Hz
 in the SC (**g**). Panels 1-4: (1) Movement speed heatmap, (2) average speed versus time, (3) response
 duration versus stimulation duration, (4) statistical comparison of the peak speed for backing away
 or rebound running behaviours. (**g**): backing away: $F_{(2, 8)} = 2.365, P = 0.156$; rebound running: $F_{(2,$
 $8)} = 0.142, P = 0.870$; One-way repeated-measures ANOVA with Bonferroni post hoc correction
 (error bars: SDs).

- line 360-363: This is a reasonable assumption but it is not shown in the current
 study. That could go to the Discussion.

Line 360-363: “SOM neurons innervate *TeAdPAG CaMKII α* neurons. Activation of SOM
 neurons inhibits *TeAdPAG* neurons and results in the stopping of running behavior. When
 the SOM-mediated inhibition is withdrawn, the *TeAdPAG* neurons become rebound
 excited, leading to running behavior.”

We sincerely appreciate your suggestion. We now incorporate the relevant content into
 the Discussion section in our revised manuscript (line 594-597: “Activation of *scdPAG*
 *CaMKII α* neurons excites *scdPAG* SOM neurons, which in turn inhibit *TeAdPAG* neurons,

*resulting in the stopping of running behaviours. When SOM-mediated inhibition is*
*withdrawn, TeAdPAG neurons become excited reboundedly, leading to the generation of*
*running behaviours.”).*

**- Provide more details about the extracellular recording procedure. It seems that**
**the recordings in TeA are performed by inserting the pipette orthogonal relative**
**to the cortex surface, which suppose a close to horizontal angle. How is it achieve**
**practically speaking?**

This question has been previously addressed.

**Minor comments:**

**- First paragraph of Results (lines 89-94) should move to the Introduction.**

First paragraph of Results, lines 89-94: *“The neurons in TeA that project to dPAG*
*(TeAdPAG) control running behavior (H.L., J.C.). These neurons do not respond to*
*multimodal sensory stimuli such as sound, light, or air puffs. Instead, their firing*
*precedes running behavior, and the firing rate correlates with running speed. These*
*neurons are termed "running-related neurons" (R-neurons), or motor/behavior*
*command neurons (H.L., J.C.), suggesting that they encode running behavior.”*

We sincerely appreciate your suggestion. In response to this comment, we now
incorporate this content in the Introduction section of the revised manuscript (**line 73-**
**78**: *“Our accompanying study (H.L., J.C.) reports that TeAdPAG neurons do not respond*
*to multimodal sensory stimuli such as sound, light, or air puffs; however, their firing*
*precedes the initiation of running behaviours, and their firing rates are correlated with*
*running speed. These neurons are termed "running-related neurons" (R-neurons) or*
*motor/behaviour command neurons (H.L., J.C.), indicating that they encode running*
*speed.”).*

- **Line 210: it should be specified that the AAV-Cre virus used here is an**
**anterograde transsynaptic serotype that will drive expression of Cre in the post-**
**synaptic neurons receiving inputs from TeA.**

We sincerely appreciate your suggestion. In response, we have meticulously verified
and provided detailed descriptions of all viral serotypes in the manuscript, including
the transsynaptic viral variants (**the Results section: line 242-244:** “*We used*
*anterograde transsynaptic AAV-Cre viruses designed to drive the expression of Cre in*
*the postsynaptic neurons that receive input from the TeA.*”, **line 339-340:** “*After both*
*anterograde transsynaptic serotype AAV systems, Flp and Cre, were used (Fig. 4a),*”;
**the Methods section: line 708-709:** “*Specifically, AAV2/1 was used for anterograde*
*monosynaptic tracing*³⁸⁻⁴⁰.”).

- **Line 281, figure 4e, f: I do not see any sign of ‘backing away’ when the stimulus**
**is withdrawn?!**

We apologize for the lack of precision in our original phrasing. The intended emphasis
was that rebound running occurs specifically after either stopping or backing away
behaviours when the stimulus is withdrawn. We have now corrected these issues in the
revised manuscript (**the Results section: line 346-353:** “*Upon optogenetic activation*
*of SC_{dPAG} neurons at different stimulation frequencies, we observed frequency-*
*dependent behavioural responses (Fig. 4e and f). Specifically, lower-frequency (<20*
*Hz) stimulation elicited stopping (no movement) during the stimulation period, whereas*
*high-frequency stimulation triggered backing away behaviours. Notably, rebound*
*running was observed after stimulation in both frequency ranges was ended. These*
*behaviours were not observed in the control group injected with viral vectors lacking*
*ChR2 (Fig. 4f).*”).

- **Clearly indicate on each figure panel when is the stimulus presented (ex: figure**

3 a-e2 right panel, 4f, 4g2 and h2 ...)

We have incorporated the suggested revisions in accordance with your
recommendations (Fig. 3a-e2 right panel, Fig. 4f, 4g2 and h2, Fig. 5b, 5g, 5j right
panel, and Fig. 6a2, b2, 6g, 6j right panel).

Fig. 3 a2-e2

Fig. 4 e,f,g,h

Fig. 5b,5g,5j

Fig. 6a2,b2,6g,6j

References:

1 Evans, D. A. *et al.* A synaptic threshold mechanism for computing escape
decisions. *Nature* **558**, 590-594 (2018).

2 Luo, B. *et al.* Frequency-Dependent Plasticity in the Temporal Association
Cortex Originates from the Primary Auditory Cortex, and Is Modified by the
Secondary Auditory Cortex and the Medial Geniculate Body. *The Journal of*
*neuroscience : the official journal of the Society for Neuroscience* **42**, 5254-
5267 (2022).

**H.L., J.C. et al. "An intralayer microcircuit underlies sensory-induced escape".**
**Submitted as a companion paper.**

Dear Reviewer,

The line numbers referenced in our point-by-point responses correspond to the
"Revised manuscript without track changes-chen.pdf" file.

**Reviewer #2-chen**

**The manuscript by Chen et al. evaluates the role of neuronal projections from the**
**temporal association cortex (TeA) and superior colliculus (SC) to the dorsal**
**periaqueductal gray (dPAG) in regulating running behavior. Using in vivo loose-**
**patch recordings, optogenetic, and chemogenetic approaches, the authors find that**
**the neuronal firing rate of the TeA-dPAG pathway correlates with running speed,**
**while the SC-dPAG pathway regulates backward movements in mice. They**
**further suggest that rebound running and stopping behaviors on the rotating disk**
**involve a complex circuit motif between these regions. While the manuscript**

**presents intriguing data, several critical issues limit its impact, and it is not**
**suitable for publication in Nature Communications in its current form.**

We sincerely appreciate the reviewers' critical comments, which challenged us to
strengthen the rigor of our study. We have implemented all suggestions to the best of
our ability.

**Major Concerns:**

**1. Lack of Ethological Relevance: While the study elegantly demonstrates the**
**involvement of the TeA-dPAG and SC-dPAG pathways in modulating running**
**behavior, the connection between the observed behaviors and their ethological**
**relevance is unclear. The authors frequently describe rebound running and**
**backward movement as "defensive responses." However, these behaviors are not**
**typically observed in naturalistic studies of rodent defense. It is possible that these**
**behaviors result from the head-fixed setup on the rotating disk, rather than**
**representing natural defensive responses.**

We apologize for any lack of clarity. This study serves as a direct continuation of our
other manuscript submitted simultaneously (Li et al.), in which we discovered running-
related neurons (R-neurons) in the TeA, defined as neurons that (1) show no sensory
responsiveness; (2) fire preceding running behaviours; and (3) demonstrate firing rate-
speed correlations. The present work specifically investigated how these neuronal
populations encode running speed. Given that SC-dPAG pathways are also implicated
in the regulation of running behaviours, we systematically investigated this circuit for
comparative analysis with the TeA-dPAG pathway identified in our previous work.
However, when studying these two pathways, we did not adopt a defence model, so it
is impossible to determine whether these behaviours are indeed involved in animal
defence responses.

We initially attributed these backing behaviours to defensive responses on the basis of
two key observations: SC-dPAG circuit activation elicited backing responses, and
stimulation of the deep SC (dlSC)-PAG pathway typically evokes escape behaviours,
as reported previously¹ (Wei, P. et al. 2015). Both the supplementary video evidence
from Wei et al.¹ (Supplementary Movie 3) and our experimental results
(Supplementary Video 3) demonstrate that backing away behaviours can occur in
freely moving conditions in open-field environments and are not limited to head-fixed,
running-wheel paradigms. However, neither prior studies nor our current work
systematically verified whether these behaviours constitute defensive responses.
Therefore, we recognize that simply categorizing backing behaviours as a defensive
reaction would be premature; however, it should be noted that Wei et al. made this
consideration.

We acknowledge that categorizing these behaviours as defensive responses was indeed
imprecise. In accordance with your suggestion, we have revised the entire manuscript
to reflect this correction.

**2. Misinterpretation of Stopping Behavior: The authors have misinterpreted**
**stopping as freezing, a well-characterized defensive state that differs**
**fundamentally from the absence of locomotion. Freezing is associated with distinct**
**postural adjustments, reduced respiratory rate, and increased muscle tension**
**(Roelofs, 2017). Clarifying this distinction is essential for accurately interpreting**
**the data.**

We sincerely appreciate your assistance in clarifying the distinction between freezing
and stopping behaviours. Upon further consideration, we recognized that the initial
interpretation of stopping as freezing was oversimplified. We have therefore revised
these statements in the manuscript and added a comparative discussion of the two
concepts in the Discussion section (line 598-601: *“Stopping is an active cessation of*
*movement achieved through the activation of specific cells that manifests as an*

*immobile state in mice during low-frequency SC stimulation. Even when high-frequency*
*stimulation induces backing away behaviours, the inhibitory effect mediated by these*
*specific neurons remains persistent and active.”; line 606-617: “The cessation of*
*running behaviours is the unit behaviour controlled by the scdPAG SOM behavioural*
*unit. However, freezing, a learned fear-related defensive behaviour, can manifest in a*
*variety of ways, such as immobility^{32,34}, the cessation of locomotion (e.g., escaping,*
*running, and walking)^{35,36}, immobility accompanied by shaking³⁷, or halting during an*
*intermediate action state³³. As shown in Fig. 4e and g, Fig. 5b and g, and Fig. 6a2, b2,*
*g, and j, the stopping or ending of motion that occurs during neural activation can be*
*classified as freezing, indicating active inhibition and cessation of locomotion. When*
*immobility occurs at the end of running or rebound running behaviours, it represents a*
*state of resting rather than a cessation of locomotion, as the firing rates of the*
*corresponding running or rebound running units are insufficient to trigger the*
*associated behaviours (Figs. 1b and 2b).”).*

The behavioural arrest induced by dPAG or SC stimulation was classified as freezing
in accordance with the study by **Evans DA et al². (Evans DA, 2018 nature)**. However,
since our experimental paradigm did not employ a defensive context, we use the term
'stopping' rather than 'freezing' to describe this behavioural state. Strictly speaking,
“stopping” is an active cessation of movement achieved through the activation of
specific cells that manifests as an immobile state in mice during low-frequency SC
stimulation. Even when high-frequency stimulation induces back-away behaviour, the
inhibitory effect mediated by these specific neurons remains persistently active. In this
revision, we have systematically removed all the references to 'freezing' and have
modified the remaining text in the manuscript accordingly.

**3. Lack of Proper Control Groups: The manuscript does not include optogenetic**
**or chemogenetic control data, making it difficult to determine whether the**
**observed effects result specifically from neuronal modulation. Including**
**appropriate control groups would substantially strengthen the conclusions by**

**ruling out potential confounds and providing stronger evidence for the causal role**
**of these pathways in the observed behaviors.**

We sincerely appreciate your suggestion. All the experiments did include appropriate
control groups and procedural controls, but they were intentionally omitted from the
manuscript due to concerns regarding redundancy. The complete datasets and statistical
analyses have now been incorporated into the main figures of the revised manuscript
(**Fig. 3a2-e2, line 246-248:** “*The animals responded to blue light stimulation by*
*running, the maximum speed of which increased as the stimulation frequency increased*
*(Fig. 3a2-e2); however, none of these behaviours were observed in the control group*
*(Fig. 3a2-e2).”*; **Fig. 4f, line 350-353:** “*Notably, rebound running was observed after*
*stimulation in both frequency ranges was ended. These behaviours were not observed*
*in the control group injected with viral vectors lacking ChR2 (Fig. 4f).”*; **Fig. 5c, h, and**
**k, line 410-413:** “*The control group (ChR2 virus + hM4D-free virus), meanwhile,*
*exhibited no CNO-induced specific blockade of their rebound running behaviours (Fig.*
*5c). These findings confirmed that rebound running results from activation of T_{eAdPAG}*
*cells.”*, **line 435-438:** “*The control group (ChR2 virus + hM4D-free virus) exhibited no*
*CNO-induced specific blockade of rebound running behaviours. In contrast, the*
*running behaviours induced by the activation of T_{eAdPAG} CaMKII α neurons remained*
*unchanged, which was also consistent with the findings of the control group (Fig. 5i-*
*k).”*; **Fig. 6a2, b2, h, and j, line 459-460:** “*These behaviours were not observed in the*
*control group, which was injected with virus vectors lacking ChR2 (Fig. 6a2).”*, **line**
**465-466:** “*These behaviours were likewise not observed in the ChR2-free control group*
*(Fig. 6b2).”*, **line 498-501:** “*The control group, which received coinjections of ChR2-*
*expressing virus and hM4D-free virus, showed no CNO-induced specific blockade of*
*backing away or rebound running behaviours (Fig. 6 g and h).”*, **line 508-509:** “*Similar*
*to the findings above, these behaviours were not observed in the ChR2-negative control*
*group (Fig. 6j).”*).

Fig. 4f

Fig. 3 a2-e2

Fig. 5 c,h,k

Fig. 6 a2,b2,h,j

**4. Normalization of Viral Expression Data: Direct comparisons of the effects of**
 **different viral preparations (e.g., in Figure 3) may be confounded by differences**
 **in expression efficiency and tropism. To address this, data on the number of**
 **neurons with viral expression should be provided. Ideally, behavioral data should**

**be normalized to account for variations in viral expression between different**
**stimulation approaches.**

We sincerely appreciate your suggestion. However, as evidenced by our histological
imaging results, ChR2-expressing neurons exhibited uniformly high fluorescence
intensity, with clear labelling not only in the neuronal nucleus but also throughout the
axonal and dendritic arbours. This comprehensive labelling pattern made accurate
determination of the numbers of transfected neurons exceptionally challenging.

However, our experimental results demonstrated that despite the use of different viral
injection approaches, the magnitude of the behavioural responses was significantly
influenced by multiple factors, including the location where the virus was injected; the
type, titre and expression of the virus; the source, intensity and frequency of the light
stimulus, and so on, all of which affected running speed. Therefore, the behavioural
data should be normalized to account for differences in viral expression and other
factors. To normalize the behavioural data, we converted the optogenetic stimulation
frequency into the corresponding running speed and plotted the running latency against
running speed to eliminate the influence of the latter (**Fig. 3k, line 306-318**).

**Fig. 3k**

**Figure legends**

**3k**, Running latency versus running speed.

**5. Clarity of Statistical Analysis: The manuscript lacks clarity on the statistical**

analyses used, and some data comparisons, such as those in Figure 3f, are
confusing. Detailed explanations of statistical methods and justification for the
comparisons made are necessary to ensure transparency and reproducibility.

Thank you for bringing this to our attention. In this revision, we have significantly
improved the readability, reproducibility, and transparency of the manuscript by
incorporating additional details regarding the statistical methodologies and rationale.
These improvements, including comprehensive descriptions of our statistical
approaches and their justifications, have been systematically integrated throughout the
main text, figure legends, and Methods section.

In Fig. 3f, we aimed to determine whether there were significant differences in speed
following the activation of neurons in different brain regions and neurons labelled with
different viral strategies in the same region. Therefore, our original direct comparison
between TeA neurons (Fig. 3a) and TeAdPAG neurons (Fig. 3e) was inappropriate. In
this revision, we have corrected the details of the comparative analysis (**The relevant
content is described in the Results section: line 252-263:** “*A comparison of the
maximum speed values revealed differences in the driving effects of TeA and dPAG
neurons. The v_m for TeAdPAG neurons (35.10 ± 3.92 cm/s) was the highest, followed by
TeA CaMKII α neurons (22.81 ± 3.07 cm/s). However, there were no significant
differences among the v_m values for TeAdPAG fibres (9.44 ± 0.77 cm/s), TeAdPAG
CaMKII α neurons (8.6 ± 0.61 cm/s), and TeAdPAG neurons (8.68 ± 0.53 cm/s) (Fig. 3f).
Notably, activation of TeA CaMKII α neurons differed from that of TeAdPAG fibres,
TeAdPAG CaMKII α neurons, and TeAdPAG neurons, as the latter three involved
projections from the TeA to the dPAG, whereas TeA CaMKII α neurons included
TeAdPAG neurons and neurons projecting to TeAdPAG neurons. The neurons projecting
to TeAdPAG neurons increased activation of the latter, explaining why the activation of
TeA CaMKII α neurons evoked greater running speeds than did the activation of
TeAdPAG fibres, TeAdPAG CaMKII α neurons, or TeAdPAG neurons.”).*

Fig. 3 f

**3f**, Values of v_m (**f**) obtained from **a3-e3**. (**f**): $F_{(4, 24)} = 157.4$, $P < 0.0001$, one-way ANOVA with
 Bonferroni post hoc correction.

**6. Reciprocal Connectivity of Pathways: Since AAV-Cre transsynaptic viruses can**
 **express retrogradely, the authors should investigate whether the TeA-dPAG and**
 **SC-dPAG pathways have reciprocal connectivity. This information would provide**
 **critical insights into the circuit dynamics underlying the observed behaviors.**

Thank you for your comment. On the basis of our experimental results and data from
 the Allen Brain Atlas: for the TeA-dPAG pathway, retrograde viral labelling via
 injection into the TeA did not result in labelled somata in the dPAG (as shown in the
 figures below). When retrograde tracers were injected at selected SC injection sites, we
 still observed no somata expression in the dPAG, whereas expression in other regions
 (e.g., V1) appeared normal (see the figures below). These results indicate that with our
 injection approach, there is no detectable cross-communication between the TeA-dPAG
 and SC-dPAG pathways.

The Allen Brain Atlas data show no evidence of dPAG→TeA projections. Regarding
 dPAG→SC projections, while viral injections targeted the dPAG region, there was
 minor leakage into deep SC motor-related regions. However, labelled terminals were
 primarily observed in sensory-related SC region (medial region of intermediate layers),
 which is distinct from our injection site (lateral region). This confirms the anatomical
 specificity of our experimental approach.

Allen Brain

**a**, No retrograde mCherry expression in dPAG after SC AAVretro injection. **b**, AAVretro-hSyn-

mCherry injected in SC shows retrograde labeling in V1. **c**, AAVretro-hSyn-mCherry injection in

TeA. **d**, No retrograde viral expression in dPAG following AAVretro-hSyn-mCherry injection in

TeA. **e**, Anterograde viral tracing from PAG (left and middle) shows labeling in deep SC (left and

middle) but not TeA (right) (Allen Brain Atlas).

**References:**

1 Wei, P. *et al.* Processing of visually evoked innate fear by a non-canonical
thalamic pathway. *Nature communications* **6**, 6756 (2015).

2 Evans, D. A. *et al.* A synaptic threshold mechanism for computing escape
decisions. *Nature* **558**, 590-594 (2018).

**H.L., J.C. et al. "An intralayer microcircuit underlies sensory-induced escape".**

**Submitted as a companion paper.**

Dear Reviewer,

The line numbers referenced in our point-by-point responses correspond to the
"Revised manuscript without track changes-chen.pdf" file.

**REVIEWER COMMENTS**

**Reviewer #1 (Remarks to the Author):**

**This revised version of the manuscript present convincing evidence that SC**
**neurons induce backing away movement via the monosynaptic activation of dPAG**
**CaMKII positive neurons followed by rebound running via the activation of SOM**
**neurons in the dPAG likely producing rebound excitation of another, non-**
**overlapping population of dPAG CaMKII positive neurons receiving inputs from**
**TeA. Despite an evident effort of clarification, this revised version of the**
**manuscript still suffers from important flaws in the way the data are presented**
**and analyzed in my opinion. In particular, I am still not convinced by the way the**
**correlation between the firing-rate and the running speed is analyzed. I still**
**strongly believe that aligning each running event to the previous peak in firing**
**rate present major risk of false positive. In fact, figure 1 shows quite a few neurons**
**for which the Vm value fitted to the data is very low, close to 0, indicating almost**
**no correlation between the firing rate and the running speed. In addition, the**
**overall logic throughout the result section is often difficult to follow with some**
**experiment lacking clear rational and some results that are misinterpreted.**

**Major concerns:**

**- One of the major issues is the way the correlation between the neuronal activity**
**and the running speed is analyzed. First, it is still unclear how exactly R-neurons**
**are defined: which statistical analysis is used to determine whether the neurons**
**respond to the sensory stimuli or not? is the correlation between firing rate and**

**running speed computed throughout the entire recording or just for running**
**events? I would find surprising to see any significant correlation between the firing**
**rate and the running speed with a ~2 s delay between the two. Again, it would seem**
**fairer to me to compute the cross-correlogram between the firing rate and running**
**speed to obtain the peak of the correlation and time lag between the two.**

We apologize, as it seems that we did not fully understand your previous question,
particularly regarding the cross-correlation function (CCF) analysis. After carefully
reviewing the current suggestions and combining them with the first round of feedback,
we now understand your inquiry. Our previous statistical analysis of the relationship
between the firing rate and running speed was insufficient. In this revision, in
accordance with your comments, we have added new figures and modified the
manuscript accordingly.

In this revision, first, in the Li et al. manuscript (Manuscript NCOMMS-24-78434T), based
on your suggestion, we performed the following analyses to define R-neurons: We first
determined that the recorded neurons did not respond to sensory stimuli. Based on the
sensory stimulus–running time window statistics (0–3.4 s) for mice running on the
turntable obtained by Li et al., we selected events where the sensory stimulus did not
elicit running and analysed the relationship between neuronal firing and the sensory
stimulus. An analysis of the peristimulus time histogram (PSTH) (**Figs. 1c1 and c2;**
**2c1 and c2**) confirmed that the neurons did not respond to the sensory stimuli. We
identified R-neurons by analysing their peri-event time histograms (PETHs) of firing
relative to running onset during spontaneous running (**Figs. 1d1 and d2; 2d1 and d2**).
The time window of the increase in firing observed in these PETHs was subsequently
determined.

After identifying the R-neurons, we subsequently performed a CCF analysis between
the firing rate and running speed (**Figs. 1e2, f, g and 2e2, f, g**). We analysed the firing
rate and running speed using the entire recording session directly, without segmenting

it into individual running events. Our results showed that the strongest correlation
between firing and running speed was achieved when the running speed curve was
shifted forwards, with the best correlation coefficients (TeA $r_{\max} = 0.898$, dPAG $r_{\max} =$
0.761) indicating a positive correlation. The optimal time lags were TeA $\tau = 1.97 \pm$
0.6114 s and dPAG $\tau = 0.94 \pm 0.4001$ s, respectively. These lags also fell within the
firing–running time window we statistically determined (TeA: $-3.14-0$ s; dPAG: $-$
$1.197-0$ s), indicating that changes in firing activity still occurred before changes in
behaviour and that neural activity encoded the behaviour. The CCF plots showed that
their peaks were distributed across a relatively broad time window (full width at half
maximum, FWHM TeA = 10.57 s, dPAG = 6.81 s), implying a strong association
between neural activity and running speed parameters over a sustained period rather
than being locked to a precise moment. This pattern is consistent with the hypothesis
of “running speed encoding”, where the neuronal firing rate is continuously modulated
throughout the phases of movement preparation and execution to reflect speed
information.

We validated our hypothesis by plotting the relationship between the neuronal firing
rate and the animal’s running speed curve using the zero time lag from the CCF analysis
(i.e., no shift). This plot shows that as neuronal activity increases, the animal’s running
speed also gradually increases. However, neural encoding is often phase-locked to
behavioural event timing. The complete curve includes extensive periods of animal
immobility (low speed) and low spontaneous neuronal firing; this “noise” can obscure
the encoding signals phase-locked to movement initiation, making fitting the functional
relationship between the two difficult. Furthermore, the time difference between firing
events and the corresponding running events varied across trials (**Fig. 1b, values**).
Therefore, we analysed the data by segmenting them into individual events based on
each running bout.

We did not use the optimal time lag obtained from the CCF analysis for this event-based
analysis because, while this lag maximizes the statistical association strength from a

purely mathematical and signal processing perspective, using it would disrupt the real
temporal structure of the neural population activity. Each neuron would be shifted to a
different time point, and the resulting “averaged firing rate curve” is something that
never actually occurs in the real world. It resembles an “optimal statistical template”
more than a “real physiological process”. Therefore, we adhered to the original
analytical method, using the maximum value and the onset point, and compared the
results.

Based on our results, the firing of TeA R-neurons is indeed correlated with running
speed, and their firing encodes information about running speed.

Regarding the point about the low match (close to 0) between the neuronal v_m and the
data in Fig. 1, indicating little correlation between the firing rate and running speed, we
believe this comment refers to **Fig. 1o and 1p**. In response, we have added a
clarification regarding this point to **the Discussion section, line 765-774**: “*It should be*
*noted that the peak firing rate–peak/onset-aligned shifting running speed analysis*
*yields relatively low fit values for some neurons (Fig. 1o and p). This does not*
*necessarily imply that these neurons are unrelated to locomotor activity, but rather*
*reflects the limitation of reducing each trial to a single peak value. Such an approach*
*discards the temporal structure of both firing and behavior, thereby underestimating*
*the true relationship. Indeed, when neural activity is aligned to running onset and the*
*full firing–speed time course is analyzed, the majority of neurons show high fit values*
*(Fig. 1r), indicating a strong modulation by running speed. These results highlight the*
*importance of considering temporal dynamics when quantifying neuronal encoding of*
*behavior.”.*

Fig. 1

Fig. 2

**Fig. 1 c1 and 2, (c1)** Responses of a representative neuron **(b)** to three sensory stimuli: auditory
 (sound), visual (light), and tactile (air puff) stimuli. Top panel: Raster plots of individual trials and
 their corresponding running speeds. Bottom panel: Peristimulus time histogram (PSTH) of firing
 activity ($n = 4$ trials). **(c2)** Population neuronal responses to three sensory stimuli. Top
 panel: Population raster plot and the corresponding PSTH of firing activity. Bottom
 panel: Superimposed running speed from all trials ($n = 158$ trials). The blue bars indicate the
 stimulus period (duration: 1 s). **d1 and 2, (d1)** The same as **(c1)**, analysing spontaneous running
 events encoded by a representative neuron **(b)**. Bottom panel: Peri-event time histogram (PETH) of
 firing relative to running onset during spontaneous running ($n = 5$ trials). **(d2)** The same as **(c2)**,
 analysis of the neuronal population ($n = 33$ trials). **e2**, Cross-correlation function (CCF) analysis for
 population analysis ($n = 29$ neurons, $r_{max} P < 10^{-4}$) **(e2)**. The black line represents the mean CCF
 curve; grey lines indicate individual neuronal CCF traces. **f**, Distribution of the maximum
 correlation coefficient (r_{max}) from the analysis of **Fig. 1e2**. The red dashed line represents the
 Gaussian fit curve ($n = 29$ cells, $R^2 = 0.6755$, $P = 4.3414 \times 10^{-4}$). **g**, Distribution of optimal time lag
 counts from the analysis of **Fig. 1e2** ($R^2 = 0.8564$, $P = 4.0999 \times 10^{-5}$).

**Fig. 2 c1 and 2, (c1)** The same as **Fig. 1c1**, an analysis of a representative dPAG neuron **(b)** ($n = 4$
trials). **(c2)** The same as in **Fig. 1c2**, an analysis of the dPAG neuronal population ($n = 76$ trials).
The blue bars indicate the stimulus period (duration: 1 s). **d1 and 2, (d1)** The same as **(2c1)**,
analysing spontaneous running events of a representative neuron **(b)** ($n = 5$ trials). **(d2)** The same
as **(2c2)**, analysis of the dPAG neuronal population ($n = 24$ trials). **e2**, The same as **Fig. 1e2**, CCF
analysis for population analysis ($n = 22$ neurons, $r_{\max} P < 10^{-4}$) **(e2)**. **f**, Distribution of r_{\max} from the
analysis of **Fig. 2e2**. The red dashed line represents the Gaussian fit curve (for the dPAG: $n = 22$
cells, $R^2 = 0.7993$, $P = 0.0027$). **g**, Distribution of the optimal time lag counts from the analysis of
**Fig. 2e2** ($n = 22$ cells, $R^2 = 0.9910$, $P = 0.0214$).

- **Also, in the companion paper the authors clearly state that the activity of some**
**neurons is not correlated with running. I suppose that the authors only report**
**running related neurons but no indication is provided as to how many neurons**
**were recorded in total, in how many mice, and how many of them where running-**
**correlated. It is also not clear whether the neurons recorded in TeA are newly**
**recorded neurons for this study or simply from the companion paper study. This**
**should be explicitly mentioned.**

In the original manuscript, we had already indicated the total number of running-related
neurons recorded for TeA and dPAG, as well as the number of animals used in **the**
**figure legends**. For the TeA: **Fig. 1q**, “*Count distribution of time differences based on*
*the onset time ($n = 42$ trials from 29 cells of 15 animals).” For the dPAG: **Fig. 2l**,
“*Firing rate–running speed events and their corresponding fitted curves ($n = 30$ trials*
*from 22 cells of 15 animals) for dPAG R-neurons.” In this revision, we have additionally*
reported the total number of recorded neurons: 102 neurons for the TeA and 57 neurons
for the dPAG. These values have now been included in **the Results section, line 179-**
**180**: “*We analysed a total of 29 R-neurons, which were obtained from a total of 102*
*neurons recorded in the TeA of 15 mice.” line 234-235*: “*We analysed a total of 22 R-*
*neurons obtained from 57 recorded neurons in the dPAG region across 15 mice.”**

The question of whether the data analysed originated from another manuscript has been
addressed in **the Methods section, line 1099-1100**: *“The data for TeA R-neurons*
*presented in this study differ from those reported in a previous paper by H.L. and J.C.”*

**- All along the paper, many experiments seem redundant or only bring minor**
**additional information. A better selection of what is shown in the main figures**
**could be done to make the flow of the paper easier and clearer.**

We sincerely thank the reviewer for this critical and constructive suggestion. We fully
agree that streamlining the figures is crucial for improving the clarity and impact of the
manuscript. In response, we have thoroughly reorganized the main figures to focus
squarely on the central narrative.

We have added representative peri-stimulus time histograms (PSTHs) in response to
sensory stimuli and peri-event time histograms (PETHs) aligned to spontaneous
movement onset in Figs. 1 and 2 to more definitively establish the basic response
properties of the recorded neurons. These new panels provide immediate visual
validation of how we identified sensory-responsive and running-related neurons,
thereby making the classification of R-neurons much clearer and more robust for the
reader from the outset.

We have removed several panels from the original Fig. 6 that contained supplementary
analyses to improve the flow and eliminate any perception of redundancy. This deletion
allows the reader to follow the main argument without unnecessary digressions,
ensuring that each figure in the manuscript delivers a decisive point.

We believe these revisions have significantly improved the manuscript by creating a
more focused and logical progression from characterizing the neurons to exploring their

functional roles, directly addressing the reviewer's concern.

- **Some statements are inaccurate or lack logical link or state the obvious (see below)**

**Other less critical issues:**

- **Introduction, line 73-74: in fact, there is no direct evidence that TEAdPAG**
**neurons do not respond to sensory stimuli in the companion paper.**

In the other manuscript, we did not directly demonstrate that TeAdPAG neurons do not
respond to sensory stimuli. Instead, we relied on several indirect lines of evidence: for
example, their neuronal distributions are highly similar within TeAdPAG (located in layer
5), their firing—whether spontaneous or optogenetically evoked by blue light—
consistently elicits running behaviour in mice, and the differences in time between
neuronal activity and running onset are also comparable. Furthermore, in the present
study, the morphology of biocytin-labelled neurons (**Fig. 1n**) is consistent with the
viral-based morphological characteristics of labelled neurons in the manuscript by Li
et al. (**Li et al., Fig. 6c**). Together, these findings provide strong indirect evidence that
TeAdPAG neurons represent a subset of TeA R-neurons.

Revisions related to this part have been made in **the Introduction, line 71-84**: “*Our*
*accompanying study (H.L., J.C.) revealed a population of neurons in the TeA that do not*
*respond to multimodal sensory stimuli such as sound, light, or air puff; however, their*
*firing precedes the initiation of running behaviours, and their firing rates are correlated*
*with the running speed. These neurons are termed “running-related neurons” (R-*
*neurons) or motor/behaviour command neurons (H.L., J.C.). We hypothesize that these*

*neurons likely encode running speed (H.L., J.C.).*

*Furthermore, we reported in an accompanying study (H.L., J.C.) that TeA_{dPAG} neurons*
*exhibit characteristics highly similar to those of R-neurons. These characteristics*
*include their laminar distribution in the TeA and anatomical morphology, ability to*
*directly drive running behaviour when optogenetically stimulated, and temporal*
*coupling between firing and running. These strong similarities suggest that TeA_{dPAG}*
*neurons represent a subclass of R-neurons, providing a strategic entry point for*
*genetically targeted investigations of R-neuron function.”.*

**- Introduction, line 78: this paper does not “investigate the mechanism by which**
**[TEAdPAG neurons] encode running speed”.**

Regarding the description of the neurons, we have replaced the specific reference to
“TeA_{dPAG} neurons” with “TeA R-neurons” in **the Introduction, line 86-87: “In this**
*study, we sought to investigate whether the firing rate of R-neurons encodes running*
*speed.”*

**- Results, lines 150-1452: “To initiate a movement, the animal appears to calculate**
**its maximum running speed (v_m) on the basis of τ , which corresponds to synaptic**
**efficiency.” The link between the τ of the fitted function that links firing rate and**
**running speed, and efficiency of a synapse (which one?) is totally speculative.**

Thank you for your suggestion. We have removed this speculative statement in the

current revision to increase the rigor of our manuscript.

- **Results, line 160: “this relationship is somewhat counterintuitive” Why ?!**

This relationship may appear somewhat counterintuitive because, although a linear
function also provides a high goodness-of-fit, from a biological perspective, an animal’s
running speed cannot increase indefinitely—each movement inevitably reaches a
maximum speed. Therefore, the single-phase association equation offers a better fit to
the biologically plausible neurobehavioural relationship. Revisions reflecting this
understanding have now been incorporated into **the Results section, line 214-218:**
*“While the linear function also demonstrates a high goodness of fit, from a biological*
*perspective, an animal’s running speed cannot increase indefinitely—each movement*
*invariably reaches a maximum speed. Therefore, the single-phase association equation*
*better captures the underlying neurobehavioural relationship in biological contexts.”*

A comparison of the rationality and advantages of the linear equation versus the single-
phase association equation in analysing the relationship between neuronal firing and
movement speed has been elaborated in **the Discussion section** of this manuscript
under the heading “Quantitative and qualitative encoding of behavioural speed”, **line**
**734-750:** *“A linear correlation between running speed and stimulation frequency or*
*intensity has been previously reported^{5,9}. We also performed this analysis and plotted*
*the peak firing rate against the peak running speed (Fig. 1o) or the corresponding*
*running speed at the time of peak firing (Fig. 1p and Fig. 2j). Both relationships could*
*be well fitted to linear equations as generalized linear models (data not shown).*
*However, after a thorough analysis of the data trends and careful consideration of their*
*physiological implications, we concluded that the single-phase association equation*

provided a more biologically plausible representation of the underlying neural–
behavioural relationship. Two explanations support this choice. First, while both the
linear and single-phase association equations fit the data trends in Fig. 1i well, the
single-phase association equation provided superior fitting accuracy (Fig. 1i, j, o, p),
as evidenced by its greater R^2 value than that of the linear regression equation (Fig. 1o
and p). Second, the linear fit function $y = k \times (x - x_0)$ yields a slope (k) with units of
$\text{cm}/(\text{s}\cdot\text{Hz})$ —representing distance (cm) rather than movement speed (cm/s)—which
contradicts the physiological interpretation of R-neuron activity as a controller of
speed. In contrast, the single-phase association equation yields a dimensionless time
constant (τ), enabling direct comparison with the data from dPAG neurons in Fig. 2.”.

- **Results, line 179-180: Statistics missing**

The statistical values have now been added to the main text in **the Results section, line**
**262-264**: “ The f_0 for dPAG neurons (6.68 ± 3.47 Hz) was lower than that for TeA
neurons (12.79 ± 8.56 Hz), whereas the v_m value did not differ significantly between the
two regions (TeA: 17.89 ± 12.02 cm/s ; dPAG: 23.39 ± 10.5 cm/s).”.

- **Results, line 184-189: trial-by-trial variability in neuronal response is highly**
**expected in the most neuronal circuits. I suppose the authors mean to suggest that**
**the encoding of the running speed most likely occurs at the level of a population of**
**neurons. Which something well described and largely admitted in both sensory**
**and motor systems.**

Yes, that is precisely the point we intended to convey, and it is the reason why we plan
to standardize the data from both TeA and dPAG neuron populations in our next steps.

In response to your comment, we have further refined the writing in this revision to
improve clarity for readers. Please see **the Results, line 267-275**: “*For both TeA and*
*dPAG R-neurons, the values of f_0 , τ and v_m calculated through curve fitting differed*
*among events and different neurons (Fig. 2p). These findings suggest that neuronal*
*activity encodes running speed through the interactions of different neurons, each with*
*distinct coding parameters (f_0 , τ and v_m), to generate a running event. From this*
*perspective, each neuron, even when described in the context of an event with three fit*
*parameters (f_0 , τ and v_m), is unique and cannot be compared with others. The encoding*
*of running speed is most likely to occur at the population level of neurons, and cells*
*within the same neuronal nucleus are likely to share common encoding*
*characteristics.”.*

- **Results, line 225-227: I do not understand this sentence. Also, the fact that**
**TeAdPAG neurons are CaMKII positive is largely expected since most cortical**
**projection neurons are excitatory (glutamatergic) and CaMKII is a widely used**
**marker for cortical excitatory neurons.**

Line 225-227: “*TeAdPAG neurons are calcium/calmodulin-dependent protein kinase*
*type II subunit alpha (CaMKII α) positive (H.L., J.C.), while inhibiting TeA neurons,*
*TeA-dPAG projections, and TeAdPAG neurons effectively halts running behaviours (H.L.,*
*J.C.)”.*

We included this explanation to clarify our subsequent use of the CaMKII promoter in
optogenetic viral constructs. Although this outcome was anticipated, whether TeA_{dPAG}
neurons belong to the CaMKII neuronal population has not been previously
demonstrated. This relationship was validated in our separate manuscript (H.L., J.C.),
and therefore, we directly cited and emphasized this conclusion here. However, we
recognize that the original wording overemphasized the identity of these neurons as
CaMKII-positive. We have therefore revised the text to present this association more
accurately in the **Results, line 309-319**: “*We aimed to manipulate these cells*
*optogenetically to explore how running speed is encoded by the activity of a specific*
*neuronal population (R-neurons). However, technical limitations prevented the direct*
*labelling and manipulation of R-neurons. Therefore, we turned to a functionally*
*relevant surrogate population. Based on our previous findings (H.L., J.C.) that TeA_{dPAG}*
*neurons likely constitute a subset of R-neurons and that the inhibition of TeA neurons,*
*TeA_{dPAG} CaMKII α neurons, TeA-dPAG projections, and $TeAdPAG$ neurons effectively*
*halts running behaviours, we focused our analysis on these cells. Using viral injection*
*strategies, we achieved specific labelling and manipulation of both TeA_{dPAG} and*
*$TeAdPAG$ neurons and optogenetically activated them at varying frequencies to probe*
*their encoding rules for running speed.”.*

- **Results, line 274-277: isn't it obvious?**

Line 274-277: “*Nevertheless, the τ_s values obtained through optogenetic activation (Fig.*
*3g) were significantly smaller than those derived from the running-firing function τ_f*

*(Fig. 2n). This difference arose from the use of different abscissa axes: stimulation*
*frequency versus firing rate.”*

We thank the reviewer for this insightful comment. We agree that the difference in the
abscissas (stimulation frequency vs. firing rate) is indeed a fundamental reason for the
discrepancy between τ_s and τ_f . Our intention in mentioning this discrepancy was not
merely to state the obvious but to provide a clear logical justification for the subsequent
experiment where we recorded neuronal activity to enable a direct comparison.

Based on the reviewer's suggestion, we have revised the text to eliminate any redundant
explanations and to more effectively articulate the experimental rationale. Please see
**the Results, line 367-370**: *“We converted the stimulation frequency to the firing rate to*
*enable a direct comparison between the dynamics obtained via optogenetic activation*
*(τ_s , Fig. 3g) and those from the running-firing function (τ_f , Fig. 2q), which were*
*measured on different abscissae (stimulation frequency vs. firing rate).”*

We believe that this revision strengthens the logical flow and concisely presents the
motivation for the next set of experiments. We thank the reviewer again for helping us
improve the clarity of our manuscript.

- **Results, line 342-343: On the picture in Figure 4b there seems to be much more**
**double positive (yellow) cells than reported?**

Yes, the labelling of double-positive (yellow) cells in the original schematic was
inaccurate. We excluded cases where red and green signals overlapped but did not
originate from the same neuron (as indicated by the blue arrows in **Fig. a, b** and **d**,
below). However, in the initial schematic, neurons labelled in red with incomplete
morphological filling were not accounted for. The revised representation of double-
positive neurons is shown in Fig. 4a below. Upon re-examination, we found that the

originally selected schematic—both in the detailed view (**Fig. a**) and the overview of
the dPAG (**Fig. b**)—yielded higher proportions of double-positive cells than the average
(**Fig. a**: 12.245%; **Fig. b**: 4.676%). In contrast, the double-positive ratios calculated
from **Fig. d, e** and **f** across the entire dPAG region were 1.613%, 2.846%, and 0.729%,
respectively. Thus, the original schematic was not ideal. Although our initial statistical
analysis had already considered neurons with incomplete morphological filling, we
have now rechecked and recalculated the data. The updated proportion of double-
positive neurons was $2.02 \pm 0.99\%$. A new schematic (**Fig. c**) was generated, and the
revised values have been updated in this revision.

**a**, Schematic from the original Fig. 4b. White arrows indicate double-positive neurons; blue arrows
 indicate partially overlapping yet nondouble-positive neurons. **b**, Full-view image from the original
 manuscript showing all labelled neurons in the dPAG region. **c**, Updated schematic now presented
 in the revised **Fig. 4b**. **d**, Full-view image corresponding to the new schematic. **e** and **f**, Two
 additional full-view images showing all labelled neurons within the dPAG region.

- **Results, line 380-381: I stand on my first impression that activation of SC neurons**
**evoke backing-away with fixe duration followed by an absence of movements that**
**last until the end of the blue-light pulse.**

We thank the reviewer for reiterating this point, which has allowed us to clarify a key
aspect of our findings. We appreciate the reviewer's interpretation, and we agree that
the primary sequence of events upon blue light onset is indeed a backing away episode
followed by a period of movement arrest.

However, based on our detailed analysis of the continuous behavioural trajectories
during the entire duration of optogenetic stimulation (5, 10, and 15 s), we observed that
the behaviour was more dynamic. While stopping is the dominant state, brief, sporadic
episodes of backing away can and do occur intermittently throughout the blue light
pulse rather than being strictly confined to the initial fixed-duration period.

Therefore, the critical finding we wish to emphasize is that the total time encompassing
both backing away and stopping (punctuated by sporadic backing) scales with the
duration of the blue light pulse. We have revised the description in the manuscript
according to your suggestion prevent any misunderstanding for future readers. Please
see **the Results, line 474-480**: "*Optogenetic activation of SC_{dPAG} neurons typically*
*evoked an initial backing away episode, which was often followed by a predominant*
*period of movement cessation that lasted until the end of the light pulse. However, the*
*detailed analysis revealed that sporadic backing away could occur intermittently*
*during the entire stimulation period (Fig. 4g1). Notably, the total duration of the evoked*
*behavioural response (combining backing away and stopping) was positively*
*correlated with the duration of the blue light pulse (Fig. 4g3)."*

- Results, line 442-444: ???

In the study by Li et al., we confirmed that the neurons within the T_{eAdPAG} region that
control running-related behaviours are CaMKII α -positive. However, during our
validation of T_{eAdPAG} neurons, we found that in addition to CaMKII α -positive neurons
(accounting for 58%), a small proportion of the neurons were SOM-positive (5%).
Although T_{eAdPAG} CaMKII α neurons are involved in running behaviour, other
T_{eAdPAG} neuronal subtypes may also participate in its regulation.

The results presented in Fig. 5i show that T_{eAdPAG} SOM neurons are not involved in
the circuitry controlling running behaviour. Therefore, based on our current and
previous findings, we conclude that rebound running is mediated by T_{eAdPAG} CaMKII
neurons and SC_{dPAG} SOM neurons activated by SC_{dPAG} CaMKII neurons.

Revisions corresponding to this content have been made in **the Results section, line**

**540-550**: “Previous studies have confirmed that T_{eAdPAG} CaMKII α neurons are

responsible for running behaviour. Although more than half of T_{eAdPAG} neurons are

CaMKII α -positive, a small subset consists of SOM neurons (H.L., J.C.). Therefore,

other T_{eAdPAG} neuronal subtypes could participate in the regulation of running

behaviour. The results shown in Fig. 5i indicate that T_{eAdPAG} SOM neurons are not

involved in the circuit controlling running behaviour. Based on these findings, we

conclude that rebound running behaviour is triggered by the discharge of T_{eAdPAG}

CaMKII α neurons following the cessation of blue light stimulation of SC_{dPAG} CaMKII α

neurons. During blue light stimulation, T_{eAdPAG} CaMKII α neurons were inhibited—an

effect resulting from the activation of SC_{dPAG} neurons, which further led to the

activation of SOM neurons in the $dPAG$.”.

**- Results, line 443-444: This is a valid assumption but there is no direct evidence**
**showing that SOM neurons innervate TeAdPAG CaMKII+ neurons.**

We apologize, but the discussion regarding the relationship between SOM and TeAdPAG
*CaMKII α* neurons was already included in the Discussion section in our previous
version. While we believe that the indirect evidence presented in Fig. 5 strongly
suggests that SOM neurons innervate TeAdPAG CaMKII α neurons, we fully
acknowledge—as you rightly suggested—the lack of direct evidence for such a synaptic
connection. In this revision, we have removed the corresponding statement and have
revised the relevant section accordingly (as detailed in our response to your previous
comment).

**- Results, paragraph 453-470: lack of logical link and rational for the experiments.**

**The only difference between the experimental protocols of Fig 6b1-2 and Fig 4d-f**
**is that in the former, only SC CaMKII + neurons projecting to dPAG express**
**ChR2 whereas in the latter all projecting neurons are labelled. First, it is not**
**immediately clear why the authors want to do this experiment to explain the**
**mechanism of backing away evoked by the stimulation of SCdPAG CaMKII+**
**neurons. Second, the interpretation does not seem entirely correct to me. It cannot**
**be concluded from the latter experiment that SC neurons make inhibitory**
**connection to SOM SC neurons. The only conclusion that can be made is that**
**stimulating all SC neurons projecting to dPAG results in less excitation of dPAG**
**neurons responsible for backing away. That could as well result from local**
**mechanisms in SC. In my opinion all this part is unnecessary to the story and just**
**bring confusion. The key experiment is the one presented in Fig 6f-g.**

We agree with the reviewer's comment. Considering that the conclusions drawn from

the original Fig. 6a–c have already been substantiated in Fig. 4, these results are not
essential for our upcoming validation of whether scdPAG neurons directly control
backwards movement. Therefore, we have removed this portion of the content and
retained the original Fig. 6d–l.

**Minor:**

- **Figure 6b3: impossible to see the difference between Backing away and Rebound**
**running.**

This set of data has been removed and is no longer discussed.

- **Results, line 492-493: “the above experiments suggests that CaMKII α in the**
**SCdPAG may play a key role in inducing backing away behaviours.”. No! that**
**‘SCdPAG CaMKII positive neurons’ may play a key role. CaMKII is just a maker.**
**There is no indication that it may play a role in this study.**

You are correct; we made an error in our description. We have now revised the text
accordingly in **the Results, line 569-570: “Thus far, the evidence from the above**
*experiments suggests that scdPAG CaMKII α neurons may play a key role in inducing*
*backing away behaviours.”*

**Sylvain Crochet**

**Reviewer #2 (Remarks to the Author):**

**The authors put forth a great effort to address previous concerns and comments.**

**They have significantly improved the figures and statistics, and the text of the**

**manuscript is also greatly improved. It is appreciated that they removed reference**

**to defensive behavior. While the relevance of some of the behavior to more**

**naturalistic settings is a minor concern, the experiments are elegant and the data**

**will be of interest to the neuroscience community.**

We sincerely thank the reviewer for the positive feedback and acknowledge the time

and effort dedicated to evaluating our manuscript. We are pleased that our revisions

have successfully addressed the previous concerns and that the reviewer finds the

experimental approach elegant and the data of potential interest to the neuroscience

community.

We appreciate the reviewer's final endorsement of our work.

**Reviewer #3 (Remarks to the Author):**

**I co-reviewed this manuscript with one of the reviewers who provided the listed**

**reports. This is part of the Nature Communications initiative to facilitate training**

**in peer review and to provide appropriate recognition for Early Career**

**Researchers who co-review manuscripts.**

Dear Reviewer,

The line numbers referenced in our point-by-point responses correspond to the "Revised manuscript without track changes-chen.pdf" file.

REVIEWER COMMENTS

Reviewer #1 (Remarks to the Author):

In this revised version of the manuscript, the authors have taken some of my comments into account. The second part of the study, which breaks down the neural circuits involved in running and backing away (Figures 3 to 6), is now clear and convincing. However, I still have doubts about the first part (juxtacellular recordings in the TEA and dPAG). I maintain that the method for measuring the correlation between neural activity and running speed—by realigning each running event with the preceding neural event (increase in the firing rate) with varying delays—is flawed and carries a high risk of spurious correlations that are not properly accounted for (e.g., by using random permutations between neural activity and running speed) (see Harris 2020: <https://doi.org/10.1101/2020.11.29.402719>). This is particularly problematic given the timescale: lines 196-198 state, "The time difference between a firing event and its corresponding running event, calculated from their onset, ranged from 0.3 to 5.1 s." So much can happen in the brain during such long periods of time. This should also be considered in light of the escape latency of less than 2 s observed in the related study. I expect that the correlation analysis as conducted in this study would yield similar results regardless of where neuronal activity was recorded from. I remain convinced that the most appropriate method for assessing correlation is to define the time lag from the peak of the cross-correlogram. This defined and fixed time lag for each neuron should then be used to realign neuronal activity and running velocity.

Thank you very much for your careful review of the revised manuscript. In this revision, for each neuron, we first performed a cross-correlation analysis between its firing rate and running speed. We then used the peak of the cross-correlation (r_{\max}) to define a determined, fixed time lag for that neuron's activity relative to the running behavior. Finally, we shifted the running speed curve forward in time by this fixed lag to align it with the neural activity. Based on this alignment, we replotted the firing rate–running speed relationship and quantified the functional relationship between the two.

To address concerns about spurious correlations, we incorporated permutation testing into the cross-correlation analysis. By randomly shuffling the temporal relationship between firing rate and running speed, we established a null distribution. This allowed us to exclude neurons whose correlation was not statistically significant (i.e., potentially spurious), ensuring that only neurons with a statistically significant activity-behavior association were included in the final analysis. Individual neurons that did not meet the inclusion criteria were excluded. Furthermore, to address the limited sample size, we incorporated additional neuronal recordings into this re-analysis. The current sample size is 36 neurons in the TeA region and 30 neurons in the dPAG region.

For the aligned data, we maintained the previously described nonlinear function for fitting. The theoretical rationale for choosing this specific function and model comparisons has been elaborated in the revised "Results" section.

[revised manuscript text omitted]

Fig.1 j, Cross-correlation function (CCF) analysis of a representative neuron. The significance of the r_{\max} was assessed using a permutation test with 1,000 random circular shuffles of the running speed trace. The observed r_{\max} was statistically significant ($P < 10^{-4}$, two-tailed permutation test). **k**, Firing rate plotted against the original running speed ($r = 0.844$, $P < 10^{-4}$). **l**, Firing rate versus running speed plot aligned based on the optimal time lag. Red line: fit with a single-phase association equation (nonlinear regression, $P < 10^{-4}$). **m**, CCF analyses for all R-neurons ($n = 36$ neurons). The black line represents the mean CCF curve; grey lines indicate individual neuronal CCF traces. **n**, Distribution of the maximum correlation coefficient (r_{\max}) from the analysis of **Fig. 1m**. The red dashed line represents the Gaussian fit curve ($n = 36$ cells, $R^2 = 0.791$, $P = 2.68 \times 10^{-6}$). **o**, Distribution of optimal time lag counts from the analysis of **Fig. 1m** ($R^2 = 0.788$, $P = 1.32 \times 10^{-7}$). **p**, Fitted curves for firing rate versus running speed across all recorded R-neurons ($n = 36$ cells). $P < 10^{-4}$ for all fits (nonlinear regression). Bottom right panel: Count distribution of the coefficient of determination (R^2) between the firing rate curves and running speed curves. The red dotted line represents the Gaussian fit curve ($R^2 = 0.759$; $n = 36$ cells; $P = 0.008$). **q**, Fitted curves of peak firing rate versus peak running speed for all R-neurons. $R^2 = 0.875$, $P < 10^{-4}$ (nonlinear regression). **r**, Three-dimensional scatter plot of f_0 , v_m , and τ_f . Red line: linear fit curve, $P < 10^{-4}$.

For dPAG section: **the result, Line 207-273**: “*The relationship between the firing rate of the example neuron (Fig. 2b) and the running speed was analyzed using CCF*

[revised manuscript text omitted]

Fig.2 h, The same as **Fig. 1j**, CCF analysis of a representative neuron. The observed r_{max} was statistically significant ($P < 10^{-4}$, two-tailed permutation test). **i**, Firing rate plotted against the optimized running speed ($r = 0.744$, $P < 10^{-4}$). **j**, Firing rate versus running speed plot aligned based on the optimal time lag. Red line: fit with a single-phase association equation (nonlinear regression, $P < 10^{-4}$). **k**, CCF analyses for all R-neurons ($n = 30$ neurons). The black line represents the mean CCF curve; grey lines indicate individual neuronal CCF traces. **l**, Distribution of r_{max} from the analysis of **Fig. 2k**. (Gaussian fit curve for the dPAG: $R^2 = 0.829$, $P = 2.37 \times 10^{-5}$). **m**, Distribution of the optimal time lag counts from the analysis of **Fig. 2k** (Gaussian fit curve for the dPAG; $R^2 = 0.767$, $P = 3.86 \times 10^{-6}$). **n**, Fitted curves for firing rate versus running speed across all recorded R-neurons ($n = 30$ cells). $P < 10^{-4}$ for all fits (nonlinear regression). Bottom right panel: Count distribution of the coefficient of determination (R^2) between the firing rate curves and running speed curves. The red dotted line represents the Gaussian fit curve ($R^2 = 0.842$; $n = 30$ cells; $P = 6.87 \times 10^{-7}$). **o**, Fitted curves of peak firing rate versus peak running speed for all R-neurons. $R^2 = 0.844$, $P < 10^{-4}$ (nonlinear regression). **p**, τ values from the firing–running function (τ_f) (error bars: SDs). **q**, Three-dimensional scatter plot of v_m , f_0 , and τ values for dPAG (black) ($v_m = 1.43 \times \tau + 7.37$, $P < 10^{-4}$) and TeA (grey) neurons; data from the latter were originally from **Fig. 1r**. **r**, Normalized data for all recorded TeA (red) and dPAG neurons (blue) with fitted curves (TeA: $P < 10^{-4}$; dPAG: $P < 10^{-4}$). **s**, Distribution of normalized values, obtained by setting the measured maximum speed value

to its mean and f_0 to zero for each event. Left panel, X-axis values were normalized by shifting f_0 to zero. $t_{(64)} = 0.011$, $P = 0.991$, two-sample t test. Right panel, Y-axis values were normalized by scaling (avg. measured v_m /measured v_m) for TeA and dPAG neurons. $t_{(64)} = 1.643$, $P = 0.105$, two-sample t test. TeA $n = 36$ cells; dPAG = 30 cells.

Furthermore, the way the results are reported still contains some inaccuracies and imprecisions:

- The objective of the study remains unclear: the title and introduction are centered on "running speed coding" - which, in my opinion, is the weak point of the article - while two-thirds of the article actually deals with the TeA/SC/dPAG circuits that control the different behaviors observed.

Thank you for your feedback regarding the clarity of the research objectives. We have now refined our research aims and narrative logic based on the original manuscript and implemented revisions. A brief summary is provided below:

We have reframed the title, abstract, and introduction to unify the research objectives:

New Title: *Running and backing away share a speed coding by neuronal firing in segregated neural circuits*

The Abstract and Introduction now frame the narrative from the perspective of hierarchical signal transmission within a neural circuit—comprising an upstream driver and a downstream actuator—specifically focusing on how behavioral intensity information, such as speed of running, is encoded and transmitted through neuronal firing in the underlying circuit. In the introduction, we position the previously studied (H.L., J.C.) TeA to dPAG pathway that drives running behavior as the foundational circuit model for our investigation. The presence of running-related neurons (R-neurons) in TeA establishes a basis for addressing the fundamental question of how motor signals are quantitatively translated into action outputs at the cellular level. The present study

aims to elucidate the quantitative control logic and functional integration architecture within the motor control hierarchy. We first investigate: in the TeA-dPAG pathway, is there a common quantitative encoding rule shared between the neural activities of the upstream (TeA) and downstream (dPAG) components? Does this rule constitute a causal mechanism for driving behaviour? Furthermore, as a key hub for integration and execution, dPAG also receives various upstream inputs, such as from the superior colliculus (SC), which likewise mediates escape running behaviour. This raises a deeper architectural question: in the SC-dPAG pathway, how is the upstream signal from SC transmitted to the dPAG? Does it follow the same quantitative encoding rule? Moreover, do distinct signals from TeA and SC converge onto the same neuronal population to execute the same behaviour, or are they processed via functionally segregated microcircuits to achieve coordinated or distinct behavioural outputs?

In the revised version, we have moved away from strongly emphasizing the definition that “R-neurons encode running speed.” In Figures 1 and 2, we now adopt a more cautious description: the firing of R-neurons exhibits a sustained covariation with speed, and we established a quantitative model (the single-phase association equation) to describe this relationship. To test whether this relationship is causal—i.e., whether firing encodes running speed—we performed optogenetic manipulations in Figure 3. We found that actively manipulating neurons within the TeA-dPAG circuit could drive running behavior, and the relationship between stimulation frequency (corresponding to neuronal firing rate) and speed could be fitted by the same quantitative function. This indicates that TeA_{dPAG} neurons and TeA_{dPAG} neurons in the TeA-dPAG pathway are consistent with the conclusion that neuronal firing encodes running speed.

Summary of the results section:

Figures 1 & 2: We confirmed that R-neurons in both TeA and dPAG employ the same firing rate-based function model to quantify speed.

Figure 3: Using optogenetic manipulations, we validated that TeA_{dPAG} neurons and TeA_{dPAG} neurons, which likely constitute subsets of the previously recorded TeA and dPAG R-neurons, are indeed a class of neurons that encode running speed. Furthermore, by stimulating TeA $\text{CaMKII}\alpha$ ChR2^+ neurons and recording the evoked firing rates in their downstream dPAG ChR2^- neurons, we were able to elucidate how motor signals are synaptically transmitted downstream within this neural circuit.

Figure 4: Turning to the SC-dPAG pathway, which also mediates running behavior, we addressed a fundamental question regarding the organizational architecture of the dPAG as an integrative hub: How does it process different inputs? We therefore characterized the SC-dPAG pathway to determine whether it employs the same quantitative encoding model and, more importantly, whether inputs from TeA and SC converge onto the same or distinct neuronal ensembles within the dPAG.

Figures 5 & 6: Building on this, we investigated the neuronal circuit mechanisms by which the SC-dPAG and TeA-dPAG pathways mediate distinct behaviors.

Through the above reframing, the narrative arc of this study is now unified as follows: starting from a quantitative behavioral correlation, transforming it into a tool for probing the internal circuit architecture of a complex brain region, and ultimately revealing the microcircuit principles by which the dPAG implements action selection. This revision establishes the circuit mechanism as the ultimate scientific goal of the article, while affirming the indispensable strategic value of the speed-encoding analysis in the first half.

We are grateful for your insightful comments, which have significantly enhanced the clarity and impact of our work.

- In several places, the reported mean values are incorrect. This appears to be due to the authors calculating the mean values by fitting the distributions to a normal

distribution, even when the observed distribution is clearly not normal. This is particularly evident in Figure 1f, where the distribution is clearly bimodal and certainly cannot be accurately modeled by a single normal distribution. This results in an overestimation of the mean value of r_{\max} , whereas a significant proportion of neurons exhibit a much weaker correlation. The authors should report the true mean or median values. It is also difficult to reconcile these high correlation values with the authors' emphasis on variability between trials (events) (see lines 168–171).

In this revision, we have discontinued the practice of estimating the mean by fitting a normal distribution. For all relevant parameters, including r_{\max} and optimal time lag, among others, we now directly report the true arithmetic mean \pm standard deviation, calculated from the raw data. These changes have been consistently updated throughout the main text, figure legends, and related tables.

Your observation regarding the bimodal distribution of r_{\max} in the original Figure 1f is correct. We performed cross-correlation analysis on all neurons, which yielded the initial distribution of r_{\max} . This distribution was indeed non-normal (potentially bimodal or skewed), reflecting heterogeneity in correlation strength across the neuronal population. In this revision, we applied permutation testing to evaluate the statistical significance of the r_{\max} value for each neuron. Only those neurons whose correlation passed the permutation test (i.e., $p < 0.05$) were classified as “running-related neurons” and included in subsequent population-level analyses and mean calculations. Following this rigorous statistical screening, the r_{\max} values of the retained, significantly correlated neuronal sub-population conformed to normality. All current analyses—including cross-correlation, time-lag determination, and r_{\max} computation—are now based on continuous, whole-recording time-series data. No further subdivision into “trials” or “events,” nor any comparison of variability across such subdivisions, is involved. Consequently, the related statements and the potential logical tension you noted in lines 168–171 of the original manuscript are no longer present in the revised version.

We have removed the relevant paragraphs to ensure consistency in our narrative.

- Figure 1r and 2l plots the fittings of neuronal activity for 42 trials from 29 cells (1r) and 30 trials from 22 cells (2l), which suggest that the fitted parameters (τ , v_m and f_0) were estimated from only 1 trial (running event) for many cells ! This further calls into question the reliability of these analyses.

Thank you for raising an important point regarding the reliability of the fitted parameters. You noted that in the original analysis, the parameters for many neurons were estimated based on only a single trial (running event), which could indeed compromise the robustness of the conclusions. In this revision, we have addressed this issue through several key improvements.

Following your earlier suggestion, we first performed cross-correlation analysis on all neurons, supplemented by permutation tests. Only "running-related neurons" showing a statistically significant correlation with running behavior ($p < 0.05$) were selected for subsequent quantitative modeling. This pre-screening step ensures that our analysis focuses on a neuronal population with a stable and significant association with the behavior. Building on this, we prioritized fitting analysis for neurons with multiple valid trials (≥ 2). The current analysis is based on 36 TeA neurons (78 trials) and 30 dPAG neurons (65 trials).

However, we must acknowledge an inherent methodological challenge in this study: due to limitations in recording stability, some R-neurons were captured with only a limited number of valid trials. If this limited trial count had significantly compromised parameter reliability, we would have expected high variability in the estimated τ values across the entire population. In contrast, as shown in the Figure. 2p, the distribution is relatively concentrated, and calculation of the coefficient of variation ($CV = (\text{standard deviation} / \text{absolute mean}) \times 100\%$) reveals low values for both TeA and dPAG conditions (TeA $CV = 23.16\%$; dPAG $CV = 15.49\%$). This indicates that the τ estimates

are consistent across neurons and that our conclusions are not unduly influenced by neurons with fewer trials. More importantly, we independently validated the reliability of the key parameter τ through optogenetic manipulation at the population level. Specifically, optogenetic stimulation repeatedly elicited speed-dependent behavioral outputs, and the standardized τ value derived from this approach (Fig. 3l) closely matches the τ value obtained from neuronal activity recorded during the mouse's spontaneous running (spontaneous running τ vs optogenetic stimulation τ : TeA: 36.86 vs 35.7; dPAG: 18.36 vs 18.13). This critical evidence demonstrates that the movement speed encoding parameter τ derived from our population analysis is robust and reliable, despite the limitations in trial counts for individual neurons.

In summary, we have significantly strengthened the reliability of our conclusions by implementing stricter pre-screening, expanding the dataset, and, most importantly, anchoring the key findings in optogenetic causal experiments. We are grateful for your comments, which have led to more rigorous analyses.

- When looking at the PETHs presented in figure 1d2 and 2d2, it is not easy to be convinced that there is indeed a clear causal relationship between the neuronal activity and running onset.

As mentioned earlier, we performed cross-correlation analysis and permutation testing on the neurons, which allowed us to identify running-related neurons. For these selected neurons, we regenerated event-related firing activity heatmaps. In the new Figure. 1f and 2f, each row represents the trial-averaged firing rate change of one neuron across all valid trials. As shown in the figure, before the running onset (time = 0 s), a clear and significant increase in synchronized population activity can be observed.

Fig. 1f, Top: Spontaneous running-related activity in R-neurons without sensory stimuli. Heatmaps show firing rates of all recorded R-neurons during spontaneous running, aligned to running onset (0 s; green dashed line). Each row corresponds to the averaged activity of a single R-neuron across all spontaneous running trials ($n = 36$ neurons). The color bar represents firing rate intensity (Hz). Middle: PETH for all R-neurons aligned to the onset of spontaneous running. Bottom: Running speed of each neuron aligned to running onset (each trace is the average of all trials presented during the recording of that neuron; $n = 36$ neurons). **2f**, The same as in **Fig. 1f**, an analysis of the dPAG neuronal population ($n = 30$ neurons for spontaneous running).

To further characterize the properties of these neurons, we also display—side by side in the same figure—their response heatmaps to sensory stimulation (Fig. 1e and 2e). This additionally confirms the functional specificity of this neuronal population.

Fig. 1e, Responses of all TeA R-neurons to three types of sensory stimuli: sound, light, and air puff. Top: Heatmaps depicting the firing rates of R-neurons in response to the three sensory stimuli. Activity is aligned to stimulus onset (1 s) and trials with running (speed ≥ 0.5 cm/s) are excluded. Each row represents the average activity of a single R-neuron across all trials. The color bar on the right indicates firing rate intensity (Hz). ($n = 36$ neurons for sound, light and air puff, respectively) Middle, Population peri-stimulus time histograms (PSTHs) for all R-neurons in response to sound, light, and air puff stimuli, respectively. Bottom, Running speed aligned to stimulus onset for each neuron (each trace is the average across all trials presented during that neuron's recording; $n = 36$ neurons for sound, light, air puff, respectively). The blue bars represent stimulus period duration: 1 s. **2e**, The same as **Fig. 1e**, analyses of the dPAG neuronal population ($n = 30$ neurons for sound, light and air puff, respectively). The blue bars indicate the stimulus period (duration: 1 s).

Minors:

- Lines 352-356: this is again just a valid hypothesis but this is not supported by any data. There is no indication that TeA CaMIIa neurons that do not project to dPAG excite TeAdPAG neurons. They may as well project to other structures, such SC?

Thank you for your insightful comment. We fully acknowledge the possibility you

raised that TeA CaMKII α neurons might project to other structures, such as the superior colliculus (SC). This hypothesis is indeed reasonable, given that SC projections to PAG also drive running and are supported by existing literature. (Evans, D. A. et al. A synaptic threshold mechanism for computing escape decisions. *Nature* 558, 590–594 (2018)). We would like to clarify that the hypothesis we initially proposed stemmed from findings demonstrated in another manuscript of ours, where we showed that ^{sens}TeA neurons—which receive projections from the sensory cortex in the TeA—project to TeA_{dPAG} neurons. To strengthen the rationale for this speculation, we have now incorporated a reference to this previous study in the revised version.

In the Results section, line 350-353: “*Notably, the activation of TeA CaMKII α neurons differed from that of TeA_{dPAG} fibres, TeA_{dPAG} CaMKII α neurons, and TeA_{dPAG} neurons, as the latter three involved projections from the TeA to the dPAG, whereas TeA CaMKII α neurons included TeA_{dPAG} neurons and neurons projecting to TeA_{dPAG} neurons (H.L., J.C.).*”

In response to your comment, we have also expanded the discussion to address this alternative hypothesis. **In the Discussion section, line 755-775:** “*In this study, optogenetic stimulation of TeA following injection of AAV-CaMKII α -ChR2 successfully induced running behaviour (Fig. 3a), with a higher running speed compared to stimulation of TeA_{dPAG} neurons (Fig. 3f). We tend to consider that one of the major pathways through which TeA CaMKII α neurons drive running is their direct projection to dPAG (TeA_{dPAG} neurons). Of course, we cannot rule out the possibility that optogenetic stimulation simultaneously activates other neural circuits within TeA, such as neurons from different sensory cortices that project to TeA and, in turn, innervate TeA_{dPAG} neurons (H.L. J.C.), or TeA neurons projecting to other brain regions. For example, studies have shown that activation of SC can also drive running¹¹. However, the focus of this study and the data directly support the sufficiency and necessity of the TeA to dPAG pathway. Whether TeA CaMKII α neurons directly regulate SC or influence running through other indirect pathways requires further investigation using*

circuit-specific tracing and manipulation tools in future studies. More importantly, this study discovered and demonstrated that running driven by the SC to dPAG pathway differs significantly in behavioural pattern from that driven by the TeA to dPAG pathway (Fig. 6j). Inputs from upstream regions such as TeA or SC converge onto distinct microcircuits and cell types in dPAG, thereby generating difference behaviours. This does not, however, negate the role of SC in driving running, because as illustrated in Fig. 4, differences in the precise location of stimulation within SC, as well as in the frequency and duration of blue-light stimulation, could all contribute to the observed behavioural variations.”

- Figure 3 i: we would like to see more than just one example trial for the characterization of dPAG ChR2⁺ and dPAG ChR2⁻ neurons.

We have followed your suggestion and supplemented five representative example trials each for dPAG ChR2⁺ and dPAG ChR2⁻ neurons in the revised manuscript. These additional example data have been compiled into a new supplementary figure (Supplemental Fig. 1).

Supplemental Fig 1

Supplemental Figure 1 | Optogenetic manipulation of dPAG neurons via direct activation versus indirect disynaptic inputs from TeA neurons. **a**, In vivo loose-patch recordings from dPAG neurons expressing ChR2 (ChR2⁺) (**Fig. 3e1**). **b**, In vivo loose-patch recordings from dPAG neuron not expressing ChR2 (ChR2⁻) (*TeA fibre*dPAG) (**Fig. 3b1**). The duration of the light stimulation pulse was 10 ms.

- **Discussion.** An alternative interpretation would be the existence of 2 only behavioural units in the dPAG ('running' and 'backing away') with inhibition of

one onto the other through SOM interneurons resulting in 3 possible behaviors depending on the balance between the two: backing away, stopping and running.

Thank you for this insightful comment. We agreed on the alternative interpretation: the existence of two opposing behavioral units in the dPAG, “running” and “backing away” with one inhibiting the other. This is indeed the fundamental circuit logic our study establishes. This line of reasoning enables us to propose a more refined and mechanistic explanation: Regarding “stopping”: Our data suggest it is not a mere ‘balance’ but the specific outcome when the “backing away” unit is active enough to fully suppress the “running” unit, yet not active enough to drive overt backing away motion. This refines “balance” into an active, unidirectional suppression. Regarding the fourth state — “rebound running”: This state emerges naturally from the transient dynamics of the same unidirectional inhibitory circuit (e.g., post-inhibitory rebound), providing a circuit-based prediction that a steady-state “balance” model might not explicitly generate.

Thus, your framework provides a powerful and simplified theoretical perspective, which our detailed circuit model complements and extends. We have now revised the entire manuscript accordingly and expanded the discussion to include an exploration of this content, elucidating how our mechanistic model can account for the four fundamental states we observed.

The discussion section, line 629-675: *“In this study, T_eAdPAG CaMKII α neurons (Figs. 3e, 4h, and 5j), which are identified by their innervation by TeA neurons (Fig. 6j, red), represent a behavioural unit whose unit behaviour is running. Similarly, $scdPAG$ CaMKII α neurons (Fig. 6j, green), which are driven by SC neurons, form another behavioural unit whose unit behaviour is backing away. Activation of either SC neurons or SC_{dPAG} CaMKII α neurons can excite $scdPAG$ SOM neurons (Fig. 6j, blue). These*

scdPAG SOM neurons, in turn, inhibit the activity of T_{eAdPAG} CaMKII α neurons, thereby generating two behavioural outcomes: stopping and rebound running (Figs. 4e, g; 5b, g; 6e, h), depending on the dynamic balance between the two behavioural units. This classification method enables the separation of specific functional neurons (behavioural units) in a mixed population and the identification of different behaviours (unit behaviours) from previously ambiguous definitions.

*Our data reveal an active motor arrest mechanism in the dPAG mediated by a specific neural circuit. This “stopping” state is not merely the absence of movement but results from the selective inhibition of the “running” behavioural unit (T_{eAdPAG} CaMKII α neurons), driven by the activation of SOM interneurons receiving SC input (Fig. 5). Crucially, during this state, the activity of the “backing away” behavioural unit (*scdPAG* CaMKII α neurons) is not equally suppressed. This indicates that “stopping” represents an equilibrium state under asymmetric competition, rather than a general quiescence of all neuronal activity.*

This distinction raises a central theoretical question: should the SOM neurons executing this crucial inhibitory function be defined as an independent “behavioural unit”? Functionally, their activity is sufficient to trigger the “stopping” state, resembling a “state switch.” However, in terms of coding properties, they do not directly represent any movement parameters (e.g., speed, direction) but only encode the inhibitory strength applied to the “running” behavioral unit. In terms of dependency,

their activity is primarily controlled by the “backing away” behavioral unit, making them more akin to a dedicated inhibitory effector or functional module of that unit. Therefore, we posit that classifying SOM neurons as an “embedded regulatory unit” is more precise than elevating them to a “behavioral unit” on par with the $TeAdPAG$ or $sCdPAG$ driver units. This classification highlights an efficient division of labor within the action selection circuit: a few core driver units can produce diverse discrete behavioral states through specific local regulatory modules.

In summary, we propose a parsimonious action selection model implemented by the $dPAG$: two core driver units, “running” and “backing away”, compete via an asymmetric inhibitory connection executed by SOM neurons. Modulation of the strength of this inhibitory pathway directly determines whether the network output is “backing away,” “stopping,” or “running.” “Rebound running” can be understood as a dynamic overshoot following the sudden release of strong inhibition. This model explains how flexible behavioral outputs can be achieved with limited hardware. Future studies recording the natural activity patterns of SOM neurons in freely behaving animals and manipulating them independently of their upstream drivers will ultimately clarify their computational role in the complete process from decision to action execution.”

- Rebuttal lines 285-290: these are just mean values not statistics supporting the comparisons.

We realized that the original way of presenting the chart was inappropriate and could easily lead to misunderstanding among readers. In this revision, we have changed the visualization format for this data. In the new chart (Fig. 1s), we have adopted a bar chart with individual data points, which better presents the data itself.

s, Distribution of normalized values, obtained by setting the measured maximum speed value to its mean and f_0 to zero for each event. Left panel, X-axis values were normalized by shifting f_0 to zero. $t_{(64)} = 0.011$, $P = 0.991$, two-sample t test. Right panel, Y-axis values were normalized by scaling (avg. measured v_m /measured v_m) for TeA and dPAG neurons. $t_{(64)} = 1.643$, $P = 0.105$, two-sample t test. TeA $n = 36$ cells; dPAG = 30 cells.

- The legends of figure 1 and 2 describe panels c2 and d2 as ‘population neuronal responses’ but I believe that it is in fact several trials from 1 neuron (not a population of neurons).

Thank you for your important comment regarding the accuracy of the figure legends. We also realized that the way population neuronal responses were presented was inappropriate. In this comprehensive revision, we have thoroughly addressed this issue. As mentioned in our previous response, for any figures intended to illustrate population-level neuronal activity patterns, we have now consistently adopted population heatmaps based on trial-averaged activity. A specific example can be found in the new **Fig. 1f and 2f**. This format clearly and intuitively displays the average activity patterns of multiple neurons aligned to behavioral events.

Sylvain Crochet